# Realizable Abstractions: Near-Optimal Hierarchical Reinforcement Learning

## Abstract

The main focus of Hierarchical Reinforcement Learning (HRL) is studying how large Markov Decision Processes (MDPs) can be more efficiently solved when addressed in a modular way, by combining partial solutions computed for smaller subtasks. Despite their very intuitive role for learning, most notions of MDP abstractions proposed in the HRL literature have limited expressive power or do not possess formal efficiency guarantees. This work addresses these fundamental issues by defining Realizable Abstractions, a new relation between generic low-level MDPs and their associated high-level decision processes. The notion we propose avoids non-Markovianity issues and has desirable near-optimality guarantees. Indeed, we show that any abstract policy for Realizable Abstractions can be translated into near-optimal policies for the low-level MDP, through a suitable composition of options. As demonstrated in the paper, these options can be expressed as solutions of specific constrained MDPs. Based on these findings, we propose RARL, a new HRL algorithm that returns compositional and near-optimal low-level policies, taking advantage of the Realizable Abstraction given in the input. We show that RARL is Probably Approximately Correct, it converges in a polynomial number of samples, and it is robust to inaccuracies in the abstraction.

## 1 Introduction

Hierarchical Reinforcement Learning (HRL) is the study of abstractions of decision processes and how they can be used to improve the efficiency and compositionality of RL algorithms (Barto & Mahadevan, 2003; Abel et al., 2018). To pursue these objectives, most HRL methods augment the low-level Markov Decision Process (MDP) (Puterman, 1994) with some form of abstraction, often a simplified state representation or a high-level policy. As was immediately identified (Dayan & Hinton, 1992), compositionality is arguably one of the most important features for HRL algorithms, as it is commonly associated with increased efficiency (Wen et al., 2020) and policy reuse for downstream tasks (Brunskill & Li, 2014; Abel et al., 2018; Tasse et al., 2020; 2022). There is a common intuition that drives many authors in HRL. That is, abstract states correspond to sets of ground states, and abstract actions correspond to sequences of ground actions. This was evident since the early work in HRL (Dayan & Hinton, 1992), and was largely derived from hierarchical planning. However, the main question that still remains unanswered is *which sequence of ground actions should each abstract action correspond to?* The answer to this question also requires the identification of a suitable state abstraction. The resulting notion of MDP abstraction has a strong impact on the applicability and the guarantees of the associated HRL methods.

There is no shared consensus on what "MDP abstractions" should refer to. In the literature, the term is loosely used to refer to a variety of concepts, including state partitions (Abel et al., 2020; Wen et al., 2020), bottleneck states (Jothimurugan et al., 2021b), subtasks (Nachum et al., 2018; Jothimurugan et al., 2021a), options (Precup & Sutton, 1997; Khetarpal et al., 2020), entire MDPs (Ravindran & Barto, 2002; Cipollone et al., 2023), or even the natural language (Jiang et al., 2019). In addition, most HRL methods have been validated only experimentally (Nachum et al., 2018; Jinnai et al., 2020; Jothimurugan et al., 2021b; Lee et al., 2021; 2022b), leading to a limited theoretical understanding of general abstractions and their use in RL. Some notable exceptions (Brunskill & Li, 2014; Fruit et al., 2017; Fruit & Lazaric, 2017; Wen et al., 2020) give formal definitions of state and temporal abstractions, and provide formal near-optimality and efficiency guarantees. However, they do not

define a high-level decision process, enforcing requirements that are often impractical on the ground MDP directly.

In this work, we propose a new formal definition of MDP abstractions, based on a high-level decision process. This definition enables algorithms, such as the one proposed in this paper, that do not require specific knowledge about the ground MDP. In particular, we identify a new relation that links generic low-level MDPs to their high-level representations, and we use a second-order MDP as the high-level decision process in order to overcome non-Markovian dependencies. As we show, the abstractions we propose are widely applicable, do not incur the non-Markovian effects that are often found in HRL (Bai et al., 2016; Nachum et al., 2018; Jothimurugan et al., 2021b), and provide near-optimality guarantees for their associated low-level policies. Such near-optimal policies only result from the compositions of smaller options, without any global constraint. Due to this feature, we call them *Realizable Abstractions*. An important feature of our work is also that we do not restrict the cardinality of the state and action spaces for the ground MDP that does not need to be finite. Instead, we require that the abstract decision process has finite state and action sets, so that we can compute an exact tabular representation of the abstract value function.

We also address the associated algorithmic question of how to learn the ground options that *realize* each high-level behavior. As we show, the realization problem, that is, the problem of learning suitable options from experience, can be cast as a Constrained MDP (CMDP) (Altman, 1999) and solved with off-the-shelf online RL algorithms for CMDPs (Zhang et al., 2020; Ding et al., 2022). Based on these principles, we develop a new algorithm, called *RARL* (for "Realizable Abstractions RL"), which learns compositional policies for the ground MDP and it is Probably Approximately Correct (PAC) (Fiechter, 1994). An additional novelty of this work is that the proposed algorithm iteratively refines the high-level decision process given in input by sampling in the ground MDP and it exploits the solution obtained from the current abstraction to drive exploration in the ground MDP.

**Summary of contributions**   The contributions of this work are theoretical and algorithmic. We propose *Realizable Abstractions* (Definition 2), we show a formal relation between the abstract and the ground values (Theorem 1 and Corollary 3), and we provide original insights on the conditions that must be met to reduce the effective horizon in the abstraction (Proposition 7). Regarding the algorithmic contributions, RARL is sample efficient, PAC, and robust with respect to approximately realizable abstractions and overly optimistic abstract rewards (Theorem 8).

## RELATED WORK

**State-action abstractions**   Similarly to Abel (2020), we organize most of the work in HRL in two groups: state abstractions, which primarily focus on simplified state representations, and action abstractions, which focus on high-level actions and temporal abstractions. From the first group, we mention the MDP homomorphisms (Ravindran & Barto, 2002; 2004), stochastic bisimulation (Givan et al., 2003; Ferns et al.) and the irrelevance criteria listed in Li et al. (2006). These are all related to the language of MDP abstractions used here. However, the limited expressive power of these early works mainly captures specific projections of state features or symmetries in MDP dynamics. As we discuss in Appendix B, our framework extends both MDP homomorphisms and stochastic bisimulations of Givan et al. (2003). In the second group, the *options* framework (Precup & Sutton, 1997; Sutton et al., 1998; 1999) is one of the first to successfully achieve temporal abstraction in MDPs. Options are partial policies, which can be interpreted as abstract actions, and can be fully learned from experience (Bacon et al., 2017; Machado et al., 2017; Khetarpal et al., 2020). Thanks to these properties, many works implemented HRL principles within the theory of options, such as the automatic discovery of landmarks and sub-goals (Simsek & Barto, 2004; Castro & Precup, 2011; Kulkarni et al., 2016; Nachum et al., 2018; Jinnai et al., 2019; Ramesh et al., 2019; Jinnai et al., 2020; Jiang et al., 2022; Lee et al., 2022b). Nonetheless, since the state space is usually not affected by the use of options, the lack of a simplified state representation limits the reuse of previously acquired skills. The study of abstractions that involve both states and actions is the most natural progression for HRL research. Nonetheless, many works in this direction (Ravindran & Barto, 2003; Abel et al., 2020; Abel, 2020; Jothimurugan et al., 2021b; Wen et al., 2020; Infante et al., 2022) only consider a ground MDP and a partition of the state space, without any explicit dynamics at the abstract level. The main difficulty in defining abstract dynamics comes from the non-Markovian and non-stationary effects that often arise in HRL (Jothimurugan et al., 2021b; Gürtler et al., 2021).

This works overcomes these issues and defines MDP abstractions as distinct decision process with independent reward and transition dynamics.

**HRL theory** The theoretical work on HRL is still less developed compared to the empirical studies. The first PAC analysis for RL in the presence of options is by Brunskill & Li (2014). Later, Fruit & Lazaric (2017) and Fruit et al. (2017) strongly contributed to the characterization of the conditions that cause options to be beneficial (or harmful) for learning, such as the near-optimality of options and the reduction in the MDP diameter. Both of these findings are consistent with our results. Lastly, Wen et al. (2020) developed PEP, an HRL algorithm, and derived its regret guarantee. Similarly to our algorithm, PEP learns low-level policies in a compositional way. However, our algorithm does not receive the "exit profiles" in input, which are unlikely to be known in practice.

**Compositional HRL and logic composition** Logical descriptions and planning domains can also be used to subdivide complex dynamics into smaller subtasks. This approach can be seen as a state-action abstraction with associated semantic labels and has been explored in various forms (Dietterich, 2000; Konidaris et al., 2018; Illanes et al., 2020; Jothimurugan et al., 2021a; Lee et al., 2022a), even in purely logical settings (Banihashemi et al., 2017). The main strength of logical abstractions is their suitability for compositionality and skill reuse (Andreas et al., 2017; Jothimurugan et al., 2021a; Neary et al., 2022). However, because of the difficulty of aligning logical representations with stochastic environments and discounted values, most methods do not come with significant near-optimality guarantees for the low-level domain.

**Other algorithms** Regarding the algorithmic contribution, RARL is capable of correcting and adapting to very inaccurate abstract rewards. This feature has been heavily inspired by RL algorithms for multi-fidelity simulators (Cutler et al., 2014; Kandasamy et al., 2016). However, the algorithm cannot update the input mapping function. Unlike other works (Jonsson & Barto, 2006; Allen et al., 2021; Steccanella & Jonsson, 2022), learning such a state partition remains outside the scope of this work.

## 2 PRELIMINARIES

**Notation** With the juxtaposition of sets, as in $\mathcal{X}\mathcal{Y}$ and $\mathcal{X}^k$, we denote the abbreviation of their Cartesian product. Similarly, $xy \in \mathcal{X}\mathcal{Y}$ is preferred to $(x, y)$, when not ambiguous. Sequences, which we write as $x_{i:j}$, are elements of $\mathcal{X}^{j-i+1}$. The set of probability distributions on a set $\mathcal{X}$ is written as $\Delta(\mathcal{X})$. For $x \in \mathcal{X}$, we use $\delta_x \in \Delta(\mathcal{X})$ for the deterministic probability distribution on $x$. For finite $\mathcal{X}$, this is $\delta_x(x') := \mathbb{I}(x' = x)$. The indicator function $\mathbb{I}(\varphi)$ evaluates to 1 if the condition $\varphi$ is true, 0 otherwise. We write $[n]$ for $\{1, \ldots, n\}$. For any surjective function $\phi : \mathcal{S} \to \bar{\mathcal{S}}$ and $\bar{s} \in \bar{\mathcal{S}}$, we define $\lfloor \bar{s} \rfloor_\phi := \{s \in \mathcal{S} \mid \phi(s) = \bar{s}\}$, also written $\lfloor \bar{s} \rfloor$, whenever the function is clear from context.

**Decision Processes** A $k$-order Markov Decision Process ($k$-MDP) is a tuple $\mathbf{M} = \langle \mathcal{S}, \mathcal{A}, T, R, \gamma \rangle$, where $\mathcal{S}$ is a set of states, $\mathcal{A}$ is a set of actions, $0 < \gamma < 1$ is the discount factor, $T : \mathcal{S}^k \times \mathcal{A} \to \Delta(\mathcal{S})$ is the transition function, and $R : \mathcal{S}^k \times \mathcal{A} \to [0, 1]$ is the reward function. To generate the next state or reward, each function receives the last $k$ states in the trajectory. In particular, at each time step $h$, $R(s_{h-k:h-1}, a_h)$ returns the immediate expected reward $r_h$. We denote the initial distribution of $s_0$ with $\mu := T(s_\star^k, a)$, for any $a \in \mathcal{A}$, where $s_\star \in \mathcal{S}$ is some distinguished dummy state. We write the cardinalities of $\mathcal{S}, \mathcal{A}$ as $S, A$. In addition, an MDP is a 1-MDP, for which we can simply write $r_h \sim R(s_{h-1}, a_h)$ and $s_h \sim T(s_{h-1}, a_h)$ (Puterman, 1994). In any $k$-MDP, the value of a policy $\pi$ in some states $s_{0:k-1} \in \mathcal{S}^k$, written $V^\pi(s_{0:k-1})$, is the expected sum of future discounted rewards, when starting from $s_{0:k-1}$, and selecting actions based on $\pi$. The function $Q^\pi(s_{0:k-1}, a_k)$ is the value of $\pi$ when the first action after $s_{0:k-1}$ is set to $a_k$. Without referring to any states, the value of a policy $\pi$ is $V_\mu^\pi := V^\pi(s_\star^k) = \mathbb{E}_{s_0 \sim \mu}[V^\pi(s_\star^{k-1} s_0)]$, which is the value from the initial distribution $\mu$. Every $k$-MDP admits an optimal policy, $\pi^* := \arg\max_\pi V_\mu^\pi$, which is deterministic and Markovian in $\mathcal{S}^k$. Thus, we often consider the set of policies $\Pi := \mathcal{S}^k \to \mathcal{A}$. The optimal value function $V^{\pi^*}$ is also written as $V^*$. Near-optimal policies are defined as follows. For $\varepsilon > 0$, a policy $\pi$ is $\varepsilon$-optimal if $V_\mu^* - V_\mu^\pi \leq \varepsilon$. Generally speaking, Reinforcement Learning (RL) is the problem of learning a (near-)optimal policy in an MDP with unknown $T$ and $R$.

**Constrained MDPs** A *Constrained MDP* (CMDP) (Ross, 1985; Altman, 1999) is defined as a tuple $\mathbf{M} = \langle \mathcal{S}, \mathcal{A}, T, R, \{R_i\}_{i \in [m]}, \{l_i\}_{i \in [m]}, \gamma \rangle$, where $\langle \mathcal{S}, \mathcal{A}, T, R, \gamma \rangle$ forms an MDP and each

$R_i : \mathcal{S} \times \mathcal{A} \to [0, 1]$ is an auxiliary reward function with an associated $l_i \in [0, 1]$. Given a CMDP $\mathbf{M}$, let $V_{\mu,i}^\pi$ be the value of $\pi$ in the MDP $\langle \mathcal{S}, \mathcal{A}, T, R_i, \gamma \rangle$. Then, the set of feasible policies for $\mathbf{M}$ is $\Pi_c \subseteq \Pi$ with

$$\Pi_c := \{\pi \in \Pi \mid V_{\mu,i}^\pi \geq l_i \text{ for each } i \in [m]\} \tag{1}$$

The optimal policy of a CMDP is defined as $\arg\max_{\pi \in \Pi_c} V^\pi$. Near-optimal policies are defined as usual. Extending this relaxation to constraints, we also define the set of $\eta$-feasible policies as: $\Pi_{c,\eta} := \{\pi \in \Pi \mid V_{\mu,i}^\pi \geq l_i - \eta \text{ for each } i \in [m]\}$. To capture negative cost functions, some works do not restrict auxiliary rewards to the $[0, 1]$ range. However, this will be enough for our purposes.

**Occupancy measures** The state occupancy measure of any policy $\pi$, is the discounted probability of reaching some $s$, when starting from some previous state $s_p$, and selecting actions with $\pi$. That is, $d_s^\pi(s \mid s_p) := (1 - \gamma) \sum_{t=0}^\infty \gamma^t \mathbb{P}(s_t = s \mid s_0 = s_p, \pi)$. Similarly, the state-action occupancy is $d_{sa}^\pi(sa \mid s_p) := d_s^\pi(s \mid s_p) \pi(a \mid s)$. The value function of any policy can be expressed as a scalar product between $d_{sa}^\pi$ and the reward function. Using the vector notation, this is $V^\pi(s) = \langle d_{sa}^\pi(s), R \rangle / (1 - \gamma)$, and from the initial distribution, $V_\mu^\pi = \langle V^\pi, \mu \rangle$. Often, we will simply write both distributions as $d^\pi$. For simplicity, the notation we use is specific to discrete distributions. However, values and occupancy measures remain well defined for continuous state and action spaces.

**Options** An *option* is a temporally extended action (Sutton et al., 1998), defined as $o = \langle \mathcal{I}_o, \pi_o, \beta_o \rangle$, where $\mathcal{I}_o \subseteq \mathcal{S}^2$ is an initiation set composed of pairs of states, $\pi_o \in \Pi$ is the policy that $o$ executes and $\beta_o : \mathcal{S} \to \{0, 1\}$ is a termination condition. An option is applicable at some $a_{t-1} r_{t-1} s_{t-1} a_t r_t s_t$ if $s_{t-1} s_t \in \mathcal{I}_o$. With respect to the classic definition (Sutton et al., 1998), we have extended the initiation sets to pairs, instead of single states. In this work, we focus on a specific class of options, called $\phi$-relative options (Abel et al., 2020). For clarity, we recall that $\lfloor \cdot \rfloor_\phi$ and $\lfloor \cdot \rfloor$ denote the inverse image of $\phi$. Given some surjective $\phi : \mathcal{S} \to \bar{\mathcal{S}}$, an option $o = \langle \mathcal{I}_o, \pi_o, \beta_o \rangle$ is said to be $\phi$-*relative* if there exists two distinct $\bar{s}_p, \bar{s} \in \bar{\mathcal{S}}$ such that $\mathcal{I}_o = \lfloor \bar{s}_p \rfloor_\phi \times \lfloor \bar{s} \rfloor_\phi$, $\beta_o(s) = \mathbb{I}(s \notin \lfloor \bar{s} \rfloor_\phi)$, and $\pi_o \in \Pi_{\bar{s}}$, where $\Pi_{\bar{s}} := \lfloor \bar{s} \rfloor_\phi \to \mathcal{A}$ is the set of policies defined for the block. In essence, $\phi$-relative options always start in some block and terminate as soon as the block changes. Any set of options $\Omega$ is $\phi$-relative iff all of its options are. In the remainder of this paper, we only consider sets $\phi$-relative options. Regarding the notation, we use $\Omega_{\bar{s}_p \bar{s}}$ for the set of all options starting in $\lfloor \bar{s}_p \rfloor \lfloor \bar{s} \rfloor$, and $\Omega_{\bar{s}} := \cup_{\bar{s}_p} \Omega_{\bar{s}_p \bar{s}}$ for all options in one block. Finally, we call *policy of options* any set $\Omega$ that contains a single $\phi$-relative option for each pair $\bar{s}_p \bar{s}$. We use this name because $\Omega$ can be fully treated as a policy for the ground MDP. $V^\Omega$ is the value of the policy that always executes the only applicable option in $\Omega$ until each option terminates.

## 3 REALIZABLE ABSTRACTIONS

This section defines Realizable Abstractions and studies the properties they satisfy. To explain the main intuitions behind our abstractions, we use the running example of Figure 1 (left), which represents a ground MDP $\mathbf{M}$, with a simple grid world dynamics. In this work, an MDP abstraction is a pair $\langle \bar{\mathbf{M}}, \phi \rangle$, where $\bar{\mathbf{M}}$ is a decision process over states $\bar{\mathcal{S}}$, and $\phi : \mathcal{S} \to \bar{\mathcal{S}}$ is the state mapping function. In the example, $\bar{\mathcal{S}} = \{\bar{s}_1, \bar{s}_2, \bar{s}_3\}$. The association of abstract states with ground blocks in the partition is intuitive, as shown by the colors in the example. Actions and ground options, on the other hand, are much less trivial to associate. In this work, with Realizable Abstractions, we encode the following intuition: if an abstract transition $(\bar{s}_1, \bar{a}, \bar{s}_3)$ has a high probability of occurring from $\bar{s}_1$ in the abstract model $\bar{\mathbf{M}}$, then there must exist a $\phi$-relative option that moves the agent from block $\lfloor \bar{s}_1 \rfloor$ to block $\lfloor \bar{s}_3 \rfloor$ in the ground MDP, with high probability and in a few steps. In other words, we choose to interpret abstract transitions as representations of what is possible to replicate, we say "realize", in the ground MDP. Both conditions above, in terms of probability and time, are equally important and consistent with the meaning of discounted values in MDPs. Here, we choose $\phi$-relative options because they only terminate after leaving each block. In the example, no direct transition is possible between $\lfloor \bar{s}_2 \rfloor$ and $\lfloor \bar{s}_3 \rfloor$. So, we should have $\mathbb{P}(\bar{s}_3 \mid \bar{s}_2, \bar{a}, \bar{\mathbf{M}}) = 0$, for any $\bar{a}$. Instead, selecting an appropriate value for $\mathbb{P}(\bar{s}_3 \mid \bar{s}_1, \bar{a}, \bar{\mathbf{M}})$ is much more complex, as it strongly depends on the initial state of the option in the gray block. This is at the heart of non-Markovianity in HRL. In this work, we address this issue through a careful treatment of entry states and allowing abstractions to model second-order dependencies. This allows us to condition the probability of the

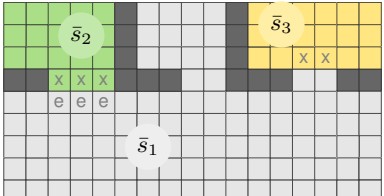 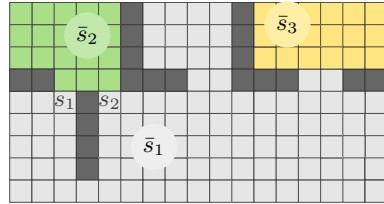

Figure 1: (left) The running example. The ground MDP is a grid world domain and $\bar{S} = \{\bar{s}_1, \bar{s}_2, \bar{s}_3\}$. Each e is an entry in $\mathcal{E}_{\bar{s}_2 \bar{s}_1}$ and each x is an exit in $\mathcal{X}_{\bar{s}_1}$. (right) A different ground MDP.

transition $(\bar{s}_1, \bar{a}, \bar{s}_3)$ on the previous abstract state visited, which might be $\bar{s}_2$ or $\bar{s}_3$ in the example. This modeling advantage motivates our choice of abstract second-order abstract MDPs.

We now proceed to formalize all the previous intuitions. In this work, the abstraction of an MDP $\mathbf{M}$ is a pair $\langle \bar{\mathbf{M}}, \phi \rangle$, where $\bar{\mathbf{M}}$ is a 2-MDP $\bar{\mathbf{M}} = \langle \bar{S}, \bar{\mathcal{A}}, \bar{T}, \bar{R}, \gamma \rangle$ with finite states and actions spaces, and $\phi : \mathcal{S} \to \bar{S}$ is a surjective function. Our formal statements will also be valid when $\bar{\mathbf{M}}$ is an MDP, since it can also be regarded as a 2-MDP with restricted dynamics. For each two distinct abstract states $\bar{s}_p, \bar{s} \in \bar{S}$, we define the set of *entry states* $\mathcal{E}_{\bar{s}_p \bar{s}}$ as the set of ground states in $\lfloor \bar{s} \rfloor$, at which it is possible to enter $\lfloor \bar{s} \rfloor$ from $\lfloor \bar{s}_p \rfloor$. Also, the *exit states* $\mathcal{X}_{\bar{s}}$ contains all ground states outside the block $\lfloor \bar{s} \rfloor$ that are reachable in one transition. In Figure 1 (left), each entry in $\mathcal{E}_{\bar{s}_2 \bar{s}_1}$ is marked with an e, and each exit in $\mathcal{X}_{\bar{s}_1}$ is marked with an x. More generally, $\mathcal{E}_{\bar{s}_p \bar{s}} := \{s \in \lfloor \bar{s} \rfloor \mid \exists s_p \in \lfloor \bar{s}_p \rfloor, \exists a \in \mathcal{A}, T(s \mid s_p, a) > 0\}$ and $\mathcal{X}_{\bar{s}} := \cup_{\bar{s}' \neq \bar{s}} \mathcal{E}_{\bar{s} \bar{s}'}$. Exit and entry states are an intuitive way to discuss the boundaries of contiguous state partitions and are often found in the HRL literature (Wen et al., 2020; Infante et al., 2022). There is one last possibility of entering a block, that is, through the initial distribution $\mu$. However, this specific case is also captured by $\mathcal{E}_{\bar{s}_* \bar{s}}$. The previous abstract state carries fundamental information to characterize entry states, while knowledge of abstract states further back are not nearly as crucial and would make the model significantly more cumbersome. For this reason, we only use 2-MDPs to represent the abstract MDP, and never a k-MDP with k>2. A careful treatment of exits and entries is essential, as it allows us to develop a truly compositional approach where each block is treated separately. For this purpose, associated with each abstract state, we define the *block MDP* as the portion of the original MDP that is restricted to a single block, its exit states and a new absorbing state.

**Definition 1.** Given an MDP $\mathbf{M}$ and $\phi : \mathcal{S} \to \bar{S}$, we define the *block MDP* of some $\bar{s} \in \bar{S}$ as $\mathbf{M}_{\bar{s}} = \langle \mathcal{S}_{\bar{s}}, \mathcal{A}, T_{\bar{s}}, R_{\bar{s}}, \gamma \rangle$, with states $\mathcal{S}_{\bar{s}} := \lfloor \bar{s} \rfloor \cup \mathcal{X}_{\bar{s}} \cup \{s_\perp\}$, where $s_\perp$ is a new absorbing state; the transition function is $T_{\bar{s}}(s, a) := T(s, a)$ if $s \in \lfloor \bar{s} \rfloor$ and $T(s, a) := \delta_{s_\perp}$, otherwise, which is a deterministic transition to $s_\perp$; the reward function is $R_{\bar{s}}(s, a) := R(s, a)$ if $s \in \lfloor \bar{s} \rfloor$ and 0 otherwise.

Therefore, the dynamics of each block MDP remains unchanged while in the relevant block, but is modified to reach the aborbing state with a null reward, from the exits. With respect to analogous definitions from the literature (Fruit et al., 2017), our exit states are *not* absorbing. This small change is essential to preserve the original occupancy distributions at the exits. Now, since any $\phi$-relative option is a complete policy for the block MDP of $\bar{s}$, we will use $d_{\bar{s}}^o$ to denote the state occupancy measure for policy $\pi_o$ in $\mathbf{M}_{\bar{s}}$. Since abstract transitions should only reflect the "external" behavior of the options at the level of blocks, regardless of the specific ground paths followed, we marginalize the occupancy measure with respect to the exit blocks. More precisely, we define the *block occupancy* measure of $\bar{s}$ and $o \in \Omega_{\bar{s}}$ at some $s \in \lfloor \bar{s} \rfloor$ as the probability distribution $h_{\bar{s}}^o(s) \in \Delta(\bar{S} \cup \{s_\perp\})$, with $h_{\bar{s}}^o(\bar{s}' \mid s) := \sum_{s' \in \lfloor \bar{s}' \rfloor} d_{\bar{s}}^o(s' \mid s)$, if $\bar{s}' \neq s_\perp$, and $h_{\bar{s}}^o(s_\perp \mid s) := d_{\bar{s}}^o(s_\perp \mid s)$, otherwise. Block occupancies are perfect candidates for relating the abstract transitions to the options in the ground MDP. Similarly, abstract rewards will be related to the total return accumulated within the block. This second term is exactly captured by $V_{\bar{s}}^o(s)$, the value of the option $o$ in the block MDP of $\bar{s}$. These two elements, $h_{\bar{s}}^o$ and $V_{\bar{s}}^o$, that are relative to the ground MDP, will be related to analogous quantities in the abstraction. Specifically, this work identifies that these quantities should be compared with the probability of the associated abstract transitions and the associated abstract rewards. Expanding these terms for 2-MDPs in any $\bar{s}_p, \bar{s}, \bar{s}' \in \bar{S}$ and $\bar{a} \in \bar{\mathcal{A}}$, with $\bar{s}_p \neq \bar{s}$ and $\bar{s} \neq \bar{s}'$, these are:

$$\tilde{h}_{\bar{s}_p \bar{s} \bar{a}}(\bar{s}') := (1 - \gamma)(\bar{\gamma} \bar{T}(\bar{s}' \mid \bar{s}_p \bar{s}, \bar{a}) + \bar{\gamma}^2 \bar{T}_{\bar{s}_p \bar{s} \bar{a}} \bar{T}(\bar{s}' \mid \bar{s} \bar{s}, \bar{a})) \tag{2}$$

$$\tilde{V}_{\bar{s}_p \bar{s} \bar{a}} := \bar{R}(\bar{s}_p \bar{s}, \bar{a}) + \bar{\gamma} \bar{T}_{\bar{s}_p \bar{s} \bar{a}} \bar{R}(\bar{s} \bar{s}, \bar{a}) \tag{3}$$

with $\bar{T}_{\bar{s}_p\bar{s}\bar{a}} = \frac{\bar{T}(\bar{s}|\bar{s}_p\bar{s},\bar{a})}{1-\bar{\gamma}\,\bar{T}(\bar{s}|\bar{s}\bar{s},\bar{a})}$. Their structure is mainly motivated by the fact that these expressions sum over an indefinite number of self-loops in $\bar{s}$ before transitioning to a different state in the 2-MDP. Equation 2 encodes the discounted cumulative probability of visiting $\bar{s}'$ immediately after $\bar{s}$ in the abstract model, which is $\sum_{t=0}^{\infty}\gamma\mathbb{P}(\bar{s}_t = \bar{s}' \mid \bar{s}_{0:t-1} = \bar{s}, \bar{a}_{1:t} = \bar{a}, \bar{s}_{-1} = \bar{s}_p, \bar{\mathbf{M}})$. This expression is similar to what the occupancy measure captures in the ground MDP, with the difference that the action becomes an option and, instead of a single state, $\bar{s}$ is associated with all states in the block $\lfloor\bar{s}\rfloor$. Expanding the probability above leads to Equation 2. In particular, $\bar{T}_{\bar{s}_p\bar{s}\bar{a}}$ is the result of the geometric series $\bar{T}(\bar{s} \mid \bar{s}_p\bar{s}, \bar{a})\sum_t \gamma^t \bar{T}(\bar{s} \mid \bar{s}\bar{s}, \bar{a})$ that accounts for an indefinite number of self loops in $\bar{s}$. Analogously, Equation 3 is the discounted cumulative return accumulated in $\bar{s}$. All self loops in $\bar{s}$ contribute with a reward of $\bar{R}(\bar{s}\bar{s}, \bar{a})$, which explains the second term in the sum. The first term is the reward achieved after the first transition in $\bar{s}$. Note that if the abstraction is a standard 1-MDP, then $\bar{R}(\bar{s}\bar{s}, \bar{a}) = \bar{R}(\bar{s}_p\bar{s}, \bar{a})$ and $\bar{T}(\bar{s}' \mid \bar{s}_p\bar{s}, \bar{a}) = \bar{T}(\bar{s}' \mid \bar{s}\bar{s}, \bar{a})$ the expressions simplify to:

$$\tilde{h}_{\bar{s}_p\bar{s}\bar{a}}(\bar{s}') := \frac{(1-\gamma)\,\bar{\gamma}\,\bar{T}(\bar{s}' \mid \bar{s}, \bar{a})}{1 - \bar{\gamma}\,\bar{T}(\bar{s} \mid \bar{s}, \bar{a})} \qquad\qquad \tilde{V}_{\bar{s}_p\bar{s}\bar{a}} := \frac{\bar{R}(\bar{s}, \bar{a})}{1 - \bar{\gamma}\,\bar{T}(\bar{s} \mid \bar{s}, \bar{a})} \qquad (4)$$

which does not depend on $\bar{s}_p$.

**Realizable Abstractions**  Using the concepts above, we are now ready to provide a complete description of our MDP abstractions. We say that an abstract action is *realizable* if the behavior described by the abstract transitions and rewards can be replicated (realized) in the ground MDP.

**Definition 2.** Given an MDP $\mathbf{M}$ and an abstraction $\langle\bar{\mathbf{M}}, \phi\rangle$, any abstract tuple $(\bar{s}_p\bar{s}, \bar{a})$, with $\bar{s}_p \neq \bar{s}$, is said $(\alpha, \beta)$-*realizable* if there exists a $\phi$-relative option $o \in \Omega_{\bar{s}_p\bar{s}}$, such that

$$(1 - \gamma)(\tilde{V}_{\bar{s}_p\bar{s}\bar{a}} - V_{\bar{s}}^o(s)) \leq \alpha \qquad (5)$$

$$\tilde{h}_{\bar{s}_p\bar{s}\bar{a}}(\bar{s}') - h_{\bar{s}}^o(\bar{s}' \mid s) \leq \beta \qquad (6)$$

for all $\bar{s}' \neq \bar{s}$ and $s \in \mathcal{E}_{\bar{s}_p\bar{s}}$. The option $o$ is called $(\alpha, \beta)$-*realization* of $(\bar{s}_p\bar{s}, \bar{a})$. An abstraction $\langle\bar{\mathbf{M}}, \phi\rangle$ is said $(\alpha, \beta)$-realizable in $\mathbf{M}$ if any $(\bar{s}_p\bar{s}, \bar{a}) \in \bar{\mathcal{S}}^2 \times \bar{\mathcal{A}}$ with $\bar{s}_p \neq \bar{s}$ also is. A $(0, 0)$-realizable abstraction is *perfectly realizable*.

This definition essentially requires that the desired block occupancy and value, computed from the abstraction, should be similar to the ones that are possible in the ground MDP from each entry state. We observe that if the abstraction is an MDP, as is often the case in the literature (Li et al., 2006; Ravindran & Barto, 2002), eqs. (2) and (3) simplify and our definition can still be applied. Finally, we note that the scale factor of $(1 - \gamma)$ was added to (5) to obtain two parameters $\alpha$ and $\beta$ in the same range of $[0, 1]$, although the appropriate magnitude for such values should scale with $(1 - \gamma)$. Any given $\alpha$ and $\beta$ put a restriction not only on the abstract decision process but also on the possible mapping functions. Indeed, some partitions may not admit any dynamics that satisfies Definition 2 over the induced abstract states. Consider, for example, the ground MDP and the partition of Figure 1 (right). If the grid world is deterministic, there exists an option $o$ for which $h_{\bar{s}_1}^o(\bar{s}_3 \mid s_2) = \gamma^{11} \approx 0.57$, but, due to the higher number of steps required from $s_1$, $h_{\bar{s}_1}^o(\bar{s}_3 \mid s_1) = \gamma^{21} \approx 0.34$. Let $\gamma = 0.95$ and $\bar{\mathbf{M}}$ be so that, for some $\bar{a}$, $\tilde{h}_{\bar{s}_2\bar{s}_1\bar{a}}(\bar{s}_3) = 0.6$. Then, due to the very diverse behaviors from $s_1$ and $s_2$, the tuple $(\bar{s}_2\bar{s}_1\bar{a})$ is not realizable with $\beta = 0.09$ in Figure 1 (right), while it is in the MDP of Figure 1 (left).

The most important feature of our abstractions is that any policy for a Realizable Abstraction can be associated with a near-optimal policy for the ground MDP, which can be expressed as a simple composition of $\phi$-relative options. Indeed, if some abstraction $\langle\bar{\mathbf{M}}, \phi\rangle$ is $(\alpha, \beta)$-realizable, then it is possible to associate to each tuple $(\bar{s}_p\bar{s}, \bar{a})$ the option that realizes it. In the following, we say that some policy of options $\Omega$ is the *realization* of some abstract policy $\bar{\pi}$, if $\Omega$ contains the realization of every tuple $(\bar{s}_p\bar{s}, \bar{\pi}(\bar{s}_p\bar{s}))$. We can finally state the main property below. Its proof is in the appendix.

**Theorem 1.** *Let $\langle\bar{\mathbf{M}}, \phi\rangle$ be an $(\alpha, \beta)$-realizable abstraction of an MDP $\mathbf{M}$. Then, if $\Omega$ is the realization of some abstract policy $\bar{\pi}$, then, for any $\bar{s}_p \in \bar{\mathcal{S}}$, $s_p \in \lfloor\bar{s}_p\rfloor$, $s \in \mathcal{X}_{\bar{s}_p}$, $\bar{s} = \phi(s)$,*

$$\bar{V}^{\bar{\pi}}(\bar{s}_p\bar{s}) - V^{\Omega}(s) \leq \frac{\alpha}{(1-\gamma)^2} + \frac{\beta\,|\bar{\mathcal{S}}|}{(1-\gamma)^2(1-\bar{\gamma})} \qquad (7)$$

*Moreover, if $\bar{\mu}(\bar{s}) = \sum_{s \in \lfloor\bar{s}\rfloor} \mu(s)$ for every $\bar{s}$, the same bound also holds for $\bar{V}_{\bar{\mu}}^{\bar{\pi}} - V_{\mu}^{\Omega}$.*

This bound relates the value of $\bar{\pi}$ in the abstract $\bar{\mathbf{M}}$ with the value of the realization $\Omega$ in the ground $\mathbf{M}$. Essentially, this theorem proves that $(\alpha, \beta)$-realizability is sufficient to have formal guarantees on the minimal value achieved by the realization. Importantly, such a value can be achieved by a composition of options that are only characterized by local constraints on the individual block MDPs. In this light, the value loss in (7) is the maximum cost to be paid for finding a policy as a union of shorter policies, without any global optimization. To evaluate the scale of the bound, we note that $\beta$ and $\alpha$ are in $[0, 1]$, and that $\bar{\gamma} \leq \gamma$. Also, the size of abstract state space $\bar{S} := |\bar{\mathcal{S}}|$ is always finite, and the size of the ground state space, which is usually very large or infinite, does not appear. Theorem 1 does not yet imply near-optimality of $\Omega$ in $\mathbf{M}$, because pessimistic abstractions with null target occupancies and block values would also satisfy Definition 2, trivially. Thus, in addition to realizability, we require that abstractions should be always optimistic, in the following sense.

**Definition 3.** An abstraction $\langle \bar{\mathbf{M}}, \phi \rangle$ of an MDP $\mathbf{M}$ is *admissible* if for any $(\bar{s}_p \bar{s}, \bar{a})$, with $\bar{s}_p \neq \bar{s}$, and any option $o \in \Omega_{\bar{s}_p \bar{s}}$, $\tilde{h}_{\bar{s}_p \bar{s} \bar{a}}(\bar{s}') \geq h_{\bar{s}}^o(\bar{s}' \mid s)$ and $\tilde{V}_{\bar{s}_p \bar{s} \bar{a}} \geq V_{\bar{s}}^o(s)$, for all $\bar{s}' \neq \bar{s}$ and $s \in \mathcal{E}_{\bar{s}_p \bar{s}}$.

As we show below, admissible abstractions provide optimistic estimates of ground values. Together with Theorem 1, this property allows us to guarantee the near-optimality of the realizations in $\mathbf{M}$.

**Proposition 2.** *Let $\langle \bar{\mathbf{M}}, \phi \rangle$ be an admissible abstraction of an MDP $\mathbf{M}$. Then, for any abstract policy $\bar{\pi}$, ground policy $\pi$, it holds $\bar{V}^{\bar{\pi}}(\bar{s}_p \bar{s}) \geq V^\pi(s)$, at any $\bar{s}_p \in \bar{\mathcal{S}}$, $s_p \in \lfloor \bar{s}_p \rfloor$, $s \in \mathcal{X}_{\bar{s}_p}$, $\bar{s} = \phi(s)$.*

**Corollary 3.** *Any realization of the optimal policy of any admissible and $(\alpha, \beta)$-realizable abstraction is $\varepsilon$-optimal, for $\varepsilon = \frac{\alpha(1-\bar{\gamma}) + \beta \bar{S}}{(1-\gamma)^2(1-\bar{\gamma})}$, as long as $\bar{\mu}(\bar{s}) = \sum_{s \in \lfloor \bar{s} \rfloor} \mu(s)$, for all $\bar{s}$.*

Realizable Abstractions are flexible representations that can capture both very coarse state partitions and much more fine-grained subdivisions. For example, on one extreme, we verify that any MDP $\mathbf{M}$ can be abstracted by itself as $\langle \mathbf{M}, \mathrm{I} \rangle$, where $\mathrm{I} : x \mapsto x$ is the identity function.

**Proposition 4.** *Any MDP $\mathbf{M}$ admits $\langle \mathbf{M}, \mathrm{I} \rangle$ as an admissible and perfectly realizable abstraction.*

Although our abstractions are able to represent compressions along the time dimension, they are not restricted to those. As a special case, they can capture any MDP homomorphism (Ravindran & Barto, 2002; 2004) and any stochastic bisimulation (Givan et al., 2003). These two formalisms have equivalent expressive power Ravindran (2004, Theorem 6) and their compression takes place only with respect to parallel symmetries of the state space. We report the main results for MDP homomorphisms here. The relevant definitions and proofs are deferred to Appendix B.

**Proposition 5.** *If $\langle f, \{g_s\}_{s \in \mathcal{S}} \rangle$ is an MDP homomorphism from $\mathbf{M}$ to $\bar{\mathbf{M}}$, then $\langle \bar{\mathbf{M}}, f \rangle$ is an admissible and perfectly realizable abstraction of $\mathbf{M}$.*

**Proposition 6.** *There exists an MDP $\mathbf{M}$ and an admissible and perfectly realizable abstraction $\langle \bar{\mathbf{M}}, \phi \rangle$ for which no surjections $\{g_s\}_{s \in \mathcal{S}}$ exist such that $\langle \phi, \{g_s\}_{s \in \mathcal{S}} \rangle$ is an MDP homomorphism from $\mathbf{M}$ to $\bar{\mathbf{M}}$.*

**Reducing the effective horizon** A reduction in the effective planning horizon, which scales with $(1 - \gamma)^{-1}$ for the ground MDP, can have a very strong impact on learning. As we already know, $\bar{\gamma} \leq \gamma$. Moreover, this inequality can become strict for Realizable Abstractions, as long as the two discount factors satisfy Definition 2. However, to make the relation between $\gamma$ and $\bar{\gamma}$ more explicit, we prove the following proposition.

**Proposition 7.** *If $\langle \bar{\mathbf{M}}, \phi \rangle$ is an admissible abstraction for an MDP $\mathbf{M}$, then, for any tuple $(\bar{s}_p \bar{s}, \bar{a})$ with $\bar{s}_p \neq \bar{s}$, option $o \in \Omega_{\bar{s}_p \bar{s}}$, and $s \in \mathcal{E}_{\bar{s}_p \bar{s}}$, it holds $h_{\bar{s}}^o(\bar{s} \mid s) \geq (1 - \bar{\gamma}) \max\{1, V_{\bar{s}}^o\}$.*

This statement is composed of two results, $h_{\bar{s}}^o(\bar{s} \mid s) \geq 1 - \bar{\gamma}$, which only constrains the occupancy, and $V_{\bar{s}}^o \leq h_{\bar{s}}^o(\bar{s} \mid s)/(1 - \bar{\gamma})$ which also involves value. These inequalities encode the necessary conditions for reducing the effective horizon in the abstraction. The first inequality says that $\bar{\gamma}$ can only be low if the occupancy in every block is high. In particular, if there exists an option $o$ that leaves some $\lfloor \bar{s} \rfloor$ in one step, then $h_{\bar{s}}^o(\bar{s} \mid s) = 1 - \gamma$ and $\bar{\gamma} = \gamma$ is the only feasible choice. The second says that $\bar{\gamma}$ can only be low if $V_{\bar{s}}^o$ is also low with respect to $h_{\bar{s}}^o$. In particular, if $o$ collects in $\lfloor \bar{s} \rfloor$ a reward of 1 at each step, then $V_{\bar{s}}^o = h_{\bar{s}}^o(\bar{s} \mid s)/(1 - \gamma)$ and $\bar{\gamma} = \gamma$ is the only feasible choice. This allows us to conclude that a time compression in the abstraction is possible only if: (i) the changes between blocks occur at some lower timescale; (ii) rewards are temporally sparse. This confirms some common intuitions among the HRL literature, while it shows that sparse rewards are also important.

**Learning realizations**   To conclude this section, we now study how each realizing option can be learned from experience. Although the constraints in Definition 2 could be direcly used, here we propose a slight relaxation that is more suitable for online learning. Specifically, instead of quantifying for each entry state $\mathcal{E}_{\bar{s}_p\bar{s}}$, we consider some initial distribution $\nu \in \Delta(\mathcal{E}_{\bar{s}_p\bar{s}})$.

**Definition 4.** Given an MDP $\mathbf{M}$ and an abstraction $\langle\bar{\mathbf{M}}, \phi\rangle$, an abstract tuple $(\bar{s}_p\bar{s}, \bar{a})$, with $\bar{s}_p \neq \bar{s}$, is $(\alpha, \beta)$-*realizable from* a distribution $\nu \in \Delta(\mathcal{E}_{\bar{s}_p\bar{s}})$, if there exists a $\phi$-relative option $o \in \Omega_{\bar{s}_p\bar{s}}$, such that $\tilde{h}_{\bar{s}_p\bar{s}\bar{a}}(\bar{s}') - h_\nu^o(\bar{s}') \leq \beta$ and $(1-\gamma)(\tilde{V}_{\bar{s}_p\bar{s}\bar{a}} - V_\nu^o) \leq \alpha$, for all $\bar{s}' \neq \bar{s}$, where $h_\nu^o(\bar{s}') \coloneqq \sum_s h_{\bar{s}}^o(\bar{s}' \mid s)\,\nu(s)$ and $V_\nu^o \coloneqq \sum_s V_{\bar{s}}^o(s)\,\nu(s)$. The option $o$ is the realization of $(\bar{s}_p\bar{s}, \bar{a})$ from $\nu$.

Being a relaxed definition, every realization is also a realization from any distribution, but not vice versa. However, given some policy of options $\Omega$, if each $o \in \Omega$ is a $(\alpha, \beta)$-realization of a tuple $(\bar{s}_p\bar{s}, \bar{a})$ from some $\nu$, and $\nu$ is the entry distribution of $\lfloor\bar{s}\rfloor$ from $\lfloor\bar{s}_p\rfloor$ given $\Omega$, then, $\bar{V}_{\bar{\mu}}^{\bar{\pi}} - V_\mu^\Omega$ still satisfy the bound in Theorem 1. The advantage is that, due to marginalization, the terms $h_\nu^o$ and $V_\nu^o$ are no longer dependent on ground states. This means that realizability from distribution consists exactly of $|\bar{\mathcal{S}}|$ constraints, of which $|\bar{\mathcal{S}}| - 1$ come from the block occupancies and one from the value.

In this work, we identify two techniques for learning realizations: by solving Constrained MDPs, or by Linear Programming. The Linear Programming formulation provides insteresting insights and is discussed in Appendix C. However, realizing with Constrained MDPs is the preferred approach, and it is the one discussed here. Unlike standard RL, CMDPs allow the encoding of both soft and hard constraints. This field has received attention because of its relevance for RL and the encoding of hard constraints in safety-critical systems. By expressing the realizability problem as a CMDP, we do not restrict ourselves to a specific technique. Rather, we could realize abstract actions with any online RL algorithm for CMDPs. This is especially relevant since the ground MDP may be non-tabular. Fortunately, there are many general RL algorithms for CMDPs already available (Achiam et al., 2017; Zhang et al., 2020; Ding et al., 2020; 2022; 2023; Wachi et al., 2024).

Among all $\bar{S}$ constraints of Definition 4, we choose to represent the $\bar{S} - 1$ inequalities for the target occupancies as hard constraints and the single inequality for the value as a soft constraint. By assuming the realizability of each abstract tuple, the option $o^* \in \Omega_{\bar{s}_p\bar{s}}$, obtained as the maximization of the soft objective $V_\nu^o$, will satisfy all the $\bar{S}$ original constraints. To express the hard constraints, we observe that $h_\nu^o(\bar{s}') = \sum_{s,s'\in\mathcal{S}} d_{\bar{s}}^o(s' \mid s)\,\mathbb{I}(s' \in \lfloor\bar{s}'\rfloor)\,\nu(s) = (1-\gamma)\,V_{\nu,\bar{s}'}^o$, where $V_{\nu,\bar{s}'}^o$ is the value function of $o$ in the MDP $\langle\mathcal{S}_{\bar{s}}, \mathcal{A}, T_{\bar{s}}, R'_{\bar{s}'}, \gamma\rangle$, with $R'_{\bar{s}'}(s, a) \coloneqq \mathbb{I}(s \in \lfloor\bar{s}'\rfloor)$. The only difference from this MDP and the block MDP $\mathbf{M}_{\bar{s}}$ is that a reward of 1 is placed in $\lfloor\bar{s}'\rfloor$, while every other internal reward is 0. This means that we can reformulate the problem of realizing any tuple $(\bar{s}_p\bar{s}, \bar{a})$ in $\mathbf{M}$ as:

$$\arg\max_{\pi\in\Pi} V_\nu^\pi \qquad s.t. \quad V_{\nu,\bar{s}'}^\pi \geq \frac{\tilde{h}_{\bar{s}_p\bar{s}\bar{a}}(\bar{s}') - \beta}{1-\gamma} \quad \forall\bar{s}' \neq \bar{s} \tag{8}$$

In other words, this is a CMDP with auxiliary reward functions $R'_{\bar{s}'}$ and associated lower limits $l_{\bar{s}'} \coloneqq (\tilde{h}_{\bar{s}_p\bar{s}\bar{a}}(\bar{s}') - \beta)/(1-\gamma)$. Its solution can be seen as a $\phi$-relative option for $\mathbf{M}$.

## 4   RARL: A NEW HRL ALGORITHM

Taking advantage of the properties of Realizable Abstractions, in this section, we develop a new sample efficient HRL algorithm called *RARL* (Realizable Abstractions RL). The algorithm learns a ground policy of options in a compositional way. Moreover, in case the rewards of the input abstraction are strongly overestimated, RARL can correct and update the abstraction accordingly. The complete procedure is shown in algorithm 1. Although we assume that some abstraction $\langle\bar{\mathbf{M}}, \phi\rangle$ is given explicitly, the algorithm only accesses the ground MDP $\mathbf{M}$ through online simulations. The appropriate values for the other input parameters will be described later in Assumptions 1 and 2. Initially, for each abstract tuple, RARL instantiates one individual online RL algorithm for CMDPs. The dictionary $O$, which contains all the realizing options, is initially empty. At convergence, after all relevant tuples have been realized, the algorithm repeatedly executes the lines 8–11. In this exploitation phase, the abstract policy is responsible for selecting the option to execute. During the exploration phase, instead, the algorithm reaches some block for which no option is already known. In this case, the online CMDP solver has full control over the samples collected in block $\lfloor s \rfloor$ (line 13). When the CMDP solver finds a near-optimal option for the block (line 14), as in Assumption 1, the

---

**Algorithm 1:** RARL

**inputs :** MDP simulator $\mathbf{M}$, abstraction $\langle \bar{\mathbf{M}}, \phi \rangle$, and parameters $\alpha, \beta, \zeta$.

**1 foreach** $(\bar{s}_p \bar{s}, \bar{a})$ **do**

  **2** $\quad \mathfrak{A}(\bar{s}_1 \bar{s}_2, \bar{a}_1) \leftarrow \text{REALIZER}(\mathbf{M}, \lfloor \bar{s} \rfloor, \tilde{h}_{s_p sa}, \beta)$  $\quad$ // Online RL algorithm for CMDPs

  **3** $\quad O(\bar{s}_1 \bar{s}_2, \bar{a}_1) \leftarrow null$  $\qquad\qquad\qquad\qquad\qquad$ // policy of options

**4** $s_p \leftarrow s_\star$; $\ s \leftarrow \mathbf{M}.\text{RESET}()$

**5 repeat**

  **6** $\quad \bar{\pi} \leftarrow \text{VALUEITERATION}(\bar{\mathbf{M}}, \frac{1}{1-\gamma} \log \frac{2}{(1-\gamma)\varepsilon})$

  **7** $\quad$ **repeat**

  **8** $\qquad \bar{s}_p \bar{s} \leftarrow \phi(s_p)\,\phi(s)$

  **9** $\qquad \bar{a} \leftarrow \bar{\pi}(\bar{s}_p \bar{s})$

  **10** $\qquad$ **if** $O(\bar{s}_p \bar{s}, \bar{a})$ *is not null* **then**

  **11** $\qquad\quad s_p s \leftarrow \text{ROLLOUT}(\mathbf{M}, O(\bar{s}_p \bar{s}, \bar{a}))$  $\qquad\qquad\qquad$ // until $s \in \mathcal{X}_{\bar{s}}$

  **12** $\qquad$ **else**

  **13** $\qquad\quad s_p s \leftarrow \mathfrak{A}(\bar{s}_p \bar{s}, \bar{a}).\text{ROLLOUT}()$  $\qquad\qquad\qquad$ // until $s \in \mathcal{X}_{\bar{s}}$

  **14** $\qquad\quad$ **if** $\mathfrak{A}(\bar{s}_p \bar{s}, \bar{a}).$*found* **then**

  **15** $\qquad\qquad O(\bar{s}_p \bar{s}, \bar{a}), \hat{V} \leftarrow \mathfrak{A}(\bar{s}_p \bar{s}, \bar{a}).\text{GET}()$

  **16** $\qquad\qquad$ **if** $\tilde{V}_{\bar{s}_p \bar{s} \bar{a}} - \hat{V} > \frac{\alpha}{1-\gamma} + \zeta$ **then**

  **17** $\qquad\qquad\quad \bar{\mathbf{M}} \leftarrow \text{ABSTRACTONER}(\bar{\mathbf{M}}, (\bar{s}_p \bar{s}, \bar{a}), \hat{V} + \frac{\alpha}{1-\gamma} + \zeta)$

  **18** $\qquad\qquad\quad s_p s \leftarrow \text{ROLLOUT}(\mathbf{M})$  $\qquad\qquad\qquad\qquad$ // conclude episode

  **19** $\qquad\qquad\quad$ **break**

  **20** $\qquad\quad s_p s \leftarrow \text{ROLLOUT}(\mathbf{M})$  $\qquad\qquad\qquad\qquad\qquad$ // conclude episode

**21 Function** ABSTRACTONER$(\bar{\mathbf{M}}, (\bar{s}_p \bar{s}, \bar{a}), V)$

  **22** $\quad$ **foreach** $\bar{s}'_p \notin \{\bar{s}_p, \bar{s}\}$ **do** $V^-_{\bar{s}'_p \bar{s} \bar{a}} \leftarrow \tilde{V}_{\bar{s}'_p \bar{s} \bar{a}}$

  **23** $\quad \bar{R}(\bar{s}_p \bar{s}, \bar{a}) \leftarrow \max\{0, \bar{R}(\bar{s}_p \bar{s}, \bar{a}) + V - \tilde{V}_{\bar{s}_p \bar{s} \bar{a}}\}$

  **24** $\quad$ **if** $\bar{R}(\bar{s}_p \bar{s}, \bar{a}) = 0$ **then** $\bar{R}(\bar{s}\bar{s}, \bar{a}) \leftarrow V \left( \frac{\bar{\gamma} \bar{T}(\bar{s}|\bar{s}_p \bar{s}, \bar{a})}{1 - \bar{\gamma} \bar{T}(\bar{s}|\bar{s}\bar{s}, \bar{a})} \right)^{-1}$

  **25** $\quad$ **foreach** $\bar{s}'_p \notin \{\bar{s}_p, \bar{s}\}$ **do** $\bar{R}(\bar{s}'_p \bar{s}, \bar{a}) \leftarrow \min\{1, \bar{R}(\bar{s}'_p \bar{s}, \bar{a}) + V^-_{\bar{s}'_p \bar{s} \bar{a}} - \tilde{V}_{\bar{s}'_p \bar{s} \bar{a}}\}$

---

option is returned, along with its associated value (line 15). These are the estimated $\pi$ and $V^\pi_\nu$ of Eq. (8). Importantly, each tuple is realized at most once. Then, whenever the block value of the realization is below some threshold (line 16), the algorithm calls ABSTRACTONER, which updates the rewards of $\bar{\mathbf{M}}$ to correct for this mismatch. In this case, the break statement triggers a new re-planning in $\bar{\mathbf{M}}$ with Value Iteration, which is executed for the number of iterations specified in input.

**Sample complexity**  In this conclusive section, we provide formal guarantees on the sampe efficiency of RARL. Since the algorithm is modular and depends on the specific CMDP algorithm adopted, we first characterize PAC online algorithms for CMDPs. An RL algorithm $\mathfrak{A}$ is PAC-Safe if, for any unknown CMDP $\mathbf{M}$ and positive parameters $\eta, \zeta$ and $\delta$, whenever $\Pi_c$ is not empty, with probability exceeding $1 - \delta$, $\mathfrak{A}$ returns some $\zeta$-optimal and $\eta$-feasible policy in $\Pi_{c,\eta}$. Moreover, the number of episodes collected from $\mathbf{M}$ must be less than some polynomial in the relevant quantities.

**Assumption 1.** REALIZER is a PAC-Safe online RL algorithm with parameters $\zeta, \eta$ and confidence $1 - \delta/(2\bar{S}^2 \bar{A})$, where $\zeta$ is the input of RARL.

The second assumption ensures that the input abstraction is admissible, and the transition function $\bar{T}$ is $\beta$-realizable. The same is not assumed for rewards, which can be severely overestimated by $\bar{\mathbf{M}}$.

**Assumption 2.** Let $\langle \bar{\mathbf{M}}, \phi \rangle$ and $\beta, \alpha$ be the inputs of RARL. We assume that $\langle \bar{\mathbf{M}}, \phi \rangle$ is admissible and that there exists some admissible $(\alpha, \beta)$-realizable abstraction $\langle \bar{\mathbf{M}}^*, \phi \rangle$, in which $\bar{\mathbf{M}}^*$ only differs from $\bar{\mathbf{M}}$ by its reward function.

The main intuition that we use to prove the sample complexity is that, although sampling occurs in $\mathbf{M}$, all decisions of RARL take place at the level of blocks and high-level states. This allows us to use $\bar{\mathbf{M}}$ as a proxy to refer to the returns that are possible in $\mathbf{M}$. Moreover, thanks to the admissibility ensured by Assumption 2, we can show that RARL is optimistic in the face of uncertainty, because the overestimated rewards of $\bar{\mathbf{M}}$ play the role of exploration bonuses for tuples that have not yet been realized. Finally, to discuss the third and last assumption, we consider each $\nu_{t,\bar{s}_p\bar{s}}$, that represents the entry distribution for block $\lfloor\bar{s}\rfloor$ from $\lfloor\bar{s}_p\rfloor$ at episode $t$. Due to the way the algorithm is constructed, these distributions can remain mostly fixed, and they only depend on the available options in $O$ at the beginning of episode $t$ (let $O_t$ represent this set). Still, since the addition of new options might change such distributions, we assume that the old realizations remain valid in the future, as follows.

**Assumption 3.** During any execution of $RARL$, if $O_t(\bar{s}_p\bar{s},\bar{a})$ is an $(\alpha,\beta)$-realization of $(\bar{s}_p\bar{s},\bar{a})$ in $\bar{\mathbf{M}}^*$ from $\nu_{t,\bar{s}_p\bar{s}}$, then the same is true from $\nu_{t',\bar{s}_p\bar{s}}$, for any $t' > t$.

This is a quite nuanced dependency, and it only arises when learning realizations from specific entry distributions, instead of all entry states. We omit the treatment of this marginal issue here. We can finally state our bound, which limits the sample complexity of exploration (Kakade, 2003) of RARL.

**Theorem 8.** *Under Assumptions 1 to 3, and any positive inputs $\varepsilon, \delta$, with probability exceeding $1 - \delta$, RARL is $\varepsilon'$-optimal with $\varepsilon' = \frac{\alpha(1-\bar{\gamma})+\beta\bar{S}}{(1-\gamma)^2(1-\bar{\gamma})} + \frac{3\varepsilon}{1-\gamma}$ on all but the following number of episodes $\frac{2\bar{S}^2\bar{A}}{\varepsilon}\left(f_{\mathsf{r}}(\zeta,\eta) + \log\frac{2S^2A}{\delta}\right)$, where $f_{\mathsf{r}}(\zeta,\eta)$ is the sample complexity of the realization algorithm.*

Thanks to the compositional property of Realizable Abstractions, the sample complexity of each CMDP learner, which is $f_{\mathsf{r}}(\zeta,\eta)$, only contributes linearly to the bound and scales with the number of tuples to realize. This number, which we bound with $\bar{S}^2\bar{A}$, may be often much smaller because not all tuples are relevant for near-optimal behavior. For example, Wen et al. (2020) shows that HRL has an advantage over *flat* RL when the subMDPs can be grouped into $K$ equivalence classes and $K \ll \bar{S}$. The same argument can be applied here. When two block MDPs are equivalent, they can be regarded as one, the collected samples can be shared, and the resulting options can be used in both blocks. Therefore, the above bound can also be written with a multiplying factor of $\bar{S}\bar{A}K$ instead of $\bar{S}^2\bar{A}$.

## 5 CONCLUSION

This work answers one important open question for HRL regarding how to relate the abstract actions with the ground options. The answer is *to relate the probability of abstract transitions with the probability of the temporally-extended transitions that can be obtained with options in the ground MDP*. More specifically, this is given by the Realizable Abstractions, described in this paper. This notion also implies suitable state abstractions that formally guarantee near-optimal and sample efficient solutions of the ground MDP. In future work, the sample complexity of Theorem 8 could be expressed as an instance-dependent bound. This would highlight when HRL can be more efficient than standard RL, in presence of accurate abstractions, even without relying on equivalence classes.

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

APPENDICES

## A   PROPERTIES OF REALIZABLE ABSTRACTIONS

**Theorem 1.** *Let $\langle \bar{\mathbf{M}}, \phi \rangle$ be an $(\alpha, \beta)$-realizable abstraction of an MDP $\mathbf{M}$. Then, if $\Omega$ is the realization of some abstract policy $\bar{\pi}$, then, for any $\bar{s}_p \in \bar{\mathcal{S}}$, $s_p \in \lfloor \bar{s}_p \rfloor$, $s \in \mathcal{X}_{\bar{s}_p}$, $\bar{s} = \phi(s)$,*

$$\bar{V}^{\bar{\pi}}(\bar{s}_p \bar{s}) - V^{\Omega}(s) \leq \frac{\alpha}{(1 - \gamma)^2} + \frac{\beta \, |\bar{\mathcal{S}}|}{(1 - \gamma)^2 (1 - \bar{\gamma})} \tag{7}$$

*Moreover, if $\bar{\mu}(\bar{s}) = \sum_{s \in \lfloor \bar{s} \rfloor} \mu(s)$ for every $\bar{s}$, the same bound also holds for $\bar{V}_{\bar{\mu}}^{\bar{\pi}} - V_{\mu}^{\Omega}$.*

*Proof.* To relate the value functions of different decision processes, we inductively define a sequence of functions $V_0, V_1, \ldots$ as $V_0(s_p s) := \bar{V}^{\bar{\pi}}(\phi(s_p)\phi(s))$, and, if $k \in \mathbb{N}_+$,

$$V_k(s_p s) := \mathbb{E}\left[ g^o + \gamma^j V_{k-1}(s_{j-1} s_j) \mid s_p s, o \in \Omega \cap \Omega_{\phi(s_p)\phi(s)} \right] \tag{9}$$

where $g^o$ is the cumulative discounted return of the option $o$ and $j$ its random duration. In practice, $V_k$ is the value of executing $k$ consecutive options, then computing the value on the abstraction. Now, with an inductive proof, we show that, for every $k \in \mathbb{N}$, $\bar{s}_p \in \bar{\mathcal{S}}_\star$, $s_p \in \lfloor \bar{s}_p \rfloor$, $s \in \mathcal{X}_{\bar{s}_p}$,

$$\bar{V}^{\bar{\pi}}(\bar{s}_p \bar{s}) - V_k(s_p s) \leq \sum_{i=0}^{k} \gamma^i \, \frac{\alpha (1 - \bar{\gamma}) + \beta \, \bar{S}}{(1 - \gamma)(1 - \bar{\gamma})} \tag{10}$$

where, for this derivation, we are using the syntactic abbreviation $\bar{s} := \phi(s)$ and $\bar{S} := |\bar{\mathcal{S}}|$. For the base case, $k = 0$ and $V_0(s_p s) = \bar{V}^{\bar{\pi}}(\bar{s}_p \bar{s})$. Now, for the inductive step, we apply Lemma 9 and Lemma 10 to the two value functions, respectively. We also use $\bar{T}_{\bar{s}_3 | \bar{s}_1 \bar{s}_2}$ and $\bar{R}_{\bar{s}_1 \bar{s}_2}$, the same abbreviations of Lemma 9. Then,

$$\bar{V}^{\bar{\pi}}(\bar{s}_p \bar{s}) - V_k(s_p s) = \tag{11}$$

$$= \bar{R}_{\bar{s}_p \bar{s}} + \frac{\bar{\gamma} \, \bar{T}_{\bar{s} | \bar{s}_p \bar{s}}}{1 - \bar{\gamma} \, \bar{T}_{\bar{s} | \bar{s}\bar{s}}} \, \bar{R}_{\bar{s}\bar{s}} + \sum_{\bar{s}' \in \bar{\mathcal{S}} \setminus \{\bar{s}\}} \left( \bar{\gamma} \, \bar{T}_{\bar{s}' | \bar{s}_p \bar{s}} + \frac{\bar{\gamma}^2 \, \bar{T}_{\bar{s} | \bar{s}_p \bar{s}} \, \bar{T}_{\bar{s}' | \bar{s}\bar{s}}}{1 - \bar{\gamma} \, \bar{T}_{\bar{s} | \bar{s}\bar{s}}} \right) \bar{V}^{\bar{\pi}}(\bar{s}\bar{s}')$$

$$- \sum_{s' \in \mathcal{S}_{\bar{s}}} \frac{d_{\bar{s}}^o(s' \mid s)}{1 - \gamma} \left( \mathbb{I}(s' \in \lfloor \bar{s} \rfloor) \, R(s', o(s')) + \mathbb{I}(s' \in \mathcal{X}_{\bar{s}}) \, V_{k-1}(s') \right) \tag{12}$$

$$= \bar{R}_{\bar{s}_p \bar{s}} + \frac{\bar{\gamma} \, \bar{T}_{\bar{s} | \bar{s}_p \bar{s}}}{1 - \bar{\gamma} \, \bar{T}_{\bar{s} | \bar{s}\bar{s}}} \, \bar{R}_{\bar{s}\bar{s}} - \sum_{s' \in \lfloor \bar{s} \rfloor} \frac{d_{\bar{s}}^o(s' \mid s)}{1 - \gamma} \, R(s', o(s'))$$

$$+ \sum_{\bar{s}' \in \bar{\mathcal{S}} \setminus \{\bar{s}\}} \left( \bar{\gamma} \, \bar{T}_{\bar{s}' | \bar{s}_p \bar{s}} + \frac{\bar{\gamma}^2 \, \bar{T}_{\bar{s} | \bar{s}_p \bar{s}} \, \bar{T}_{\bar{s}' | \bar{s}\bar{s}}}{1 - \bar{\gamma} \, \bar{T}_{\bar{s} | \bar{s}\bar{s}}} \right) \bar{V}^{\bar{\pi}}(\bar{s}\bar{s}') - \sum_{s' \in \mathcal{X}_{\bar{s}}} \frac{d_{\bar{s}}^o(s' \mid s)}{1 - \gamma} \, V_{k-1}(s') \tag{13}$$

If $V_{\bar{s}}^o$ is the value function of $o$ in the block-restricted MDP $\mathbf{M}_{\bar{s}}$,

$$= \bar{R}_{\bar{s}_p \bar{s}} + \frac{\bar{\gamma} \, \bar{T}_{\bar{s} | \bar{s}_p \bar{s}}}{1 - \bar{\gamma} \, \bar{T}_{\bar{s} | \bar{s}\bar{s}}} \, \bar{R}_{\bar{s}\bar{s}} - V_{\bar{s}}^o(s)$$

$$+ \sum_{\bar{s}' \in \bar{\mathcal{S}} \setminus \{\bar{s}\}} \left( \left( \bar{\gamma} \, \bar{T}_{\bar{s}' | \bar{s}_p \bar{s}} + \frac{\bar{\gamma}^2 \, \bar{T}_{\bar{s} | \bar{s}_p \bar{s}} \, \bar{T}_{\bar{s}' | \bar{s} \bar{s}}}{1 - \bar{\gamma} \, \bar{T}_{\bar{s} | \bar{s} \bar{s}}} \right) \bar{V}^{\bar{\pi}}(\bar{s} \bar{s}') - \sum_{s' \in \mathcal{E}_{\bar{s} \bar{s}'}} \frac{d_{\bar{s}}^o(s' \mid s)}{1 - \gamma} V_{k-1}(s') \right) \tag{14}$$

using the fact that $s' \in \mathcal{E}_{\bar{s} \bar{s}'}$, and $\langle \bar{\mathbf{M}}, \phi \rangle$ is an $(\alpha, \beta)$-realizable abstraction,

$$\leq \frac{\alpha}{1 - \gamma} + \sum_{\bar{s}' \in \bar{\mathcal{S}} \setminus \{\bar{s}\}} \left( \left( \bar{\gamma} \, \bar{T}_{\bar{s}' | \bar{s}_p \bar{s}} + \frac{\bar{\gamma}^2 \, \bar{T}_{\bar{s} | \bar{s}_p \bar{s}} \, \bar{T}_{\bar{s}' | \bar{s} \bar{s}}}{1 - \bar{\gamma} \, \bar{T}_{\bar{s} | \bar{s} \bar{s}}} \right) \bar{V}^{\bar{\pi}}(\bar{s} \bar{s}') - \sum_{s' \in \mathcal{E}_{\bar{s} \bar{s}'}} \frac{d_{\bar{s}}^o(s' \mid s)}{1 - \gamma} V_{k-1}(s') \right) \tag{15}$$

Now, we add and subtract $\sum_{\bar{s}' \in \bar{\mathcal{S}} \setminus \{\bar{s}\}} \sum_{s' \in \mathcal{E}_{\bar{s} \bar{s}'}} \frac{d_{\bar{s}}^o(s' | s)}{1 - \gamma} \bar{V}^{\bar{\pi}}(\bar{s} \bar{s}')$,

$$= \frac{\alpha}{1 - \gamma} + \sum_{\bar{s}' \in \bar{\mathcal{S}} \setminus \{\bar{s}\}} \left( \sum_{s' \in \mathcal{E}_{\bar{s} \bar{s}'}} \frac{d_{\bar{s}}^o(s' \mid s)}{1 - \gamma} \bar{V}^{\bar{\pi}}(\bar{s} \bar{s}') - \sum_{s' \in \mathcal{E}_{\bar{s} \bar{s}'}} \frac{d_{\bar{s}}^o(s' \mid s)}{1 - \gamma} V_{k-1}(s') \right)$$
$$+ \sum_{\bar{s}' \in \bar{\mathcal{S}} \setminus \{\bar{s}\}} \left( \left( \bar{\gamma} \, \bar{T}_{\bar{s}' | \bar{s}_p \bar{s}} + \frac{\bar{\gamma}^2 \, \bar{T}_{\bar{s} | \bar{s}_p \bar{s}} \, \bar{T}_{\bar{s}' | \bar{s} \bar{s}}}{1 - \bar{\gamma} \, \bar{T}_{\bar{s} | \bar{s} \bar{s}}} \right) \bar{V}^{\bar{\pi}}(\bar{s} \bar{s}') - \sum_{s' \in \mathcal{E}_{\bar{s} \bar{s}'}} \frac{d_{\bar{s}}^o(s' \mid s)}{1 - \gamma} \bar{V}^{\bar{\pi}}(\bar{s} \bar{s}') \right) \tag{16}$$

$$= \frac{\alpha}{1 - \gamma} + \sum_{\bar{s}' \in \bar{\mathcal{S}} \setminus \{\bar{s}\}} \sum_{s' \in \mathcal{E}_{\bar{s} \bar{s}'}} \frac{d_{\bar{s}}^o(s' \mid s)}{1 - \gamma} \left( \bar{V}^{\bar{\pi}}(\bar{s} \bar{s}') - V_{k-1}(s') \right)$$
$$+ \sum_{\bar{s}' \in \bar{\mathcal{S}} \setminus \{\bar{s}\}} \bar{V}^{\bar{\pi}}(\bar{s} \bar{s}') \left( \left( \bar{\gamma} \, \bar{T}_{\bar{s}' | \bar{s}_p \bar{s}} + \frac{\bar{\gamma}^2 \, \bar{T}_{\bar{s} | \bar{s}_p \bar{s}} \, \bar{T}_{\bar{s}' | \bar{s} \bar{s}}}{1 - \bar{\gamma} \, \bar{T}_{\bar{s} | \bar{s} \bar{s}}} \right) - \frac{h_{\bar{s}}^o(\bar{s}' \mid s)}{1 - \gamma} \right) \tag{17}$$

applying the inductive hypothesis to the first line and the definition of an $(\alpha, \beta)$-realizable abstraction to the second line,

$$\leq \frac{\alpha}{1 - \gamma} + \sum_{s' \in \mathcal{X}_{\bar{s}}} \frac{d_{\bar{s}}^o(s' \mid s)}{1 - \gamma} \sum_{i=0}^{k-1} \gamma^i \frac{\alpha \, (1 - \bar{\gamma}) + \beta \, \bar{S}}{(1 - \gamma)(1 - \bar{\gamma})} + \frac{\beta \, \bar{S}}{(1 - \gamma)(1 - \bar{\gamma})} \tag{18}$$

It only remains to quantify $\sum_{s' \in \mathcal{X}_{\bar{s}}} d_{\bar{s}}^o(s' \mid s)$. To do this, we apply Lemma 11 which gives,

$$\sum_{s' \in \mathcal{X}_{\bar{s}}} d_{\bar{s}}^o(s' \mid s) = \left( 1 - h_{\bar{s}}^o(\bar{s} \mid s) \right) (1 - \gamma) \tag{19}$$

However, since the option starts in $s \in \lfloor \bar{s} \rfloor$, the occupancy $h_{\bar{s}}^o(\bar{s} \mid s)$ cannot be less than $(1 - \gamma)$. This allows us to complete the inequality and obtain

$$\bar{V}^{\bar{\pi}}(\bar{s}_p \bar{s}) - V_k(s_p s) \leq \frac{\alpha \, (1 - \bar{\gamma}) + \beta \, \bar{S}}{(1 - \gamma)(1 - \bar{\gamma})} + \gamma \sum_{i=0}^{k-1} \gamma^i \frac{\alpha \, (1 - \bar{\gamma}) + \beta \, \bar{S}}{(1 - \gamma)(1 - \bar{\gamma})} \tag{20}$$

$$= \sum_{i=0}^{k} \gamma^i \frac{\alpha \, (1 - \bar{\gamma}) + \beta \, \bar{S}}{(1 - \gamma)(1 - \bar{\gamma})} \tag{21}$$

This concludes the inductive step. To verify eq. (7), we observe that $V^\Omega(s_p, s) = \lim_{k \to \infty} V_k(s_p s)$.

We conclude by verifying the statement for the values from the respective initial distributions.

$$\bar{V}_{\bar{\mu}}^{\bar{\pi}} - V_{\mu}^{\Omega} = \sum_{\bar{s} \in \bar{\mathcal{S}}} \bar{\mu}(\bar{s}) \, \bar{V}^{\bar{\pi}}(\bar{s}_\star \bar{s}) - \sum_{s \in \mathcal{S}} \mu(s) \, V^{\Omega}(s) \tag{22}$$

$$= \sum_{\bar{s} \in \bar{\mathcal{S}}} \bar{\mu}(\bar{s}) \, \bar{V}^{\bar{\pi}}(\bar{s}_\star \bar{s}) - \sum_{\bar{s} \in \bar{\mathcal{S}}} \sum_{s \in \lfloor \bar{s} \rfloor} \mu(s) \, \bar{V}^{\bar{\pi}}(\bar{s}_\star \bar{s})$$
$$+ \sum_{\bar{s} \in \bar{\mathcal{S}}} \sum_{s \in \lfloor \bar{s} \rfloor} \mu(s) \, \bar{V}^{\bar{\pi}}(\bar{s}_\star \bar{s}) - \sum_{s \in \mathcal{S}} \mu(s) \, V^{\Omega}(s) \tag{23}$$

$$= \sum_{\bar{s} \in \bar{\mathcal{S}}} \bar{V}^{\bar{\pi}}(\bar{s}_\star \bar{s}) \, (\bar{\mu}(\bar{s}) - \sum_{s \in \lfloor \bar{s} \rfloor} \mu(s)) + \sum_{s \in \mathcal{S}} \mu(s) \, (\bar{V}^{\bar{\pi}}(\bar{s}_\star \phi(s)) - V^{\Omega}(s)) \tag{24}$$

Using the assumption on initial distributions and the derivation above we obtain the second result. $\quad \square$

**Proposition 2.** *Let $\langle \bar{\mathbf{M}}, \phi \rangle$ be an admissible abstraction of an MDP $\mathbf{M}$. Then, for any abstract policy $\bar{\pi}$, ground policy $\pi$, it holds $\bar{V}^{\bar{\pi}}(\bar{s}_p \bar{s}) \geq V^{\pi}(s)$, at any $\bar{s}_p \in \bar{\mathcal{S}}$, $s_p \in \lfloor \bar{s}_p \rfloor$, $s \in \mathcal{X}_{\bar{s}_p}$, $\bar{s} = \phi(s)$.*

*Proof.* To start, we observe that any ground policy $\pi$ can be equivalently represented as a unique policy of options $\Omega$. Therefore, to relate the values in the two decision processes, we inductively define a sequence of functions $V_0, V_1, \ldots$ as $V_0(s_p s) := \bar{V}^{\bar{\pi}}(\phi(s_p)\phi(s))$, and, if $k \in \mathbb{N}_+$,

$$V_k(s_p s) := \mathbb{E}\left[ g^o + \gamma^j \, V_{k-1}(s_{j-1} s_j) \mid s_p s, o \in \Omega \cap \Omega_{\phi(s_p)\phi(s)} \right] \tag{25}$$

where $g^o$ is the cumulative discounted return of the option $o$ and $j$ its random duration. In practice, $V_k$ is the value of executing $k$ consecutive options, then computing the value on the abstraction. Since $V^{\Omega}(s) = \lim_{k \to \infty} V_k(s_p s)$, to prove the result, it suffices to show that $\bar{V}^{\bar{\pi}}(\bar{s}_p \bar{s}) \geq V_k(s_p s)$, for all $k \in \mathbb{N}$ and every $\bar{s}_p \in \bar{\mathcal{S}}$, $s_p \in \lfloor \bar{s}_p \rfloor$, $s \in \mathcal{X}_{\bar{s}_p}$, $\bar{s} = \phi(s)$. The proof is inductive. For $k = 0$, the base case holds by definition of $V_k$. For $k > 0$, we compute $\bar{V}^{\bar{\pi}}(\bar{s}_p \bar{s}) - V_k(s_p s)$ and expand it as in the proof of Theorem 1. This can be done by respecting all equalities up until

$$\bar{V}^{\bar{\pi}}(\bar{s}_p \bar{s}) - V_k(s_p s) =$$

$$= \bar{R}_{\bar{s}_p \bar{s}} + \frac{\bar{\gamma} \, \bar{T}_{\bar{s}|\bar{s}_p \bar{s}}}{1 - \bar{\gamma} \, \bar{T}_{\bar{s}|\bar{s}\bar{s}}} \, \bar{R}_{\bar{s}\bar{s}} - V_{\bar{s}}^o(s)$$

$$+ \sum_{\bar{s}' \in \bar{\mathcal{S}} \setminus \{\bar{s}\}} \sum_{s' \in \mathcal{E}_{\bar{s}\bar{s}'}} \frac{d_{\bar{s}}^o(s' \mid s)}{1 - \gamma} \left( \bar{V}^{\bar{\pi}}(\bar{s}\bar{s}') - V_{k-1}(s') \right) \tag{26}$$

$$+ \sum_{\bar{s}' \in \bar{\mathcal{S}} \setminus \{\bar{s}\}} \bar{V}^{\bar{\pi}}(\bar{s}\bar{s}') \left( \left( \bar{\gamma} \, \bar{T}_{\bar{s}'|\bar{s}_p \bar{s}} + \frac{\bar{\gamma}^2 \, \bar{T}_{\bar{s}|\bar{s}_p \bar{s}} \, \bar{T}_{\bar{s}'|\bar{s}\bar{s}}}{1 - \bar{\gamma} \, \bar{T}_{\bar{s}|\bar{s}\bar{s}}} \right) - \frac{h_{\bar{s}}^o(\bar{s}' \mid s)}{1 - \gamma} \right)$$

$$= \tilde{V}_{\bar{s}_p \bar{s}\bar{a}} - V_{\bar{s}}^o(s)$$

$$+ \sum_{\bar{s}' \in \bar{\mathcal{S}} \setminus \{\bar{s}\}} \sum_{s' \in \mathcal{E}_{\bar{s}\bar{s}'}} \frac{d_{\bar{s}}^o(s' \mid s)}{1 - \gamma} \left( \bar{V}^{\bar{\pi}}(\bar{s}\bar{s}') - V_{k-1}(s') \right) \tag{27}$$

$$+ \sum_{\bar{s}' \in \bar{\mathcal{S}} \setminus \{\bar{s}\}} \bar{V}^{\bar{\pi}}(\bar{s}\bar{s}') \left( \frac{\tilde{h}_{\bar{s}'_p \bar{s}\bar{a}}(\bar{s}')}{1 - \gamma} - \frac{h_{\bar{s}}^o(\bar{s}' \mid s)}{1 - \gamma} \right)$$

Using the definition of admissible abstractions and the inductive hypothesis, we can confirm that all three terms are positive. $\square$

**Corollary 3.** *Any realization of the optimal policy of any admissible and $(\alpha, \beta)$-realizable abstraction is $\varepsilon$-optimal, for $\varepsilon = \frac{\alpha(1-\bar{\gamma}) + \beta \bar{S}}{(1-\gamma)^2 (1-\bar{\gamma})}$, as long as $\bar{\mu}(\bar{s}) = \sum_{s \in \lfloor \bar{s} \rfloor} \mu(s)$, for all $\bar{s}$.*

*Proof.* Using Proposition 2 and Theorem 1, $V_{\mu}^{\Omega} \geq \bar{V}^{\bar{\pi}}(\bar{s}_p \bar{s}) - \varepsilon \geq V^* - \varepsilon$. $\square$

**Lemma 9.** *In any 2-MDP $\mathbf{M}$ and deterministic policy $\pi$, for any two distinct states $s_p, s \in \mathcal{S}$,*

$$V^{\pi}(s_p s) = R_{s_p s} + \frac{\gamma \, T_{s|s_p s}}{1 - \gamma \, T_{s|ss}} \, R_{ss} + \sum_{s' \in \mathcal{S} \setminus \{s\}} \left( \gamma \, T_{s'|s_p s} + \frac{\gamma^2 \, T_{s|s_p s} \, T_{s'|ss}}{1 - \gamma \, T_{s|ss}} \right) V^{\pi}(ss') \tag{28}$$

*where $T_{s_3|s_1 s_2} := T(s_3 \mid s_1 s_2, \pi(s_1 s_2))$ and $R_{s_1 s_2} := R(s_1 s_2, \pi(s_1 s_2))$.*

*Proof.* We use the abbreviations $T_{s_1|s_1 s_2}$ and $R_{s_1 s_2}$ to avoid excessive verbosity. Then,

$$V^{\pi}(s_p s) = \sum_{s' \in \mathcal{S}} T_{s'|s_p s} \left( R_{s_p s} + \gamma \, V^{\pi}(ss') \right) \tag{29}$$

$$= R_{s_p s} + \sum_{s' \in \mathcal{S} \setminus \{s\}} T_{s'|s_p s} \, \gamma \, V^{\pi}(ss') + T_{s|s_p s} \, \gamma \, V^{\pi}(ss) \tag{30}$$

$$= R_{s_p s} + \sum_{s' \in \mathcal{S} \setminus \{s\}} T_{s'|s_p s} \, \gamma \, V^{\pi}(ss') + T_{s|s_p s} \, \gamma \, R_{ss} \tag{31}$$

$$+ T_{s|s_p s} \, \gamma \, T_{s|ss} \, \gamma \, V^\pi(ss) + T_{s|s_p s} \, \gamma \sum_{s' \in \mathcal{S} \setminus \{s\}} T_{s'|s_p s} \, \gamma \, V^\pi(ss') \tag{32}$$

$$= R_{s_p s} + \gamma \, T_{s|s_p s} \, R_{ss} \sum_{t=0}^\infty \gamma^t \, T_{s|ss}^t$$

$$+ \sum_{s' \in \mathcal{S} \setminus \{s\}} \gamma \, T_{s'|s_p s} \, V^\pi(ss') + \gamma \, T_{s|s_p s} \sum_{s' \in \mathcal{S} \setminus \{s\}} \gamma \, T_{s'|ss} \, V^\pi(ss') \sum_{t=0}^\infty \gamma^t \, T_{s|ss}^t \tag{33}$$

$$= R_{s_p s} + \frac{\gamma \, T_{s|s_p s}}{1 - \gamma \, T_{s|ss}} \, R_{ss} + \sum_{s' \in \mathcal{S} \setminus \{s\}} \left( \gamma \, T_{s'|s_p s} + \frac{\gamma^2 \, T_{s|s_p s} \, T_{s'|ss}}{1 - \gamma \, T_{s|ss}} \right) V^\pi(ss') \tag{34}$$

$\square$

**Lemma 10.** *Consider any MDP $\mathbf{M}$ and surjective function $\phi : \mathcal{S} \to \bar{\mathcal{S}}$. Then, from any state $s \in \mathcal{S}$, the value of any deterministic $\phi$-relative option $o \in \Omega_{\phi(s)}$ and policy $\pi$ is*

$$Q^\pi(s, o) = \sum_{s' \in \mathcal{S}} \frac{d^o_{\phi(s)}(s' \mid s)}{1 - \gamma} \big( \mathbb{I}(s' \in \lfloor \phi(s) \rfloor) \, R(s', o(s')) + \mathbb{I}(s' \in \mathcal{X}_{\phi(s)}) \, V^\pi(s') \big) \tag{35}$$

*where $d^o_{\phi(s)}$ is the state occupancy measure of $\pi_o$ in the block-restricted MDP $\mathbf{M}_{\phi(s)}$.*

*Proof.* Let $\lfloor \bar{s} \rfloor^{(t)} := \lfloor \bar{s} \rfloor^{t-1} \times (\mathcal{S} \setminus \lfloor \bar{s} \rfloor)$ be the set that includes all trajectories leaving the block in exactly $t$ transitions. We also abbreviate $\bar{s} := \phi(s)$. Then,

$$Q^\pi(s, o) = R(s, o(s)) + \gamma \, \mathbb{E}_{s'} [\mathbb{I}(s' \in \lfloor \bar{s} \rfloor) \, Q^\pi(s', o) + \mathbb{I}(s' \notin \lfloor \bar{s} \rfloor) \, V^\pi(s'))] \tag{36}$$

$$= R(s, o(s)) + \gamma \sum_{s' \in \lfloor \bar{s} \rfloor} T(s' \mid s, o(s)) \, Q^\pi(s', o) + \gamma \sum_{s' \notin \lfloor \bar{s} \rfloor} T(s' \mid s, o(s)) \, V^\pi(s') \tag{37}$$

$$= \sum_{t=0}^\infty \gamma^t \sum_{s_{1:t} \in \lfloor \bar{s} \rfloor^t} \mathbb{P}(s_{1:t} \mid s_0 = s, o, \mathbf{M}) \, R(s_t, o(s_t))$$

$$+ \sum_{t=1}^\infty \gamma^t \sum_{s_{1:t} \in \lfloor \bar{s} \rfloor^{(t)}} \mathbb{P}(s_{1:t} \mid s_0 = s, o, \mathbf{M}) \, V^\pi(s_t) \tag{38}$$

$$= \sum_{t=0}^\infty \gamma^t \sum_{s_{1:t} \in \lfloor \bar{s} \rfloor^t} \mathbb{P}(s_{1:t} \mid s_0 = s, o, \mathbf{M}_{\bar{s}}) \, R_{\bar{s}}(s_t, o(s_t))$$

$$+ \sum_{t=1}^\infty \gamma^t \sum_{s_{1:t} \in \lfloor \bar{s} \rfloor^{(t)}} \mathbb{P}(s_{1:t} \mid s_0 = s, o, \mathbf{M}_{\bar{s}}) \, V^\pi(s_t) \tag{39}$$

In the last equation, all probabilities are computed on the block-restricted MDP $\mathbf{M}_{\bar{s}}$. This is equivalent, since all probabilities of transitions from $\lfloor \bar{s} \rfloor$ are preserved. Since every trajectory that leaves the block may only reach $s_\perp$, without further rewards in $\mathbf{M}_{\bar{s}}$, we can simplify as follows.

$$Q^\pi(s, o) = \sum_{t=0}^\infty \gamma^t \sum_{s_{1:t} \in \mathcal{S}_{\bar{s}}^t} \mathbb{P}(s_{1:t} \mid s_0 = s, o, \mathbf{M}_{\bar{s}}) \, R_{\bar{s}}(s_t, o(s_t))$$

$$+ \sum_{t=1}^\infty \gamma^t \sum_{s' \in \mathcal{X}_{\bar{s}}} \mathbb{P}(s_t = s' \mid s_0 = s, o, \mathbf{M}_{\bar{s}}) \, V^\pi(s') \tag{40}$$

$$= \mathbb{E} \left[ \sum_{t=0}^\infty \gamma^t \, r_t \mid s, o, \mathbf{M}_{\bar{s}} \right]$$

$$+ \sum_{s' \in \mathcal{S}} \sum_{t=1}^\infty \gamma^t \, \mathbb{P}(s_t = s' \mid s_0 = s, o, \mathbf{M}_{\bar{s}}) \, \mathbb{I}(s' \in \mathcal{X}_{\bar{s}}) \, V^\pi(s') \tag{41}$$

$$= V_{\bar{s}}^o(s) + \sum_{s' \in \mathcal{S}} \sum_{t=0}^{\infty} \gamma^t \, \mathbb{P}(s_t = s' \mid s_0 = s, o, \mathbf{M}_{\bar{s}}) \, \mathbb{I}(s' \in \mathcal{X}_{\bar{s}}) \, V^{\pi}(s') \tag{42}$$

$$= (1-\gamma)^{-1} \sum_{s' \in \lfloor \bar{s} \rfloor} d_{\bar{s}}^o(s' \mid s) \, R(s', o(s'))$$

$$+ (1-\gamma)^{-1} \sum_{s' \in \mathcal{S}} d_{\bar{s}}^o(s' \mid s) \, \mathbb{I}(s' \in \mathcal{X}_{\bar{s}}) \, V^{\pi}(s') \tag{43}$$

$$= \sum_{s' \in \mathcal{S}} (1-\gamma)^{-1} \, d_{\bar{s}}^o(s' \mid s) \, (\mathbb{I}(s' \in \lfloor \bar{s} \rfloor) \, R(s', o(s')) + \mathbb{I}(s' \in \mathcal{X}_{\bar{s}}) \, V^{\pi}(s')) \tag{44}$$

$\square$

**Lemma 11.** *Let $\mathbf{M}_{\bar{s}}$ be any block MDP, computed from some MDP $\mathbf{M}$, mapping function $\phi$ and abstract state $\bar{s}$. Then, for any option $o \in \Omega_{\bar{s}}$ and $s \in \lfloor \bar{s} \rfloor$, it holds:*

$$d_{\bar{s}}^o(s_{\perp} \mid s) = (1 - h_{\bar{s}}^o(\bar{s} \mid s)) \, \gamma \tag{45}$$

$$\sum_{s' \in \mathcal{X}_{\bar{s}}} d_{\bar{s}}^o(s' \mid s) = (1 - h_{\bar{s}}^o(\bar{s} \mid s)) \, (1 - \gamma) \tag{46}$$

*Proof.* In a block MDP, we remind that the occupancy measure is spread between the block $\lfloor \bar{s} \rfloor$, the exits and the sink state $s_{\perp}$. In other words,

$$\sum_{s' \in \mathcal{X}_{\bar{s}}} d_{\bar{s}}^o(s' \mid s) = 1 - \sum_{s' \in \lfloor \bar{s} \rfloor} d_{\bar{s}}^o(s' \mid s) - d_{\bar{s}}^o(s_{\perp} \mid s) = 1 - h_{\bar{s}}^o(\bar{s} \mid s) - d_{\bar{s}}^o(s_{\perp} \mid s) \tag{47}$$

From the definition of occupancy, we also know that

$$d_{\bar{s}}^o(s_{\perp} \mid s) = (1 - \gamma) \sum_{t=0}^{\infty} \gamma^t \, \mathbb{P}(s_t = s_{\perp} \mid s_0 = s, o, \mathbf{M}_{\bar{s}}) \tag{48}$$

$$= (1 - \gamma) \sum_{t=1}^{\infty} \gamma^t \, \mathbb{P}(s_t = s_{\perp} \mid s_0 = s, o, \mathbf{M}_{\bar{s}}) \tag{49}$$

$$= (1 - \gamma) \sum_{t=1}^{\infty} \gamma^t \, \mathbb{P}(s_{t-1} \in \mathcal{X}_{\bar{s}} \cup \{s_{\perp}\} \mid s_0 = s, o, \mathbf{M}_{\bar{s}}) \tag{50}$$

$$= \gamma (1 - \gamma) \sum_{t=0}^{\infty} \gamma^t \, \mathbb{P}(s_t \in \mathcal{X}_{\bar{s}} \cup \{s_{\perp}\} \mid s_0 = s, o, \mathbf{M}_{\bar{s}}) \tag{51}$$

$$= \gamma \left( \sum_{s' \in \mathcal{X}_{\bar{s}}} d_{\bar{s}}^o(s' \mid s) + d_{\bar{s}}^o(s_{\perp} \mid s) \right) \tag{52}$$

Substituting eq. (47) into eq. (52) gives the result. $\square$

**Proposition 4.** *Any MDP $\mathbf{M}$ admits $\langle \mathbf{M}, \mathrm{I} \rangle$ as an admissible and perfectly realizable abstraction.*

*Proof.* The ground domain is $\mathbf{M} = \langle \mathcal{S}, \mathcal{A}, T, R, \gamma \rangle$ and the abstraction is $\langle \mathbf{M}, \mathrm{I} \rangle$. Since admissibility is trivially satisfied, we just need to show that this is a perfectly realizable abstraction. The identity function induces the naive partitioning, in which each state is in a separate block: $\lfloor s \rfloor_{\mathrm{I}} = \{s\}$. Also, if we just consider deterministic I-relative options, we see that these are simple repetitions of the same action for the same state. We can now compute the un-normalized block occupancy measure at any state $s \in \mathcal{S}$ and deterministic $o \in \Omega_s$. Then, for $s' \neq s$,

$$\frac{h_{\mathrm{I}(s)}^o(s' \mid s)}{1 - \gamma} = \sum_{s' \in \lfloor \mathrm{I}(s') \rfloor} \sum_{t=0}^{\infty} \gamma^t \, \mathbb{P}(s_t = s' \mid s_0 = s, o, \mathbf{M}_{\mathrm{I}(s)}) \tag{53}$$

$$= \sum_{t=1}^{\infty} \gamma^t \, \mathbb{P}(s_{0:t-1} \in \lfloor s \rfloor^t, s_t = s' \mid s_0 = s, o, \mathbf{M}_s) \tag{54}$$

$$= \sum_{t=1}^{\infty} \gamma^t \, T(s \mid s, o(s))^{t-1} \, T(s' \mid s, o(s)) \tag{55}$$

$$= \frac{\gamma \, T(s' \mid s, o(s))}{1 - \gamma \, T(s \mid s, o(s))} \tag{56}$$

Now we compute un-normalized eq. (2) for $\mathbf{M}$. Importantly, since $T(s_p s, a) = T(ss, a)$, we can just write $T(s, a)$:

$$\frac{\tilde{h}_{s_p s a}(s')}{1 - \gamma} = \gamma \, T(s' \mid s, a) + \frac{\gamma^2 \, T(s \mid s, a) \, T(s' \mid s, a)}{1 - \gamma \, T(s \mid s, a)} = \frac{\gamma \, T(s' \mid s, a)}{1 - \gamma \, T(s \mid s, a)} \tag{57}$$

This proves that $\pi_o(s) = a$ is a perfect realization of $a$ with respect to eq. (6). We now consider rewards. The term $V_s^o(s)$, appearing in eq. (5), is the cumulative return obtained by repeating action $a$ (since it is the only reward in $\mathbf{M}_s$).

$$V_s^o(s) = \sum_{t=0}^{\infty} \gamma^t \, \mathbb{P}(s_t = s \mid s_0 = s, a, \mathbf{M}_{\bar{s}}) \, R(s, a) \tag{58}$$

$$= \sum_{t=0}^{\infty} \gamma^t \, T(s \mid s, a)^{t-1} \, R(s, a) \tag{59}$$

$$= \frac{\gamma \, R(s, a)}{1 - \gamma \, T(s \mid s, a)} \tag{60}$$

Following a similar procedure of eq. (57), we also verify eq. (5). $\qquad\square$

**Proposition 7.** *If $\langle \bar{\mathbf{M}}, \phi \rangle$ is an admissible abstraction for an MDP $\mathbf{M}$, then, for any tuple $(\bar{s}_p \bar{s}, \bar{a})$ with $\bar{s}_p \neq \bar{s}$, option $o \in \Omega_{\bar{s}_p \bar{s}}$, and $s \in \mathcal{E}_{\bar{s}_p \bar{s}}$, it holds $h_{\bar{s}}^o(\bar{s} \mid s) \geq (1 - \bar{\gamma}) \max\{1, V_{\bar{s}}^o\}$.*

*Proof.* Let us fix any tuple $(\bar{s}_p \bar{s}, \bar{a})$ with $\bar{s}_p \neq \bar{s}$, $\bar{s}_p, \bar{s} \in \bar{\mathcal{S}}$, option $o \in \Omega_{\bar{s}_p \bar{s}}$, and $s \in \mathcal{E}_{\bar{s}_p \bar{s}}$. Since the abstraction is admissible, we know $\tilde{h}_{\bar{s}_p \bar{s} \bar{a}}(\bar{s}') \geq h_{\bar{s}}^o(\bar{s}' \mid s)$ and $\tilde{V}_{\bar{s}_p \bar{s} \bar{a}} \geq V_{\bar{s}}^o(s)$, for any $\bar{s}' \neq \bar{s}$. Using the abbreviation $\bar{T}_{\bar{s}_3 \mid \bar{s}_1 \bar{s}_2} := \bar{T}(\bar{s}_3 \mid \bar{s}_1 \bar{s}_2, \bar{a})$, we expand the first inequality with (2),

$$h_{\bar{s}}^o(\bar{s}' \mid s) \leq (1 - \bar{\gamma}) \left( \bar{\gamma} \bar{T}_{\bar{s}' \mid \bar{s}_p \bar{s}} + \bar{\gamma}^2 \frac{\bar{T}_{\bar{s}' \mid \bar{s}\bar{s}} \bar{T}_{\bar{s} \mid \bar{s}_p \bar{s}}}{1 - \bar{\gamma} \bar{T}_{\bar{s} \mid \bar{s}\bar{s}}} \right) \tag{61}$$

$$\Leftrightarrow \frac{h_{\bar{s}}^o(\bar{s}' \mid s)}{(1 - \bar{\gamma})} \leq \frac{\bar{\gamma} \bar{T}_{\bar{s}' \mid \bar{s}_p \bar{s}} - \bar{\gamma}^2 \bar{T}_{\bar{s}' \mid \bar{s}_p \bar{s}} \bar{T}_{\bar{s} \mid \bar{s}\bar{s}} + \bar{\gamma}^2 \bar{T}_{\bar{s}' \mid \bar{s}\bar{s}} \bar{T}_{\bar{s} \mid \bar{s}_p \bar{s}}}{1 - \bar{\gamma} \bar{T}_{\bar{s} \mid \bar{s}\bar{s}}} \tag{62}$$

by summing all such inequalities over $\bar{s}' \neq \bar{s}$, and using eq. (46) for the left-hand side, we obtain

$$\Rightarrow 1 - h_{\bar{s}}^o(\bar{s} \mid s) \leq \frac{\bar{\gamma}(1 - \bar{T}_{\bar{s} \mid \bar{s}_p \bar{s}}) - \bar{\gamma}^2 \bar{T}_{\bar{s} \mid \bar{s}\bar{s}}(1 - \bar{T}_{\bar{s} \mid \bar{s}_p \bar{s}}) + \bar{\gamma}^2 \bar{T}_{\bar{s} \mid \bar{s}_p \bar{s}}(1 - \bar{T}_{\bar{s} \mid \bar{s}\bar{s}})}{1 - \bar{\gamma} \bar{T}_{\bar{s} \mid \bar{s}\bar{s}}} \tag{63}$$

$$\Leftrightarrow 1 - \bar{\gamma} \bar{T}_{\bar{s} \mid \bar{s}\bar{s}} - h_{\bar{s}}^o(\bar{s} \mid s)(1 - \bar{\gamma} \bar{T}_{\bar{s} \mid \bar{s}\bar{s}}) \leq \bar{\gamma} - \bar{\gamma} \bar{T}_{\bar{s} \mid \bar{s}_p \bar{s}} - \bar{\gamma}^2 \bar{T}_{\bar{s} \mid \bar{s}\bar{s}} + \bar{\gamma}^2 \bar{T}_{\bar{s} \mid \bar{s}_p \bar{s}} \tag{64}$$

$$\Leftrightarrow h_{\bar{s}}^o(\bar{s} \mid s)(1 - \bar{\gamma} \bar{T}_{\bar{s} \mid \bar{s}\bar{s}}) \geq (1 - \bar{\gamma} \bar{T}_{\bar{s} \mid \bar{s}\bar{s}}) - \bar{\gamma}(1 - \bar{\gamma} \bar{T}_{\bar{s} \mid \bar{s}\bar{s}}) + \bar{\gamma} \bar{T}_{\bar{s} \mid \bar{s}_p \bar{s}}(1 - \bar{\gamma}) \tag{65}$$

$$\Leftrightarrow h_{\bar{s}}^o(\bar{s} \mid s) \geq 1 - \bar{\gamma} + \frac{\bar{\gamma} \bar{T}_{\bar{s} \mid \bar{s}_p \bar{s}}(1 - \bar{\gamma})}{1 - \bar{\gamma} \bar{T}_{\bar{s} \mid \bar{s}\bar{s}}} \tag{66}$$

$$\Rightarrow h_{\bar{s}}^o(\bar{s} \mid s) \geq 1 - \bar{\gamma} \tag{67}$$

For the second statement, we expand $\tilde{V}_{\bar{s}_p \bar{s} \bar{a}} \geq V_{\bar{s}}^o(s)$,

$$V_{\bar{s}}^o(s) \leq \bar{R}(\bar{s}_p \bar{s}, \bar{a}) + \bar{\gamma} \bar{R}(\bar{s}\bar{s}, \bar{a}) \frac{\bar{T}_{\bar{s} \mid \bar{s}_p \bar{s}}}{1 - \bar{\gamma} \bar{T}_{\bar{s} \mid \bar{s}\bar{s}}} \tag{68}$$

$$\Rightarrow V_{\bar{s}}^o(s) \leq 1 + \frac{\bar{\gamma} \bar{T}_{\bar{s} \mid \bar{s}_p \bar{s}}}{1 - \bar{\gamma} \bar{T}_{\bar{s} \mid \bar{s}\bar{s}}} \tag{69}$$

$$\Leftrightarrow (1 - \bar{\gamma}) V_{\bar{s}}^o(s) \leq (1 - \bar{\gamma}) + \frac{\bar{\gamma} \bar{T}_{\bar{s} \mid \bar{s}_p \bar{s}}(1 - \bar{\gamma})}{1 - \bar{\gamma} \bar{T}_{\bar{s} \mid \bar{s}\bar{s}}} \tag{70}$$

using (66),

$$\Rightarrow V_{\bar{s}}^o(s) \leq h_{\bar{s}}^o(\bar{s} \mid s)/(1 - \bar{\gamma}) \tag{71}$$

$\square$

## B  CONNECTION WITH MDP HOMOMORPHISMS AND BISIMULATION

In this section, we relate our new concept of realizable abstractions with two existing formal definitions of MDP abstractions, namely, MDP homomorphisms (Ravindran & Barto, 2002) and stochastic bisimulation (Givan et al., 2003). As we demonstrate in this section, both MDP homomorphisms and stochastic bisimulation are strictly less expressive (meaning, their assumptions are strictly more stringent) than realizable abstractions. As we show below, any MDP abstraction obtained via an homomorphisms or a stochastic bisimulation is also an admissible and perfectly realizable abstraction. On the other hand, there exists MDP-abstraction pairs, $\mathbf{M}$ and $\langle \bar{\mathbf{M}}, \phi \rangle$, that satisfy the realizability assumptions but no associated MDP homomorphisms or bisimulation exists for the two. In an effort to make both MDP homomorphisms and stochastic bisimulation more widely applicable, they have been generalized to approximate the strict relations, respectively in Ravindran & Barto (2004) and Ferns et al.. Nonetheless, they only allow small variations around the strict equality, similarly to what we have done in this paper for realizable abstractions, but they cannot capture very different relations from their original definition. Therefore, we will relate the strict relations of the various formalisms: admissible and perfectly realizable abstractions, MDP homomorphisms, and stochastic bisimulation.

**MDP homomorphisms**  MDP homomorphisms are a classic formalism for MDP minimization (Ravindran & Barto, 2002). A homomorphism from an MDP $\mathbf{M} = \langle \mathcal{S}, \mathcal{A}, T, R, \gamma \rangle$ to another $\bar{\mathbf{M}} = \langle \bar{\mathcal{S}}, \bar{\mathcal{A}}, \bar{T}, \bar{R}, \gamma \rangle$ is a pair $\langle f, \{g_s\}_{s \in \mathcal{S}} \rangle$, with a function $f : \mathcal{S} \to \bar{\mathcal{S}}$ and surjections $g_s : \mathcal{A} \to \bar{\mathcal{A}}$, satisfying

$$\bar{T}(f(s') \mid f(s), g_s(a)) = \sum_{s'' \in \lfloor f(s') \rfloor} T(s'' \mid s, a) \tag{72}$$

$$\bar{R}(f(s), g_s(a)) = R(s, a) \tag{73}$$

for all $s, s' \in \mathcal{S}$, $a \in \mathcal{A}$. For simplicity, here we assumed that all actions are applicable in any state. MDP homomorphisms can also be generalized to be approximate as shown in Ravindran & Barto (2004).

**Proposition 5.** *If $\langle f, \{g_s\}_{s \in \mathcal{S}} \rangle$ is an MDP homomorphism from $\mathbf{M}$ to $\bar{\mathbf{M}}$, then $\langle \bar{\mathbf{M}}, f \rangle$ is an admissible and perfectly realizable abstraction of $\mathbf{M}$.*

*Proof.*  If the ground domain is $\mathbf{M} = \langle \mathcal{S}, \mathcal{A}, T, R, \gamma \rangle$, we choose as abstraction $\langle \bar{\mathbf{M}}, f \rangle$. We compute the un-normalized block occupancy measure at any state $s \in \mathcal{S}$ and deterministic option $o \in \Omega_{f(s)}$. We also assume that $o$ selects the same action for every $\lfloor f(s) \rfloor$. Then, for $\bar{s}' \neq f(s)$,

$$\frac{h_{f(s)}^o(\bar{s}' \mid s)}{1 - \gamma} = \sum_{s' \in \lfloor \bar{s}' \rfloor} \sum_{t=0}^{\infty} \gamma^t \, \mathbb{P}(s_t = s' \mid s_0 = s, o, \mathbf{M}_{f(s)}) \tag{74}$$

$$= \sum_{s' \in \lfloor \bar{s}' \rfloor} \sum_{t=1}^{\infty} \gamma^t \, \mathbb{P}(s_{0:t-1} \in \lfloor f(s) \rfloor^t, s_t = s' \mid s_0 = s, o, \mathbf{M}_{f(s)}) \tag{75}$$

$$= \sum_{t=1}^{\infty} \gamma^t \sum_{s_{0:t-1} \in \lfloor f(s) \rfloor^t} \sum_{s' \in \lfloor \bar{s}' \rfloor} \mathbb{P}(s_{0:t-1}, s_t = s' \mid s_0 = s, o, \mathbf{M}_{f(s)}) \tag{76}$$

Now, summing from $s'$ to $s_{t-1}$ back to $s_0$ and substituting eq. (72),

$$= \sum_{t=1}^{\infty} \gamma^t \, \bar{T}(f(s) \mid f(s) \, g_s(o(s)))^{t-1} \, \bar{T}(\bar{s}' \mid f(s) \, g_s(o(s))) \tag{77}$$

$$= \frac{\gamma \, \bar{T}(\bar{s}' \mid f(s) \, g_s(o(s)))}{1 - \gamma \, \bar{T}(f(s) \mid f(s) \, g_s(o(s)))} \tag{78}$$

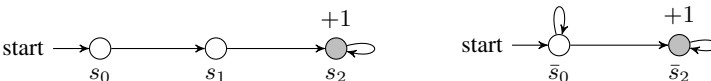

Figure 2: The ground MDP (left) and the abstract MDP (right) used in the proof of Proposition 6.

Now we compute un-normalized eq. (2) for $\mathbf{M}$. As in eq. (57), since $\bar{T}(\bar{s}_p\bar{s}, \bar{a}) = T(\bar{s}\bar{s}, \bar{a})$, we can just write $T(\bar{s}, \bar{a})$ and:

$$\frac{\tilde{h}_{\bar{s}_p\bar{s}\bar{a}}(\bar{s}')}{1-\gamma} = \frac{\gamma\,\bar{T}(\bar{s}' \mid \bar{s}\,\bar{a})}{1 - \gamma\,\bar{T}(\bar{s} \mid \bar{s}\,\bar{a})} \tag{79}$$

This proves that $\pi_o(s) \in g_s^{-1}(\bar{a})$ is a perfect realization of $\bar{a}$ with respect to eq. (6). We now consider rewards. The term $V_{f(s)}^o(s)$, appearing in eq. (5), is

$$V_{f(s)}^o(s) = \sum_{t=0}^{\infty} \gamma^t \sum_{s_{0:t} \in \lfloor f(s)\rfloor^{t+1}} \mathbb{P}(s_{0:t} \mid s_0 = s, o, \mathbf{M}_{f(s)})\, R(s, a) \tag{80}$$

$$= \sum_{t=0}^{\infty} \gamma^t\, \bar{T}(f(s) \mid f(s)\, o(a))^t\, \bar{R}(f(s)\, o(a)) \tag{81}$$

$$= \frac{\gamma\,\bar{R}(f(s)\, o(a))}{1 - \gamma\,\bar{T}(f(s) \mid f(s)\, o(a))} \tag{82}$$

By comparison with $\tilde{V}_{\bar{s}_p\bar{s}\bar{a}}$ in eq. (5), the same choice $\pi_o(s) \in g_s^{-1}(\bar{a})$ also satisfies the second constraint. $\qquad\square$

**Proposition 6.** *There exists an MDP $\mathbf{M}$ and an admissible and perfectly realizable abstraction $\langle \bar{\mathbf{M}}, \phi \rangle$ for which no surjections $\{g_s\}_{s\in\mathcal{S}}$ exist such that $\langle \phi, \{g_s\}_{s\in\mathcal{S}} \rangle$ is an MDP homomorphism from $\mathbf{M}$ to $\bar{\mathbf{M}}$.*

*Proof.* Consider the MDP in Figure 2. This is a very simple MDP $\mathbf{M} = \langle \mathcal{S}, \mathcal{A}, T, R, \gamma \rangle$ with three states $\mathcal{S} = \{s_0, s_1, s_2\}$, an action $\mathcal{A} = \{a_0\}$, deterministic transitions as indicated by the figure, and a reward of $+1$ in state $s_2$, zero otherwise. The state mapping function is constructed as $\phi(s_0) = \phi(s_1) = \bar{s}_0$ and $\phi(s_2) = \bar{s}_1$. The abstract MDP is $\bar{\mathbf{M}} = \langle \{\bar{s}_0, \bar{s}_1\}, \{\bar{a}_0\}, \bar{T}, \bar{R}, \gamma \rangle$, where the reward function returns $+1$ in state $\bar{s}_1$, zero otherwise, and the transition function $\bar{T}$ only allows the transitions indicated by the arrows. The exact values for the stochastic transitions in $\bar{s}_0$ are $\bar{T}(\bar{s}_1 \mid \bar{s}_0, \bar{a}_0) := \gamma/(1+\gamma)$ and $\bar{T}(\bar{s}_0 \mid \bar{s}_0, \bar{a}_0) := 1/(1+\gamma)$.

We now show that $\langle \bar{\mathbf{M}}, \phi \rangle$ is admissible and perfectly realizable. Let us remind that $\bar{s}_\star$ is the dummy state symbol that represents the beginning of an episode in the abstract MDP. Then, we apply the equations in (4) to get:

$$\tilde{h}_{\bar{s}_\star\bar{s}_0\bar{a}_0}(\bar{s}_1) = \frac{(1-\gamma)\,\gamma\,\bar{T}(\bar{s}_1 \mid \bar{s}_0, \bar{a}_0)}{1 - \gamma\,\bar{T}(\bar{s}_0 \mid \bar{s}_0, \bar{a}_0)} = \frac{(1-\gamma)\,\gamma^2/(1+\gamma)}{1 - \gamma/(1+\gamma)} = (1-\gamma)\,\gamma^2 \tag{83}$$

$$\tilde{V}_{\bar{s}_\star\bar{s}_0\bar{a}_0} = \frac{\bar{R}(\bar{s}_0, \bar{a}_0)}{1 - \gamma\,\bar{T}(\bar{s}_0 \mid \bar{s}_0, \bar{a}_0)} = 0 \tag{84}$$

$$\tilde{V}_{\bar{s}_0\bar{s}_1\bar{a}_0} = \frac{\bar{R}(\bar{s}_1, \bar{a}_0)}{1 - \gamma\,\bar{T}(\bar{s}_1 \mid \bar{s}_1, \bar{a}_0)} = \frac{1}{1-\gamma} \tag{85}$$

With one action, there is only one option $o$ for each block that achieve a value of 0 in $\lfloor s_0\rfloor$, since all rewards in $s_0, s_1$ are null, and a value of $1/(1-\gamma)$ in $\lfloor \bar{s}_1\rfloor$, since the rewards from $s_2$ are 1 for an infinite number of steps. Lastly, the block occupancy measure $h_{\bar{s}_0}^o(\bar{s}_1 \mid s_0)$ is also $(1-\gamma)\gamma^2$ because it can only reach $s_2$ after exactly two steps. Since the terms satisfy these equalities, the abstraction is both admissible and perfectly realizable.

To conclude the proof, we show that the state abstraction function $\phi$ prevents the existence of an MDP homomorphism from $\mathbf{M}$ to $\bar{\mathbf{M}}$. Simply, in the abstraction above, the ground states $s_0$ and $s_1$

both belong to the same block. However, this fact contradicts (72), because:

$$\sum_{s \in \lfloor \phi(s_2) \rfloor} T(s \mid s_0, a_0) = T(s_2 \mid s_0, a_0) = 0 \neq 1 = T(s_2 \mid s_1, a_0) = \sum_{s \in \lfloor \phi(s_2) \rfloor} T(s \mid s_1, a_0) \quad (86)$$

Since $\phi(s_0) = \phi(s_1)$, the transition function defined in eq. (72) is undefined for any $g_s : \mathcal{A} \to \bar{\mathcal{A}}$. $\quad \square$

We now turn our attention to stochastic bisimulation. As we will see, the same results above still hold because their expressive power is equivalent to MDP homomorphisms.

**Stochastic bisimulation** Stochastic bisimulation (Givan et al., 2003) is a technique for state minimization in MDPs, inspired by the bisimulation relations for transition systems and concurrent processes. Given an MDP M, it allows to define another MDP, that we write as $\bar{\text{M}}$, in which one or more states are grouped together if they are in the bisimilarity relation. In this section, we study this relation and we formally verify that bisimulation is strictly less expressive (meaning, more stringent), than the realizability condition that this paper proposes.

We first introduce some required notation. Given a binary relation $E \subseteq \mathcal{S} \times \mathcal{S}'$, we write $(s, s') \in E$ if any two $s, s' \in \mathcal{S}$ are in the relation. If every $s \in \mathcal{S}$ and $s' \in \mathcal{S}'$ appears in some pair in $E$, we define $E|\mathcal{S}$ to be the partition of $\mathcal{S}$ obtained by grouping all states that are reachable under the reflexive, symmetric, and transitive closure of $E$. In this section, for $s \in \mathcal{S}$, we write $\lfloor s \rfloor_{E|\mathcal{S}}$ to denote the set in the partition $E|\mathcal{S}$ to which $s$ belongs. $E|\mathcal{S}'$ and $\lfloor s' \rfloor_{E|\mathcal{S}'}$ are defined analogously.

A *stochastic bisimulation* (Givan et al., 2003) over two MDPs $\text{M} = \langle \mathcal{S}, \mathcal{A}, T, R, \gamma \rangle$ and $\text{M}' = \langle \mathcal{S}', \mathcal{A}, T', R', \gamma \rangle$ with the same action space, is any relation $Z \in \mathcal{S} \times \mathcal{S}'$ that satisfies, for any $s \in \mathcal{S}, s' \in \mathcal{S}', a \in \mathcal{A}$:

1. $s$ appears in $E$ and $s'$ appears in $E$;
2. If $(s, s') \in E$, then $R(s, a) = R(s', a)$;
3. If $(s, s') \in E$, then for any $(s_n, s'_n) \in E$,

$$\sum_{s_v \in \lfloor s_n \rfloor_{E|\mathcal{S}}} T(s, a, s_v) = \sum_{s'_v \in \lfloor s'_n \rfloor_{E|\mathcal{S}'}} T'(s', a, s'_v) \quad (87)$$

Despite the different formalisms, stochastic bisimulations over MDPs and MDP homomorphisms have exactly the same expressive power. As proven by the following theorem, an MDP homomorphism exists if and only if This has been proven in a theorem that we report below.

**Proposition 12.** *(Ravindran, 2004, Corollary of Theorem 6) Let $\langle f, \{g_s\}_{s \in \mathcal{S}} \rangle$ be an MDP homomorphism from an MDP M to an MDP $\bar{\text{M}}$. The relation $E \subseteq \mathcal{S} \times \bar{\mathcal{S}}$, defined by $(s, \bar{s}) \in E$ if and only if $\phi(s) = \bar{s}$, is a stochastic bisimulation.*

This exact statement means that the existence of an MDP homomorphism guarantees the existence of a stochastic bisimulation. The opposite implication can be also obtained as a result of Theorem 6 from Ravindran (2004). Specifically, if $E$ is a maximal stochastic bisimulation from M to $\bar{\text{M}}$, then there exits an MDP homomorphism between the two.

## C    REALIZING WITH LINEAR PROGRAMMING

For this alternative approach, we show that the realizability problem can be formulated as a linear program, which may be addressed with primal-dual techniques. This may come as little surprise, since the Lagrangian formulation is one of the possible solution methods for constrained optimization problems such as CMDPs. However, we present these two techniques separately because some CMDP methods may be more closely related to Deep RL algorithms, and they can be quite different from online stochastic optimization algorithms for linear programs. In addition, primal-dual techniques have been studied independently of CMDPs and are often developed as solution methods for unconstrained RL. Recent research focuses on finding near-optimal policies for non-tabular MDPs, both in the presence of generative simulators and online RL (de Farias & Roy, 2003; Mahadevan et al., 2014; Chen & Wang, 2016; Tiapkin & Gasnikov, 2022; Gabbianelli et al., 2024; Neu & Okolo,

2023). The advantage of online optimization is that the typically large linear programs would not be stored explicitly. This is still an open field of study and, similarly to the CMDP formulation above, the formulation we propose here may be solved with any feasible algorithm for this setting.

The linear programming (LP) formulation of optimal planning in MDPs dates back to Puterman (1994); Bertsekas (1995). We first show this classic formulation here and then add the additional constraints. Using vector notation for functions and distributions, we interpret the rewards as a vector $R \in \mathbb{R}^{SA}$ and the initial distribution as $\nu \in \mathbb{R}^S$. Transitions are written as a matrix $P \in \mathbb{R}^{SA \times S}$ where $P(sa, s') := T(s' \mid s, a)$. Let $E \in \mathbb{R}^{SA \times S}$ with $E(sa, s') := \mathbb{I}(s = s')$, be a matrix that copies elements for each action. Then, the planning problem in MDPs is expressed as:

$$\max_{b \in \mathbb{R}^{SA} : b \geq 0} \quad b^T R$$
$$\text{s.t.} \quad E^T b - \gamma P^T b = (1 - \gamma) \nu \tag{88}$$

The constraint expressed here is the Bellman flow equation on the state-action occupancy distribution. At the optimum, the solution $b^*$ is the discounted state-action occupancy measure of the optimal policy, and we have $E^T b^* = d_\nu^{\pi^*}$. In addition, the objective is the scaled optimal value $V^* = \langle b^*, R \rangle / (1 - \gamma)$. The dual linear program is

$$\min_{V \in \mathbb{R}^S} \quad (1 - \gamma) \nu^T V$$
$$\text{s.t.} \quad E V - \gamma P V \geq R \tag{89}$$

and the optimum of this problem is $V^*$, the value of the optimal policy. Solving either the primal or the dual problem is equivalent to solving the given MDP. The references cited above are only some of the works that adopt this linear formulation to find the optimal policy. For generalizing to non-tabular MDPs, the linear formulation is often expressed in feature space (de Farias & Roy, 2003). Here, we work with the tabular equations shown above for simplicity.

The LP formulation just presented can now be applied to each block MDP and modified to introduce the additional constraints. Similarly to our choice for CMDPs, we only express the constraint on occupancy distributions. Due to the equality constraint in (88), the vector $b$ is forced to be a state-action occupancy distribution. Thus, all $\bar{S} - 1$ constraints from (4) can be written in the primal program as $B^T b \geq \tilde{h}_{\bar{s}_p \bar{s} \bar{a}} - \beta$, where $B \in \mathbb{R}^{SA \times (\bar{S}-1)}$ is the matrix that sums all occupancies across states and actions for one block as $B^T(\bar{s}, sa) := \mathbb{I}(\bar{s} = \phi(s))$. The linear program becomes

$$\max_{b \in \mathbb{R}^{SA} : b \geq 0} \quad b^T R$$
$$\text{s.t.} \quad E^T b - \gamma P^T b = (1 - \gamma) \nu \tag{90}$$
$$\qquad -B^T b \leq \beta - \tilde{h}_{\bar{s}_p \bar{s} \bar{a}}$$

Computing the dual of this program we have:

$$\min_{V \in \mathbb{R}^S, \, y \in \mathbb{R}^{\bar{S}-1}, \, y \geq 0} \quad \begin{pmatrix} (1 - \gamma) \nu \\ \beta - \tilde{h}_{\bar{s}_p \bar{s} \bar{a}} \end{pmatrix}^T \begin{pmatrix} V \\ y \end{pmatrix}$$
$$\text{s.t.} \quad \begin{pmatrix} E - \gamma P & -B \end{pmatrix} \begin{pmatrix} V \\ y \end{pmatrix} \geq R \tag{91}$$

We do not need to encode the second constraint on rewards because it will be satisfied by the optimum, provided that $(\bar{s}_p \bar{s}, \bar{a})$ is realizable. The dual vector $y$ gives interesting insights about how this formulation works. Looking at the constraint in (91), we see that the variables $y$ play the role of artificial terminal values that are placed at exit states. In other words, these variables are excess values that are needed to incentivize an increased state occupancy at exit states. This is consistent with the classic interpretation of slack variables in dual programs. From an HRL perspective, on the other hand, each entry of $y$ is related to the terminal value associated with neighboring blocks. This is what causes the optimization problem to shift from pure maximization of the internal block value $V_\nu^o$, towards a compromise between the current block and other, more rewarding, blocks. Therefore, if the optimal vector $y^*$ was known in advance, the realizability problem of each abstract state and action could be solved simply by setting the rewards of the block MDP as

$$R_{\bar{s}}(s, a) := \begin{cases} R(s, a) & \text{if } s \in \lfloor \bar{s} \rfloor \\ y^*(\phi(s)) & \text{if } s \in \mathcal{X}_{\bar{s}} \\ 0 & \text{if } s = s_\perp \end{cases} \tag{92}$$

and optimizing the classic RL objective over $\mathbf{M}_{\bar{s}}$ with any (Deep) RL technique. With this interpretation, we can recognize that $y^*$ is related to what Wen et al. (2020) called "exit profiles". The main difference is that, unlike exit profiles, the values $y^*$ are homogeneous within blocks and are not assumed to be known in advance.

# D    SAMPLE COMPLEXITY OF RARL

In this section, we use $O_t$, $\bar{\mathbf{M}}_t$ and $\bar{\pi}_t$ to, respectively, denote the state of the variables $O$, $\bar{\mathbf{M}}$ and $\bar{\pi}$ in algorithm 1 at the beginning of episode $t \in \mathbb{N}_+$. Moreover we respresent the set of "known" tuples, which have been realized already, as $\mathcal{K}_t := \{(\bar{s}_p \bar{s}, \bar{a}) \mid O_t(\bar{s}_p \bar{s}, \bar{a}) \text{ is not null}\}$. The structure of this section is the following. The main sample complexity theorem comes first and the other lemmas follow below. Lemma 15 proves that ABSTRACTONER updates the rewards of $\bar{\mathbf{M}}_t$ in such a way as to obtain the intended targets $\tilde{V}$ for the block values. Lemma 14 proves that, when called with a near-optimal realization, any update made by ABSTRACTONER preserves admissibility and all the previous realizations. Lemma 13 proves that, with high probability, any option in $O_t$ is a realization for $\bar{\mathbf{M}}_t$. Finally, Theorem 8 combines these results to obtain the global sample complexity.

**Theorem 8.** *Under Assumptions 1 to 3, and any positive inputs $\varepsilon, \delta$, with probability exceeding $1 - \delta$, RARL is $\varepsilon'$-optimal with $\varepsilon' = \frac{\alpha(1-\bar{\gamma})+\beta\bar{S}}{(1-\gamma)^2(1-\bar{\gamma})} + \frac{3\varepsilon}{1-\gamma}$ on all but the following number of episodes $\frac{2\bar{S}^2\bar{A}}{\varepsilon}\left(f_r(\zeta, \eta) + \log\frac{2S^2A}{\delta}\right)$, where $f_r(\zeta, \eta)$ is the sample complexity of the realization algorithm.*

*Proof.* In this proof, $O_t$, $\bar{\mathbf{M}}_t$ and $\bar{\pi}_t$ respectively denote the state of the variables $O$, $\bar{\mathbf{M}}$ and $\bar{\pi}$ in algorithm 1 at the beginning of episode $t \in \mathbb{N}_+$.

The algorithm runs VALUEITERATION at the first episode and each time the abstraction is updated. According to Lemma 16, for any $\varepsilon_v > 0$, after $\frac{1}{1-\gamma}\log\frac{2}{(1-\gamma)^2\varepsilon_v}$ value iteration updates, the output $\bar{\pi}_t$ is always an $\varepsilon_v$-optimal policy for $\bar{\mathbf{M}}_t$ in all states, which we write $\bar{V}_t^{\bar{\pi}^*}(\bar{s}) - \bar{V}_t^{\bar{\pi}_t}(\bar{s}) \leq \varepsilon_v$ or $\bar{V}_t^{\bar{\pi}^*}(\bar{s}) - \varepsilon_v \leq \bar{V}_t^{\bar{\pi}_t}(\bar{s})$, for all $\bar{s}$. Now, for any $\varepsilon_h > 0$, to be set later, we define the abstract effective horizon as $\bar{H} := \frac{1}{1-\bar{\gamma}}\log\frac{1}{\varepsilon_h(1-\bar{\gamma})}$. Then, using Lemma 17, we obtain $\bar{V}_t^{\bar{\pi}_t}(\bar{s}) \leq \bar{V}_{t,\bar{H}}^{\bar{\pi}_t}(\bar{s}) + \varepsilon_h$, where $\bar{V}_{t,\bar{H}}^{\bar{\pi}_t}(\bar{s})$ is the expected sum of the first $H$ discounted rewards collected in $\bar{\mathbf{M}}_t$ using $\bar{\pi}_t$. So far, the chain of inequalities has led to the following.

$$\bar{V}_t^{\bar{\pi}^*}(\bar{s}) \leq \bar{V}_{t,\bar{H}}^{\bar{\pi}_t}(\bar{s}) + \varepsilon_h + \varepsilon_v \tag{93}$$

The same inequality is also true from any initial distribution. In particular, we choose $\bar{s} \sim \bar{\mu} := \mathbb{P}(\phi(s) \mid \mu)$, where $\mu$ is the initial distribution in $\mathbf{M}$. We write this as

$$\bar{V}_{t,\bar{\mu}}^{\bar{\pi}^*} \leq \bar{V}_{t,\bar{H},\bar{\mu}}^{\bar{\pi}_t} + \varepsilon_h + \varepsilon_v \tag{94}$$

Let us define the set of known tuples as $\mathcal{K}_t := \{(\bar{s}_p \bar{s}, \bar{a}) \mid O_t(\bar{s}_p \bar{s}, \bar{a}) \text{ is not null}\}$. For each $t \in \mathbb{N}_+$, we define the "escape event" $E_t$ as the event that the execution of the algorithm encounters some tuple $(\bar{s}_p \bar{s}, \bar{a})$ which is not in $\mathcal{K}_t$, in the first $\bar{H}$ blocks of episode $t$. This happens if the algorithm reaches the else branch in episode $t$ after at most $\bar{H}$ iterations. For each episode $t$, we first consider the case in which $E_t$ is does not happen. Since Assumptions 1 to 3 are satisfied, we can apply Lemma 13. This implies that $\bar{\mathbf{M}}_t$ is admissible and, for each of the first $\bar{H}$ abstract tuples encountered in that episode, there exists an associated option in $O_t$ which is a $(\alpha', \beta')$-realization. If $E_t$ does not occur, the algorithm only executes these options in the first $\bar{H}$ blocks. So, we can apply the second statement of Theorem 1 and obtain that the algorithm is executing a policy of options $\Omega_t$ that satisfies

$$\bar{V}_{t,\bar{H},\bar{\mu}}^{\bar{\pi}} - V_{\bar{H},\mu}^{\Omega_t} \leq \frac{\alpha(1-\bar{\gamma}) + \beta\bar{S}}{(1-\gamma)^2(1-\bar{\gamma})} \tag{95}$$

Theorem 1 is stated for value functions over infinite horizons, but it is applied here to value functions truncated after $\bar{H}$ blocks. This is necessary, since $\Omega_t$ is not a fully realized policy of options. Options are guaranteed to be realizations only for the first $\bar{H}$ blocks. On the other hand, the results of Theorem 1 still follow for the truncated functions, because, by definition, after $\bar{H}$ consecutive blocks

they both become equal to 0. Then, since $V_{\bar{H},\mu}^{\Omega_t} \leq V_\mu^{\Omega_t}$, we can combine the inequality above with (94) and obtain

$$\bar{V}_{t,\bar{\mu}}^{\bar{\pi}^*} \leq V_\mu^{\Omega_t} + \frac{\alpha(1-\bar{\gamma}) + \beta\bar{S}}{(1-\gamma)^2(1-\bar{\gamma})} + \varepsilon_{\mathsf{h}} + \varepsilon_{\mathsf{v}} \tag{96}$$

Now, using the fact that $\langle \bar{\mathbf{M}}_t, \phi \rangle$ is admissible for $\mathbf{M}$, we can apply Proposition 2 for the policies $\bar{\pi}^*$ and $\pi^*$, and take an expectation over $\mu$ to obtain

$$V^* - V_\mu^{\Omega_t} \leq \frac{\alpha(1-\bar{\gamma}) + \beta\bar{S}}{(1-\gamma)^2(1-\bar{\gamma})} + \varepsilon_{\mathsf{h}} + \varepsilon_{\mathsf{v}} \tag{97}$$

This proves that the algorithm is near-optimal for every episode $t$ in which $E_t$ did not occur. Note that reducing the term $\varepsilon_{\mathsf{v}}$ only increases the computational complexity, not the number of samples used. In this proof, we will pick $\varepsilon_{\mathsf{v}} := \varepsilon_{\mathsf{h}}$, for simplicity. At the end of the proof, this choice will match what is found in the main algorithm.

For the episodes in which the escape event $E_t$ does happen, instead, we do not make any guarantee on $V^{\pi_t}$, which might be zero. Here $\pi_t$ represents the low-level policy used by the algorithm in episode $t$. Let $\neg E_t$ be the negation of the escape event. Accounting for both cases, without conditioning on $E_t$, the value of the algorithm in episode $t$ is

$$V^{\pi_t} = \mathbb{E}\left[\sum_{i=1}^\infty r_i \mid \mathbf{M}, \pi_t\right] \tag{98}$$

$$\geq \mathbb{P}(\neg E_t)\, \mathbb{E}\left[\sum_{i=1}^\infty r_i \mid \mathbf{M}, \pi_t, \neg E_t\right] \tag{99}$$

using (97),

$$\geq \mathbb{P}(\neg E_t)\left(V^* - \frac{\alpha(1-\bar{\gamma}) + \beta\bar{S}}{(1-\gamma)^2(1-\bar{\gamma})} - 2\varepsilon_{\mathsf{h}}\right) \tag{100}$$

$$= V^* - \mathbb{P}(E_t)V^* - (1 - \mathbb{P}(E_t))\left(\frac{\alpha(1-\bar{\gamma}) + \beta\bar{S}}{(1-\gamma)^2(1-\bar{\gamma})} + 2\varepsilon_{\mathsf{h}}\right) \tag{101}$$

Then,

$$V^* - V^{\pi_t} \leq \mathbb{P}(E_t)V^* + (1 - \mathbb{P}(E_t))\left(\frac{\alpha(1-\bar{\gamma}) + \beta\bar{S}}{(1-\gamma)^2(1-\bar{\gamma})} + 2\varepsilon_{\mathsf{h}}\right) \tag{102}$$

$$\leq \frac{\mathbb{P}(E_t)}{1-\gamma} + \frac{\alpha(1-\bar{\gamma}) + \beta\bar{S}}{(1-\gamma)^2(1-\bar{\gamma})} + 2\varepsilon_{\mathsf{h}} \tag{103}$$

Let $\varepsilon_{\mathsf{h}}' := \varepsilon_{\mathsf{h}}(1-\gamma)$,

$$V^* - V^{\pi_t} \leq \frac{\alpha(1-\bar{\gamma}) + \beta\bar{S}}{(1-\gamma)^2(1-\bar{\gamma})} + \frac{\mathbb{P}(E_t)}{1-\gamma} + \frac{2\varepsilon_{\mathsf{h}}'}{1-\gamma} \tag{104}$$

If $\mathbb{P}(E_t) \leq \varepsilon_{\mathsf{h}}'$, then the algorithm is near-optimal in episode $t$, with

$$V^* - V^{\pi_t} \leq \frac{\alpha(1-\bar{\gamma}) + \beta\bar{S}}{(1-\gamma)^2(1-\bar{\gamma})} + \frac{3\varepsilon_{\mathsf{h}}'}{1-\gamma} \tag{105}$$

Now consider any episode $t$ in which $\mathbb{P}(E_t) > \varepsilon_{\mathsf{h}}'$. We follow a similar reasoning to the proof of theorem 10 in Strehl et al. (2009). The event $E_t$ can be modeled with a Bernoulli random variable $X_t$ with $\mathbb{E}[X_t] > \varepsilon_{\mathsf{h}}'$. We observe that the probability of this event stays constant even in the following episodes until some new option is added, because both the abstract policy and the set of options used remain constant for the known tuples. In other words, $X_t, X_{t+1}, \ldots, X_{t'}$ is a sequence of independent and identically distributed Bernoulli RVs, where $t'$ is the first episode in which $\mathcal{K}_{t'-1} \neq \mathcal{K}_{t'}$ (a new tuple becomes known). Thanks to our choices, the sum $\sum_{i=t,\ldots,t'} X_i$ represents the number of times that a new trajectory is collected and it contributes to the realization of $(\bar{s}_p\bar{s},\bar{a})$ before a new update occurs. By Assumption 1 and the guarantees of PAC-Safe algorithms, the realizer only requires a number of trajectories that is some polynomial in $(|\mathcal{S}|, |\mathcal{A}|, 1/\zeta, \log(2\bar{S}^2\bar{A}/\delta), 1/\eta, 1/(1-\gamma))$. Let

$f_r(\zeta, \eta)$ be such polynomial. We estimate the number of exits required before some tuple becomes known. In other words, we guarantee that $\sum_{i=t,\ldots,t'} X_i \geq f_r(\zeta, \eta)$ with high probability, through an application of Lemma 18. This implies that, for any $\delta_e > 0$, if

$$t' - t \geq \frac{2}{\varepsilon'_h} \left( f_r(\zeta, \eta) + \log \frac{1}{\delta_e} \right) =: \Delta \tag{106}$$

then, $\sum_{i=t,\ldots,t'} X_i \geq f_r(\zeta, \eta)$ with probability at least $1 - \delta_e$. Therefore, with probability $1 - \delta_e$, a new option will become known after at most $\Delta$ episodes from $t$. Essentially, $\Delta$ is the maximum sample complexity for learning some new option with high probability.

Concluding, since the maximum number of options to realize is at most $S^2A$, we multiply the bound above for each tuple, and apply the union bound with $\delta_e := \delta/(2S^2A)$ to verify that the event $\sum_i X_i \geq f_r(\zeta, \eta)$ holds with probability $1 - \delta/2$ for all tuples. This gives a total sample complexity of

$$\frac{2\bar{S}^2\bar{A}}{\varepsilon'_h} \left( f_r(\zeta, \eta) + \log \frac{2S^2A}{\delta} \right) \tag{107}$$

With a further application of the union bound, we guarantee that this sample complexity and the statement of Lemma 13 jointly hold with probability $1 - \delta$. To select the appropriate accuracy, we choose $\varepsilon_h := \varepsilon/(1-\gamma)$, which gives $\varepsilon'_h = \varepsilon$. $\qquad \square$

**Lemma 13.** *Under Assumptions 1 to 3, and any positive inputs $\varepsilon, \delta$, it holds, with probability $1 - \delta$, that for every $t \in \mathbb{N}_+$, $\bar{\mathbf{M}}_t$ is admissible and, for every $(\bar{s}_p\bar{s}, \bar{a}) \in \mathcal{K}_t$, $O_t(\bar{s}_p\bar{s}, \bar{a})$ is an $(\alpha', \beta')$-realization of $(\bar{s}_p\bar{s}, \bar{a})$ in $\bar{\mathbf{M}}_t$, where $\alpha' := \alpha + \zeta(1-\gamma)$ and $\beta' := \beta + \eta(1-\gamma)$.*

*Proof.* As assumed by Assumption 1, each instance of REALIZER is a PAC-Safe RL algorithm with maximum failure probability of $\delta/(2\bar{S}^2\bar{A})$. Since there are less than $\bar{S}^2\bar{A}$ instances of REALIZER, and each of them only returns the option at most once, by the union bound, the probability that any of the instances fail is at most $\delta/2$. Taking into account this failure probability, we condition the rest of the proof on the event that none of the instances in $\mathfrak{A}$ fail.

The proof is by induction. Since $\bar{\mathbf{M}}_1$ is the input of the algorithm, it is admissible, according to Assumption 2. Also, since $\mathcal{K}_1$ is emtpy, the second half of the statement trivially holds. The inductive step will occupy most of the remaining proof. For any episode $t \geq 1$, assume that $\bar{\mathbf{M}}_t$ is admissible and, for every $(\bar{s}_p\bar{s}, \bar{a}) \in \mathcal{K}_t$, $O_t(\bar{s}_p\bar{s}, \bar{a})$ is an $(\alpha', \beta')$-realization of $(\bar{s}_p\bar{s}, \bar{a})$ from $\nu_{t,\bar{s}_p\bar{s}}$ in $\bar{\mathbf{M}}_t$. Now, if the algorithm does not enter the block in line 14 during episode $t$, which can only happen at most once in any episode, then $\mathcal{K}_{t+1} = \mathcal{K}_t$, $O_{t+1} = O_t$ and $\bar{\mathbf{M}}_{t+1} = \bar{\mathbf{M}}_t$. Therefore, the inductive step is verified.

We now consider the case in which the algorithm does enter the block in line 14. In this case, a single tuple is added, $\mathcal{K}_{t+1} = \mathcal{K}_t \cup \{(\bar{s}_p\bar{s}, \bar{a})\}$, with its associated option $\hat{o} := O_{t+1}(\bar{s}_p\bar{s}, \bar{a})$. Therefore, we proceed to prove that $\hat{o}$ is an $(\alpha', \beta')$-realization of $(\bar{s}_p\bar{s}, \bar{a})$ from $\nu_{t+1,\bar{s}_p\bar{s}}$ in $\bar{\mathbf{M}}_t$. We will link this result to $\bar{\mathbf{M}}_{t+1}$ later in the proof. Consider $\mathfrak{A}(\bar{s}_p\bar{s}, \bar{a})$, the instance of REALIZER associated with the newly added tuple. Copying from (8), and accounting for $\Pi_{c,\eta}$, we observe that the realizer algorithm is solving the following problem:

$$\arg\max_{o \in \Omega_{\bar{s}_p\bar{s}}} V_\nu^o \qquad s.t. \quad V_{\nu,\bar{s}'}^o \geq \frac{\tilde{h}_{\bar{s}_p\bar{s}\bar{a}}(\bar{s}') - \beta}{1-\gamma} - \eta \quad \forall \bar{s}' \neq \bar{s} \tag{108}$$

Here, $\nu$ should be intended as $\nu_{t,\bar{s}_p\bar{s}} = \mathbb{P}(s \mid s_p \in \lfloor\bar{s}_p\rfloor, s \in \lfloor\bar{s}\rfloor, O_t)$. In other words, this is the entry distribution caused by the options available in episode $t$. Let $o^*$ be the optimal solution of the original optimization problem (8), and $o^{\eta*}$ be the optimal solution of the relaxed probelm in (108). Note that these options always exist because, by Assumption 2, the feasible sets $\Pi_c$ and $\Pi_{c,\eta}$ cannot be empty, because $\bar{\mathbf{M}}^*$ only differs from $\bar{\mathbf{M}}_t$ with respect to the reward function, and the constraint set only depends on transition probabilities. Then, by the guarantees of PAC-Safe algorithms, the instance of REALIZER returns an option $\hat{o}$ which satisfies all the constraints above and, for the objective, it holds $V_\nu^{o^{\eta*}} - V_\nu^{\hat{o}} \leq \zeta$. Now, since $\Pi_c \subseteq \Pi_{c,\eta}$, it holds $V_\nu^{o^*} \leq V_\nu^{o^{\eta*}}$, which implies $V_\nu^{o^*} - V_\nu^{\hat{o}} \leq \zeta$. Next, we consider the abstraction $\bar{\mathbf{M}}^*$, which is referenced by Assumption 2. This unknown 2-MDP is admissible and $(\alpha, \beta)$-realizable. Since $o^*$ is an optimal solution of the original realization problem, by Assumption 2, we have $\hat{V}^*_{\bar{s}_p\bar{s}\bar{a}} - V_\nu^{o^*} \leq \alpha/(1-\gamma)$, where the left-most term is $\hat{V}_{\bar{s}_p\bar{s}\bar{a}}$, computed

in $\bar{\mathbf{M}}^*$. This means that we can lower bound $V_\nu^{o^*}$ by $\tilde{V}_{\bar{s}_p\bar{s}\bar{a}}^* - \alpha/(1-\gamma)$ in the inequalities above, and obtain $(1-\gamma)(\tilde{V}_{\bar{s}_p\bar{s}\bar{a}}^* - V_\nu^{\hat{o}}) \leq \alpha + \zeta(1-\gamma)$. Regarding the constraints, instead, we can follow the same chain of equalities there were used above (8), and obtain $\tilde{h}_{\bar{s}_p\bar{s}\bar{a}}(\bar{s}') - h_\nu^{\hat{o}}(\bar{s}') \leq \beta + \eta(1-\gamma)$. Together, the two inequalities above prove that the output of $\mathfrak{A}(\bar{s}_p\bar{s}, \bar{a})$ is a $(\alpha', \beta')$-realization of $(\bar{s}_p\bar{s}, \bar{a})$ from $\nu$ in $\bar{\mathbf{M}}^*$, for $\alpha' := \alpha + \zeta(1-\gamma)$ and $\beta' := \beta + \eta(1-\gamma)$. Now, according to Assumption 3, the stame statement will be true not only from $\nu_{t,\bar{s}_p\bar{s}}$ but from any future initial distribution, that may be caused by addition of new options in $O_t$. So, we will simply say that $\hat{o}$ is an $(\alpha', \beta')$-realization in $\bar{\mathbf{M}}^*$.

We now distinguish two cases. If the algorithm does not enter the block in line 16 in episode $t$, then $\bar{\mathbf{M}}_{t+1} = \bar{\mathbf{M}}_t$. This means that $\bar{\mathbf{M}}_{t+1}$ is admissible and $\hat{o}$ is an $(\alpha', \beta')$-realization in $\bar{\mathbf{M}}_{t+1}$. Indeed, the failed if-statement ensures that $(1-\gamma)(\tilde{V}_{t,\bar{s}_p\bar{s}\bar{a}} - V_\nu^{\hat{o}}) \leq \alpha'$. Since $\bar{\mathbf{M}}^*$ and $\bar{\mathbf{M}}_t$ have the same transition function, this proves that $O_t(\bar{s}_p\bar{s}, \bar{a})$ is an $(\alpha', \beta')$-realization in $\bar{\mathbf{M}}_t$.

The last case to consider is when the algorithm enters line 16 in episode $t$. In this case, the abstraction gets updated with ABSTRACTONER. For characterizing its output, we verify if the preconditions of Lemma 14 are satisfied. First, Condition 1 is satisfied by $\bar{\mathbf{M}}_t$ because $\bar{\mathbf{M}}_1$ satisfies it due to Assumption 2 and it only differs from $\bar{\mathbf{M}}_t$ by its reward function. Second, as option we consider $\hat{o}$, which we already know from above that is an $(\alpha', \beta')$-realization in $\bar{\mathbf{M}}^*$. Lastly, we know that $\tilde{V}_{t,\bar{s}_p\bar{s}\bar{a}} - V_\nu^{\hat{o}} > \alpha'/(1-\gamma)$ because the if-statement succeeded. So, we can apply Lemma 14. Any tuple that does not involve $\bar{s}$ is not affected by the updated rewards. Therefore, we only verify the tuples $(\bar{s}'_p\bar{s}\bar{a})$. Using statement 4 of Lemma 14, we know that if $(\bar{s}'_p\bar{s}\bar{a}) \in \mathcal{K}_t$ and $O_t(\bar{s}'_p\bar{s}\bar{a})$ is an $(\alpha', \beta')$-realization for $\bar{\mathbf{M}}_t$, then, the same will be true for $\bar{\mathbf{M}}_t$, since $\mathcal{K}_t \subseteq \mathcal{K}_{t+1}$ and $O_t(\bar{s}'_p\bar{s}\bar{a}) = O_{t+1}(\bar{s}'_p\bar{s}\bar{a})$. Also, by statement 3 of Lemma 14, the same tuple is also admissible in $\bar{\mathbf{M}}_{t+1}$ thanks to the induction hypothesis. Lastly, the new tuple $(\bar{s}_p\bar{s}, \bar{a})$ is admissible in $\bar{\mathbf{M}}_{t+1}$ and realized by $\hat{o} = O_{t+1}(\bar{s}'_p\bar{s}\bar{a})$, because of statement 2. Although we are not referring to initial distributions, we also used Assumption 3, implicitly. $\qquad\square$

**Condition 1.** Given positive $\beta$ and $\alpha$, there exists some admissible $(\alpha, \beta)$-realizable abstraction $\langle \bar{\mathbf{M}}^*, \phi \rangle$ in which $\bar{\mathbf{M}}^*$ only differs from $\bar{\mathbf{M}}$ by its reward function.

**Lemma 14.** *Consider an MDP* $\mathbf{M}$, *an abstraction* $\langle \bar{\mathbf{M}}, \phi \rangle$ *satisfying Condition 1, any tuple* $(\bar{s}_p\bar{s}, \bar{a})$ *and some option* $o \in \Omega_{\bar{s}_p\bar{s}}$ *that is an* $(\alpha, \beta)$-*realization of* $(\bar{s}_p\bar{s}, \bar{a})$ *from* $\bar{\mathbf{M}}^*$ *and some* $\nu \in \Delta(\mathcal{E}_{\bar{s}_p\bar{s}})$. *If* $\tilde{V}_{\bar{s}_p s\bar{a}} \geq V_\nu^o + \alpha/(1-\gamma)$, *then, in relation to* $\bar{\mathbf{M}}' := \text{ABSTRACTONER}(\bar{\mathbf{M}}, (\bar{s}_p\bar{s}, \bar{a}), V_\nu^o + \alpha/(1-\gamma))$ *it holds*

1. *$\bar{\mathbf{M}}'$ is a valid 2-MDP;*

2. *$\tilde{V}'_{\bar{s}_p\bar{s}\bar{a}} = V_\nu^o + \alpha/(1-\gamma)$;*

3. *For any $\bar{s}'_p \neq \bar{s}$, if $(\bar{s}'_p\bar{s}\bar{a})$ is admissible in $\bar{\mathbf{M}}$ from some $\nu' \in \Delta(\mathcal{E}_{\bar{s}'_p\bar{s}})$, the same is true in $\bar{\mathbf{M}}'$.*

4. *For any $\bar{s}'_p \neq \bar{s}$, if $o'$ is an $(\alpha, \beta)$-realization of $(\bar{s}'_p\bar{s}\bar{a})$ in $\bar{\mathbf{M}}$ from some $\nu' \in \Delta(\mathcal{E}_{\bar{s}'_p\bar{s}})$, then, the same is true in $\bar{\mathbf{M}}'$.*

*Proof.* The first two points of the statements are direct consequences of Lemma 15. In fact, the assumptions taken by this statement subsume those of Lemma 15.

The third statement says that admissibility is preserved in $\bar{\mathbf{M}}'$. For transitions, this is immediately true, because they are not modified by the ABSTRACTONER. Now we focus on rewards. For the tuple $(\bar{s}_p\bar{s}, \bar{a})$, we observe that there cannot be any other option $o' \in \Omega_{\bar{s}_p\bar{s}}$ and distribution $\nu' \in \Delta(\mathcal{E}_{\bar{s}_p\bar{s}})$, for which $\tilde{V}'_{\bar{s}_p\bar{s}\bar{a}} < V_{\nu'}^{o'}$. In fact, using statement 2 and the fact that $o$ is a $(\alpha, \beta)$-realization in $\bar{\mathbf{M}}^*$, we would have

$$V_{\nu'}^{o'} > \tilde{V}'_{\bar{s}_p\bar{s}\bar{a}} = V_\nu^o + \frac{\alpha}{1-\gamma} \geq \tilde{V}_{\bar{s}_p\bar{s}\bar{a}}^* \tag{109}$$

where the two extremes of the inequality contradict the fact that $(\bar{s}_p\bar{s}, \bar{a})$ is admissible in $\bar{\mathbf{M}}^*$. Now, it only remains to verify the reward of each remaining tuple $(\bar{s}'_p\bar{s}\bar{a})$. We verify this by contradiction.

Consider an abstract state $\bar{s}'_p \notin \{\bar{s}, \bar{s}_p\}$, a distribution $\nu' \in \Delta(\mathcal{E}_{\bar{s}'_p \bar{s}})$, and an option $o' \in \Omega_{\bar{s}'_p \bar{s}}$, such that $\tilde{V}'_{\bar{s}'_p \bar{s} \bar{a}} < V^{o'}_{\nu'}$. Then,

$$V^{o'}_{\nu'} > \bar{R}'(\bar{s}'_p \bar{s}, \bar{a}) + \frac{\bar{\gamma} \bar{R}'(\bar{s}\bar{s}, \bar{a}) \bar{T}(\bar{s} \mid \bar{s}'_p \bar{s}, \bar{a})}{1 - \bar{\gamma} \bar{T}(\bar{s} \mid \bar{s}\bar{s}, \bar{a})} \tag{110}$$

$$= \min\{1, \bar{R}(\bar{s}'_p \bar{s}, \bar{a}) + \tilde{V}_{\bar{s}'_p \bar{s} \bar{a}} - V^+_{\bar{s}'_p \bar{s} \bar{a}}\} + \frac{\bar{\gamma} \bar{R}'(\bar{s}\bar{s}, \bar{a}) \bar{T}(\bar{s} \mid \bar{s}'_p \bar{s}, \bar{a})}{1 - \bar{\gamma} \bar{T}(\bar{s} \mid \bar{s}\bar{s}, \bar{a})} \tag{111}$$

Where $\min\{1, \bar{R}(\bar{s}'_p \bar{s}, \bar{a}) + \tilde{V}_{\bar{s}'_p \bar{s} \bar{a}} - V^+_{\bar{s}'_p \bar{s} \bar{a}}\}$ is the assignment of line 25. Now, we continue considering the case $\bar{R}(\bar{s}'_p \bar{s}, \bar{a}) + \tilde{V}_{\bar{s}'_p \bar{s} \bar{a}} - V^+_{\bar{s}'_p \bar{s} \bar{a}} \leq 1$:

$$V^{o'}_{\nu'} > \bar{R}(\bar{s}'_p \bar{s}, \bar{a}) + \tilde{V}_{\bar{s}'_p \bar{s} \bar{a}} - V^+_{\bar{s}'_p \bar{s} \bar{a}} + \frac{\bar{\gamma} \bar{R}'(\bar{s}\bar{s}, \bar{a}) \bar{T}(\bar{s} \mid \bar{s}'_p \bar{s}, \bar{a})}{1 - \bar{\gamma} \bar{T}(\bar{s} \mid \bar{s}\bar{s}, \bar{a})} \tag{112}$$

Expanding $V^+_{\bar{s}'_p \bar{s} \bar{a}}$, the whole right-hand term simplifies to $\tilde{V}_{\bar{s}'_p \bar{s} \bar{a}}$. However, $V^{o'}_{\nu'} > \tilde{V}_{\bar{s}'_p \bar{s} \bar{a}}$ contradicts the fact that $(\bar{s}'_p \bar{s} \bar{a})$ is admissible in $\mathbf{M}$. We now consider the case $\bar{R}(\bar{s}'_p \bar{s}, \bar{a}) + \tilde{V}_{\bar{s}'_p \bar{s} \bar{a}} - V^+_{\bar{s}'_p \bar{s} \bar{a}} > 1$. Then,

$$V^{o'}_{\nu'} > \tilde{V}'_{\bar{s}'_p \bar{s} \bar{a}} = 1 + \frac{\bar{\gamma} \bar{R}'(\bar{s}\bar{s}, \bar{a}) \bar{T}(\bar{s} \mid \bar{s}'_p \bar{s}, \bar{a})}{1 - \bar{\gamma} \bar{T}(\bar{s} \mid \bar{s}\bar{s}, \bar{a})} \tag{113}$$

Now, by assumption, we know that $\bar{\mathbf{M}}$ satisfies Condition 1. This means that there exists a reward function $\bar{R}^*$ such that $\bar{\mathbf{M}}^* = \langle \bar{\mathcal{S}}, \bar{\mathcal{A}}, \bar{T}, \bar{R}^*, \bar{\gamma} \rangle$ is admissible. Continuing from above,

$$V^{o'}_{\nu'} > \bar{R}^*(\bar{s}'_p \bar{s}, \bar{a}) + \frac{\bar{\gamma} \bar{R}'(\bar{s}\bar{s}, \bar{a}) \bar{T}(\bar{s} \mid \bar{s}'_p \bar{s}, \bar{a})}{1 - \bar{\gamma} \bar{T}(\bar{s} \mid \bar{s}\bar{s}, \bar{a})} \tag{114}$$

Now, we argue that $\bar{R}'(\bar{s}\bar{s}, \bar{a}) \geq \bar{R}^*(\bar{s}\bar{s}, \bar{a})$. In fact, under the case we were considering, $\tilde{V}_{\bar{s}'_p \bar{s} \bar{a}} > V^+_{\bar{s}'_p \bar{s} \bar{a}}$, which means that the reward function for $(\bar{s}\bar{s}, \bar{a})$ has been modified by the algorithm. In turn this only happens when $\bar{R}'(\bar{s}_p \bar{s}, \bar{a}) = 0$. However, since we know by statement 2 that $\tilde{V}'_{\bar{s}_p \bar{s} \bar{a}} = V^o_\nu + \alpha/(1-\gamma)$, if it was the case that $\bar{R}^*(\bar{s}\bar{s}, \bar{a}) > \bar{R}'(\bar{s}\bar{s}, \bar{a})$, then, the rewards of $(\bar{s}_p \bar{s}, \bar{a})$ would not be $\alpha$-realizable in $\bar{\mathbf{M}}^*$ with $o$ as we assumed. Finally, since we obtained that $\bar{R}'(\bar{s}\bar{s}, \bar{a}) \geq \bar{R}^*(\bar{s}\bar{s}, \bar{a})$, we conclude the chain of inequalities:

$$V^{o'}_{\nu'} > \bar{R}^*(\bar{s}'_p \bar{s}, \bar{a}) + \frac{\bar{\gamma} \bar{R}^*(\bar{s}\bar{s}, \bar{a}) \bar{T}(\bar{s} \mid \bar{s}'_p \bar{s}, \bar{a})}{1 - \bar{\gamma} \bar{T}(\bar{s} \mid \bar{s}\bar{s}, \bar{a})} = \tilde{V}^*_{\bar{s}'_p \bar{s} \bar{a}} \tag{115}$$

Thus, leading to contradiction with the fact that $\bar{\mathbf{M}}^*$ is admissible.

The fourth and final result of the lemma is easy to prove because it follows from the admissibility of point 3, and the fact that $V^{o'}_{\nu'} + \alpha/(1-\gamma) \geq \tilde{V}_{\bar{s}'_p \bar{s} \bar{a}} \geq \tilde{V}'_{\bar{s}'_p \bar{s} \bar{a}}$. Also, realizability in transitions is not affected by the function. $\qquad\square$

**Lemma 15.** *Consider any MDP $\mathbf{M}$, any abstraction $\langle \bar{\mathbf{M}}, \phi \rangle$, any abstract tuple $(\bar{s}_p \bar{s}, \bar{a})$ and any $V \in [0, 1/(1-\gamma)]$. If $\tilde{V}_{s_p s a} \geq V$, then $\bar{\mathbf{M}}' := \textsc{AbstractOneR}(\bar{\mathbf{M}}, (\bar{s}_p \bar{s}, \bar{a}), V)$ is a valid 2-MDP and $\tilde{V}'_{\bar{s}_p \bar{s} \bar{a}} = V$.*

*Proof.* First we check that $\bar{\mathbf{M}}'$ is a valid MDP. To verify this, we already know that $\bar{R}'(\bar{s}_p \bar{s}, \bar{a}) \geq 0$. However, $\bar{R}'(\bar{s}_p \bar{s}, \bar{a}) \leq 1$ is also true, since we assumed that $\tilde{V}_{s_p s a} \geq V$. For $\bar{R}'(\bar{s}\bar{s}, \bar{a})$, we should only verify the case $\bar{R}'(\bar{s}_p \bar{s}, \bar{a}) = 0$. In turn, this only happens if $\bar{R}(\bar{s}_p \bar{s}, \bar{a}) \leq \tilde{V}_{\bar{s}\bar{s} \bar{a}} - V$, that is $V \leq \tilde{V}_{\bar{s}_p \bar{s} \bar{a}} - \bar{R}(\bar{s}_p \bar{s}, \bar{a})$. Then,

$$\bar{R}'(\bar{s}\bar{s}, \bar{a}) = V \left( \frac{\bar{\gamma} \bar{T}(\bar{s} \mid \bar{s}_p \bar{s}, \bar{a})}{1 - \bar{\gamma} \bar{T}(\bar{s} \mid \bar{s}\bar{s}, \bar{a})} \right)^{-1} \tag{116}$$

$$\leq (\tilde{V}_{\bar{s}_p \bar{s} \bar{a}} - \bar{R}(\bar{s}_p \bar{s}, \bar{a})) \left( \frac{\bar{\gamma} \bar{T}(\bar{s} \mid \bar{s}_p \bar{s}, \bar{a})}{1 - \bar{\gamma} \bar{T}(\bar{s} \mid \bar{s}\bar{s}, \bar{a})} \right)^{-1} \tag{117}$$

$$= \bar{R}(\bar{s}\bar{s}, \bar{a}) \tag{118}$$

which proves $\bar{R}'(\bar{s}\bar{s}, \bar{a}) \in [0, 1]$. Finally, for any $\bar{s}'_p \notin \{\bar{s}, \bar{s}_p\}$, we write line 25 as:

$$\bar{R}'(\bar{s}'_p\bar{s}, \bar{a}) = \min\{1, \bar{R}(\bar{s}'_p\bar{s}, \bar{a}) + \tilde{V}_{\bar{s}'_p\bar{s}\bar{a}} - V^+_{\bar{s}'_p\bar{s}\bar{a}}\} \tag{119}$$

where $V^+_{\bar{s}'_p\bar{s}\bar{a}}$ is $\tilde{V}_{\bar{s}'_p\bar{s}\bar{a}}$, computed in the MDP obtained after the assignments above line 25. We have already verified that $\bar{R}'(\bar{s}_p\bar{s}, \bar{a}) \leq \bar{R}(\bar{s}_p\bar{s}, \bar{a})$ and $\bar{R}'(\bar{s}\bar{s}, \bar{a}) \leq \bar{R}(\bar{s}\bar{s}, \bar{a})$. This implies that $\tilde{V}_{\bar{s}'_p\bar{s}\bar{a}} - V^+_{\bar{s}'_p\bar{s}\bar{a}}$ is positive and $\bar{R}'(\bar{s}'_p\bar{s}, \bar{a}) \in [0, 1]$.

Now we verify the second point of the statement by substituting the definition of $\tilde{V}'_{\bar{s}_p\bar{s}\bar{a}}$ for $\bar{\mathbf{M}}'$. We consider two cases. If $\bar{R}(\bar{s}_p\bar{s}, \bar{a}) > \tilde{V}_{\bar{s}_p\bar{s}\bar{a}} - V$,

$$\tilde{V}'_{\bar{s}_p\bar{s}\bar{a}} = \bar{R}'(\bar{s}_p\bar{s}, \bar{a}) + \frac{\bar{\gamma}\bar{R}'(\bar{s}\bar{s}, \bar{a})\bar{T}(\bar{s} \mid \bar{s}_p\bar{s}, \bar{a})}{1 - \bar{\gamma}\bar{T}(\bar{s} \mid \bar{s}\bar{s}, \bar{a})} \tag{120}$$

$$= \bar{R}(\bar{s}_p\bar{s}, \bar{a}) + V - \tilde{V}_{\bar{s}_p\bar{s}\bar{a}} + \frac{\bar{\gamma}\bar{R}(\bar{s}\bar{s}, \bar{a})\bar{T}(\bar{s} \mid \bar{s}_p\bar{s}, \bar{a})}{1 - \bar{\gamma}\bar{T}(\bar{s} \mid \bar{s}\bar{s}, \bar{a})} \tag{121}$$

$$= V \tag{122}$$

On the other hand, if $\bar{R}(\bar{s}_p\bar{s}, \bar{a}) \leq \tilde{V}_{\bar{s}_p\bar{s}\bar{a}} - V$,

$$\tilde{V}'_{\bar{s}_p\bar{s}\bar{a}} = 0 + \frac{\bar{\gamma}\bar{R}'(\bar{s}\bar{s}, \bar{a})\bar{T}(\bar{s} \mid \bar{s}_p\bar{s}, \bar{a})}{1 - \bar{\gamma}\bar{T}(\bar{s} \mid \bar{s}\bar{s}, \bar{a})} = V \tag{123}$$

$\square$

OTHER LEMMAS

As shown in Agarwal et al. (2021), the $\gamma$-contraction property, together with Singh & Yee (1994, Corollary 2), gives the following.

**Lemma 16.** *Let $Q^{(k)}$ be the Q-function obtained after $k$ VALUEITERATION updates, and let $\pi^{(k)}$ be the greedy policy for $Q^{(k)}$. If $k \geq \frac{\log \frac{2}{(1-\gamma)^2\varepsilon}}{1-\gamma}$, then $V^*(s) - V^{\pi^{(k)}}(s) \leq \varepsilon$ for each $s \in \mathcal{S}$.*

The following statement was expressed for MDPs in Kearns & Singh (2002). However, its proof only relies on geometric discounting. So, it can also be applied to any decision process and $k$-MDP.

**Lemma 17** (Kearns & Singh (2002)). *In any decision process $\mathbf{M}$ and policy $\pi$, if $H = \frac{1}{1-\gamma}\log\frac{1}{\varepsilon(1-\gamma)}$, then, in any state $s$, $V^\pi(s) \leq V^\pi_H(s) + \varepsilon$, where $V^\pi_H$ is the expected sum of the first $H$ discounted rewards.*

Finally, we adopt the following concentration inequality from Li (2009, Corollary 2).

**Lemma 18** (Li (2009)). *Let $X_1, \dots, X_m$ be a sequence of $m$ independent Bernoulli RVs, with $\mathbb{P}(X_i) \geq a$, for all $i$, for some constant $a > 0$. Then, for any $k \in \mathbb{N}$ and $\delta > 0$, with probability at least $1 - \delta$, $\sum_{i=1,\dots,m} X_i \geq k$, provided that*

$$m \geq \frac{2}{a}\left(k + \log\frac{1}{\delta}\right) \tag{124}$$

