# OpenReview forum: "Realizable Abstractions: Near-Optimal Hierarchical Reinforcement Learning"
_ICLR.cc/2025/Conference — Submitted to ICLR 2025_

### Official Review · Reviewer_JLGr · 2024-10-26

**Soundness:** 3
**Presentation:** 2
**Contribution:** 3
**Rating:** 8
**Confidence:** 2

**Summary:**

This paper studies hierarchical MDP. To characterize how close the abstraction states to the ground truth, they proposed the idea of realizable abstractions. They further show that under this condition, the values of the abstract MDPs and ground MDP are close to each other.

Based on these properties, they proposed a new algorithm for hierarchical MDPs, and they demonstrate sample efficiency for the algorithm.

**Strengths:**

1. The setting of this paper is clear and the paper is well written.

2. The algorithm proposed in this paper is sample and computational efficient.

3. The idea of realizable abstraction is natural and useful for characterizing the sample complexity of the algorithm.

**Weaknesses:**

1. This paper proposes an algorithm for hierarchical RL. It will be better if there is some numerical experiment which demonstrate that the hierarchical RL algorithm performs better than the normal RL algorithm in some specific domain.

**Questions:**

1. The author proposes the condition of realizable abstraction and the sample complexity of the algorithm depends on this condition. Is there any way to identify an abstraction which satisfies this condition?

---

> ### Author Response · Authors · 2024-11-22
>
> We thank the reviewer for his thoughtful comments. Please find our answer to the concerns below.
>
> We can think of the two main sections of the paper, 3 and 4, as having independent contributions: showing the theoretical properties of our realizable abstractions, and developing a hierarchical algorithm with formal correctness guarantees. Both contributions have been demonstrated theoretically. However, specifically to the second contribution (the one of Section 4), we agree with the reviewer that an experimental evaluation would help to show the practical performances of the specific algorithm we propose.
>
> The definition of realizable abstractions is meant to inform researchers in HRL about which quantities should be preserved in abstract models and how these are tightly linked to the respective discount factors and termination probabilities in the ground MDP. We believe this will help to design more accurate abstractions.
>
> It is possible to verify whether some $\langle \bar{\mathbf{M}}, \phi \rangle$ is realizable for a ground MDP $\mathbf{M}$. For example, during its execution, our algorithm verifies whether the input model is realizable or not, with respect to rewards, and it corrects the abstract reward function accordingly. On the other hand, verifying admissibility may be more complex, as it requires that the abstract model is optimistic *for all* ground options.
>
> The exact characterization of realizable abstraction is given in Definition 2. However, we can give some examples of abstract models that satisfy it. Some of these are more interesting than others for practical purposes, but all help to understand how these abstractions work.
> 1.  Any MDP $\bar{\mathbf{M}}$ that has the same transition function of $\mathbf{M}$, and its rewards are equal or higher, it is an admissible and realizable abstraction.
> 2. Any MDP with bisimilar states can be simplified into an abstract $\bar{\mathbf{M}}$ that has fewer states and that is admissible and perfectly realizable (also see the results for bisimilarity that we added in Appendix B).
> 3. If an MDP is goal-directed, meaning, there is a set of rewarding states and all other states return 0, then, for a suitable state partition, it is possible to construct an admissible and realizable abstraction as follows: the reward function is preserved by the state abstraction function $\phi$ (0 for most states, and 1 for goal states), and the transition function overestimates the discounted probability of leaving each block (thus, simplifying the navigation in the environment). For a numerical example, please see the answer to reviewer gPDx, assumption 2.

---

> ### Comment · Reviewer_JLGr · 2024-11-24
>
> Thanks to the authors for the response. I have increased the score accordingly.

---

### Official Review · Reviewer_UaD5 · 2024-10-31

**Soundness:** 2
**Presentation:** 2
**Contribution:** 1
**Rating:** 3
**Confidence:** 3

**Summary:**

This paper explores HRL and introduces a framework called Realizable Abstractions, which aims to improve the efficiency of solving large MDPs by breaking them into modular components. It defines a formal relationship between high-level (abstract) and low-level (ground) values and specifies conditions required to reduce the effective planning horizon within these abstractions. Furthermore, the paper presents a sample-efficient algorithm that operates based on the assumption of an admissible and realizable abstraction.

**Strengths:**

Realizable Abstractions offer a fresh theoretical foundation for HRL, opening up a promising way for potentially reducing sample complexity in reinforcement learning. I find it particularly intriguing that sparse rewards play a crucial role in ensuring admissibility (Proposition 6).

**Weaknesses:**

While the new concept of Realizable Abstractions is interesting, the paper lacks some key definitions, making it challenging to follow. As a result, it’s difficult to fully grasp the significance of the paper’s main contributions.

The followings are main weaknesses of this paper:

- The proposed algorithm requires a **known** abstraction and assumes admissibility, which feels like a rather strong assumption. However, there is not enough rigorous discussion regarding these assumptions and their implications.
It is also unclear how stringent these assumptions are compared to other assumptions, such as known state abstraction (Abel, 2020) or known equivalence mapping (Wen et al., 2020).

- Exploration, which is crucial yet challenging to design in RL, relies heavily on the admissibility assumption. Without this assumption, it is unclear how an optimistic policy could be constructed either in practice or theoretically.

- In Algorithm 1, functions like REALIZER, VALUEITERATION, ROLLOUT, and the algorithm $\mathfrak{A}$  are not clearly defined. They should be formally described, perhaps in pseudocode, to improve clarity and ensure precise understanding.


- In Theorem 7, the sample complexity scales with
$\bar{A}$, but I could not find a formal definition for $\bar{A}$ in the paper. I guess it refers to the cardinality of the set of action sequences, which is typically much larger than $A$. If this is the case, it’s unclear whether this approach actually improves regret.

**Minor**
- Line 188: The sentence "Any set of options ..." appears incomplete.

**Questions:**

- Line 186: What is the definition of "relevant block"? This term is not clearly defined in the paper.

- Line 239: Could you clarify the statement, "For this reason, we only use 2-MDPs to
represent the abstract MDP, and never a k-MDP with k>2."? The reasoning here is unclear to me.

- Line 246: What is the formal definition of a "new absorbing state"? Additionally, why is this state necessary for defining $\mathcal{S}_{\bar{s}}$?

- Line 265: Could you provide the definition of $\bar{\gamma}$ in Equation (2) and (3)? Why does the inequality $\bar{\gamma} \leq \gamma$ hold?
Furthermore, could you give a clear definition of  $\bar{\mathcal{A}}$?

- Line 403: How are the constraints in Equation (7) derived?



I will consider raising the score once my concerns and questions have been addressed.

---

> ### Author Response · Authors · 2024-11-22
>
> We thank the reviewer for his detailed comments. This really helped us to understand the concerns and to address all of them in detail.
>
> First, a preliminary clarification will be useful for the answers that follow. In the paper, whenever we write a symbol with a top bar, we refer to some given MDP or 2-MDP $\bar{\mathbf{M}} = \langle \bar{\mathcal{S}}, \bar{\mathcal{A}}, \bar{T}, \bar{R}, \bar{\gamma} \rangle$. This will always play the role of an abstract decision process, both in the text and the examples. However, its exact role in the formal statements should be only determined by the quantifiers that appear.
>
> 1. Our contribution can be separated in two parts: Section 3 that shows the properties of our abstractions, and Section 4 which defines the algorithm. Assumption 2, which the reviewer mentions, is only relevant for the algorithm in Section 4, not for realizable abstractions in general. We argue that this requirement is reasonably weak in comparison to (Abel 2020) and (Wen 2020) because of two reasons:
> 	1. Assumption 2 does not require that the input model is a realizable abstraction. For rewards, it only assumes admissibility. As a consequence, the abstract decision process may have arbitrarily large rewards: the algorithm is still guaranteed to converge, regardless of the magnitude of the overestimation. In comparison, (Abel 2020) and (Wen 2020) only consider abstract models that are approximately accurate (according to their own notion of accuracy).
> 	2. Assumption 2 requires the knowledge of an approximate abstract transition function $\bar{T}$. However, differently from the other works, the prior knowledge that our algorithm requires has two desirable features:  it is local to each block and it is at the abstract level (meaning, it does not involve individual states of the ground MDP). This is not true for the cited works. The algorithm executed in (Abel 2020) assumes that a full set of ground options is known for each block. Instead, we do not assume that any policy is known from the start. Also, the most similar algorithm in (Wen 2020) is PEP which assumes that an accurate set of exit profiles is given in input to the algorithm. An exit profile is a value function for the ground exit states of each block. Differently to our inputs, exit profiles are defined on the ground MDP and accurate profiles require global knowledge of the ground MDP.
>
> 2. To understand what admissibility implies and how it can be satisfied, we consider two special cases. Let $\mathbf{M}$ be a goal MDP, meaning that rewards are null everywhere, exept in a terminal state $s_g$, where they equal 1. Then, in the block $\\{s\_g\\}$, we have $V^o_{\phi(s_g)}(s_g) = 1/(1-\gamma)$, while $V^o_{\bar{s}}(s) = 0$ in all other blocks. Then, an admissible abstraction for rewards is one that has $\bar\gamma = \gamma$ and a positive reward associated to the abstract goal $\phi(s_g)$ as $\bar{R}(\phi(s_g)\, \cdot) = 1$. No other constraint on the reward of the other states is required. They may assume any value from 0 to 1. Regarding transition probabilities, to satisfy $\tilde{h}\_{\bar{s} \bar{a}}(\bar{s}') \ge h^{o}\_{\bar{s}}(\bar{s}' \mid s)$ it sufficies that $\bar{T}(\bar{s}' \mid \bar{s}_p \bar{s} \bar{a})$ exceeds the discounted probability of leaving $\lfloor \bar{s} \rfloor$ via some state in $\lfloor \bar{s}' \rfloor$ with any option. One possibility could be to define go-to actions $\bar{\mathcal{A}} \coloneqq \bar{\mathcal{S}}$ and high probabilities for the success of the go-to, such as $\bar{T}(\bar{s}' \mid\bar{s} \bar{a}) = 1$ iff $\bar{s}' = \bar{a}$, 0 otherwise. These values may also be lower in specific cases. For instance, if $\mathbf{M}$ refers to the grid-world of Figure 1, and $\gamma = 0.95$, then, the probability of the transition $\bar{T}(\bar{s}_3 \mid \bar{s}_2 \bar{s}_1 \bar{a})$, with go-to action $\bar{a} = \bar{s}\_3$, may be any value in $[0.57, 1]$ because $\gamma^{11} \approx 0.57$ and any option takes at least 11 steps to complete the abstract "go-to" action. Figure 2 in appendix B of the new pdf contains a second numerical example for a 3-states MDP. As we have seen, admissibility is relatively weak assumption as it often allows a range of probabilities and rewards.
>
> (continues below)

---

> > ### Author Response · Authors · 2024-11-22
> >
> > 3. We will make sure that every term is fully explained in the final revision. To clarify for the reviewing process:
> > 	- "ValueIteration" is the classic planning algorithm for MDPs, as defined in (Puterman 1994 Markov Decision Processes);
> > 	- "Rollout" is a function that completes the episode, in lines 18 and 20, or completes a trajectory until leaving the current block, in lines 11 and 13; Line 13 may also update the internal policy, depending on the algorithm in $\mathfrak{U}$;
> > 	- Each value of the map $\mathfrak{U}$ is an instance of algorithm "Realizer" and it has been defined in line 2;
> > 	- "Realizer" is defined in Assumption 1. We are aware that Assumption 1 only defines the general behavior of the algorithm but not its interface. We will clarify this procedural aspect. The interface is simple: "Realizer.Rollout" samples a trajectory in the block according to any internal policy and may perform arbitrary internal updates; upon leaving the block, it stops. "Realizer.Get" returns the resulting policy.
> >
> > 4. $\bar{A}$ does not refer to sequences of actions. It is the cardinality of the finite set $\bar{\mathcal{A}}$, which is the action space of the abstract decision process $\bar{\mathbf{M}}$.
> >
> >
> > ## Answers:
> > Correctness and self-containment of the paper is very important to us. We hope the answers below address all the reviewer's concerns. If there is any missing aspect, we will make sure to address it.
> >
> > 1. Line 186 (now 188). The adjective “relevant” is not part of the formal definition and it can be safely omitted. We have already removed it in the new version of the paper. The set of all options’ policies for block $\lfloor \bar{s} \rfloor_\phi$ is exactly the set of functions $\lfloor \bar{s} \rfloor_\phi \to \mathcal{A}$, as stated.
> >
> > 2. The paragraph of line 239 (now line 242) explains why we propose 2-MDPs for the abstract decision process. Consider, for example, in Figure 1, how easy it is to re-enter the yellow block after just leaving it, and how hard it is, in comparison, when the option starts close to the green block. This motivates 2-MDPs instead of MDPs, for the general case. In the cited sentence, we argue that the same modeling advantage cannot be obtained when moving from 2-MDPs to 3-MDPs and beyond. In the same domain, a 3-MDP abstraction would model that a transition green->gray->yellow is more probable when the agent enters the green room from a specific previous block. Since the ground decision process is Markovian, this indirect dependency is subtle and it does not seem to justify the additional complexity.
> >
> > 3. Line 246 (now 248). The absorbing state of each block MDP is a new element of the set of states $\mathcal{S}\_{\bar{s}}$ in which a self-loop is the only possible transition. This means that as soon as the agent exists, it must fall into the absorbing state. Thanks to this addition, all the occupancy measures do not account for additional time that would be otherwise spent at the exits. So, their expression simplifies to just the discounted probability of leaving the block $\lfloor \bar{s} \rfloor$ through the exit (without staying there). This is the quantity we are interested in, and which the abstract transition function $\bar{T}$ will be compared to.
> >
> > 4. We intentionally do not define $\bar\gamma$ and $\bar{A}$ in (2) and (3) because these expressions must be applicable to any MDP and 2-MDP $\bar{\mathbf{M}} = \langle \bar{\mathcal{S}}, \bar{\mathcal{A}}, \bar{T}, \bar{R}, \bar{\gamma} \rangle$. If we were to set any of these elements to a concrete value, our definitions and theorems would not be generally applicable. It should be clear, however, that these are the discount factor and the action space of a generic 2-MDP $\bar{\mathbf{M}}$. We have modified the paper to repeat the abstract tuple of the 2-MDP in line 230. This should clear any confusion. The inequality $\bar\gamma \le \gamma$ is true because smaller values of the discount factor are associated with shorter effective horizons. In the general case, the abstraction may preserve the timescale of the ground MDP or compress it, but not enlarge it.
> >
> > 5. The constraint is derived in the paragraph preceding it. The explanation is a bit short due to space constraints. Essentially, since realizability for rewards is expressed in the objective to maximize, it only remains to formulate the constraints for the exit probabilities. In the paragraph, we observe that $h_\nu^o(\bar{s}')$, the block occupancy for the exit $\bar{s}'$, is the sum of the occupancy measure multiplied by an indicator function at the exit. If this indicator is regarded as a reward function of an MDP, then by applying line 175-177 of the paper, this expression equals $V^{\pi_o}_{\bar{s}'}$, the scaled value of the option $o$ in this block MDP. Rearranging the term $\beta$ in equation (6) and dividing by the scale $1-\gamma$ results in the optimization problem that we show.

---

> > > ### Comment · Reviewer_UaD5 · 2024-11-23
> > >
> > > I greatly appreciate the detailed exploration.
> > > For the completeness of the paper, I suggest adding formal definitions and further analysis in the revision to enhance clarity.
> > >
> > > However, I still do not fully understand the definitions of $\bar{A}$ and $\bar{\gamma}$.
> > > Could you provide quantified values for these parameters in the example in Figure 1?
> > > Additionally, please compare these values with the corresponding ones in the original MDP.
> > > I am particularly interested in understanding how large or small these values can be in the worst-case scenario.

---

> > > > ### Author Response · Authors · 2024-11-25
> > > >
> > > > We gladly explain the role of $\bar{\mathcal{A}}$ and $\bar\gamma$ further. First, it is important to remember that, while the ground MDP is usually assumed to be given, we have much more control in designing the abstraction $\langle \bar{\mathbf{M}}, \phi \rangle$. This is because the ground MDP encodes the original task, while the abstraction is manually designed to aid learning. Therefore, there is some freedom in selecting $\bar{\mathcal{A}}$ and $\bar{\gamma}$, which are the abstract action space and the abstract discount factor.
> > > >
> > > > Regarding the range of these parameters, we observe that, in the worst-case scenario, $\bar{\gamma} = \gamma$, because this assignment is always a feasible choice and $\bar{\gamma}$ never needs to be higher (if a suitable abstraction exists, then, there is also one with this choice of $\bar{\gamma}$). On the other hand, we cannot provide an upper bound for $\bar{\mathcal{A}}$, because we can introduce as many abstract actions as we prefer. In general, we argue that a small number of abstract actions suffice, because our framework relates $|\bar{\mathcal{S}}|^2 |\bar{\mathcal{A}}|$ to the number of ground options for each block, which is usually a relatively small number. We can see this with an example.
> > > >
> > > > As a ground MDP, we consider $\mathbf{M} = \langle \mathcal{S}, \mathcal{A}, T, R, \gamma \rangle$, the MDP of Figure 1, where $\mathcal{S}$ is the set of positions of the grid, the actions $\mathcal{A}$ are the 4 cardinal directions, $\gamma = 0.95$, and, just for this example, $T$ can be assumed deterministic and $R$ zero everywhere, initially. We can design many valid abstractions for this domain. For concreteness, we assume that the abstract 2-MDP is $\bar{\mathbf{M}} = \langle \bar{\mathcal{S}}, \bar{\mathcal{A}}, \bar{T}, \bar{R}, \bar{\gamma} \rangle$, for a specific choice of its entries. We select $\bar{\mathcal{S}} = \\{\bar{s}_1, \bar{s}_2, \bar{s}_3\\}$ to represent the set of "rooms", and three abstract actions $\bar{\mathcal{A}} = \\{\bar{a}_1, \bar{a}_2, \bar{a}_3\\}$, one for each room. We will set the transition probabilities so that the abstract actions $\bar{\mathcal{A}}$ play the role of "go-to" behaviors. Specifically, for each $i,j,k$, we set $\bar{T}(\bar{s}\_i \mid \bar{s}\_* \bar{s}\_j, \bar{a}\_k) = 1$, if $i = k$ and room $\bar{s}_i$ is directly connected with $\bar{s}\_j$ (here $\bar{s}\_*$ means "for any previous state"). Transitions have zero probability in all other cases. $\bar{R}$ always returns zero and we set $\bar\gamma = \gamma$. With these choices, we have that $\bar{\mathbf{M}}$ is an admissible and realizable abstraction. We can verify admissibility by considering that if some triple has $\bar{T}(\bar{s}_i \mid \bar{s}_l \bar{s}_j, \bar{a}_k) = 0$, then, no option can directly move from $\bar{s}_j$ to $\bar{s}_i$ in the ground MDP.
> > > >
> > > > If we want a more accurate model, we can alternatively modify the transtion probability $\bar{T}(\bar{s}\_3 \mid \bar{s}\_2 \bar{s}\_1, \bar{a}\_3) = 0.57$, as discussed in the comment above, point 2. Notice that, although the ground MDP is deterministic, the abstract probability can be less than one and still be admissible, in general. In particular, this abstract "go-to" action has a lower probability, not because it may fail in the ground MDP, but because any option will take at least 11 transitions to complete (that is the shortest path from green to yellow). If all options in the ground MDP require multiple steps to terminate, then, instead of lowering all the transitions probabilities of $\bar{\mathbf{M}}$, we can uniformly "scale" along the time dimension, by lowering $\bar\gamma$. We can do this because $\bar{T}$ multiplies $\bar\gamma$ in equation (2). So, another feasible choice is $\bar\gamma = 0.8$ and $\bar{T}(\bar{s}\_3 \mid \bar{s}\_2 \bar{s}\_1, \bar{a}\_3) = 0.677$. These two alternatives are instinguishable in our framework.
> > > >
> > > > As a last example, assume that one specific gray cell generates a positive reward but prevents the agent from leaving the room (for example, it falls into a trapdoor). Then, we cannot use $\bar{a}\_1, \bar{a}\_2, \bar{a}\_3$ to model both the reward collection and the movement. Instead, we should consider adding a new abstract action $\bar{a}\_{\mathsf{r}}$, for which $\bar{R}(\bar{s}_* \bar{s}_1, \bar{a}\_{\mathsf{r}}) = 1$ and $\bar{T}(\bar{s}\_1 \mid \bar{s}\_* \bar{s}\_1, \bar{a}\_{\mathsf{r}}) = 1$, which is a self-loop in the gray room. Now, at the abstract level, the policy will decide whether it is more convenient to follow some option for $\bar{a}\_{\mathsf{r}}$ that collects the reward and stops, or to move to another room with $\bar{a}_3$.
> > > >
> > > > Summarizing, the action space of the abstract decision process models the set of all "options behaviors" that is interesting to realize in the ground MDP. In the example above, $\bar{A} = 4$, regardless of the cardinality of $\mathcal{A}$ and $\mathcal{S}$. The same example would also work if $\mathcal{S}$ was continuous.

---

> > > > > ### Comment · Reviewer_UaD5 · 2024-11-26
> > > > >
> > > > > Thank you for the detailed explanation! The example really helped me understand the role of $\bar{\mathcal{A}}$ and $\bar{\gamma}$.
> > > > >
> > > > > My one last concern is: how can we control the size of  $\bar{\mathcal{A}}$? Since  $\bar{\mathcal{A}}$ represents the set of all "option behaviors," its size can grow exponentially larger than the original action space in the worst-case scenario. Are there any methods to identify or reveal a smaller set of abstract actions?
> > > > > If not, and the size of $\bar{\mathcal{A}}$  remains very large, I worry that Theorem 8 may not be very significant. This is because the sample complexity of your proposed algorithm could be substantially larger than that of the original tabular RL algorithm.

---

> > > > > > ### Author Response · Authors · 2024-11-28
> > > > > >
> > > > > > The abstract action space $\bar{\mathcal{A}}$ never needs to be exponential in size. Indeed, not all options need to be represented at the abstract level. The only option that should be modelled is the optimal one (meaning, its termination probabilities and the cumulative reward). In the previous example, if collecting the reward in the gray block is optimal, then, the abstraction only requires the action $\bar{a}_{\mathsf{r}}$, while all movement actions $\bar{a}_1, \bar{a}_2, \bar{a}_3$ can be omitted.
> > > > > > We can make this statement more precise. Suppose that $\langle \bar{\mathbf{M}}, \phi \rangle$ is a realizable abstraction of $\mathbf{M}$, and we do not pose any restriction on the cardinality of $\bar{\mathcal{A}}$, which may be exponential. Then, there exists another realizable abstraction $\langle \bar{\mathbf{M}}', \phi \rangle$ that has only one abstract action, $\bar{\mathcal{A}}' = \\{\bar{a}^*\\}$. To verify this, we can first solve $\bar{\mathbf{M}}$ and find an optimal policy $\bar{\pi}^*$. Then, in the new model $\bar{\mathbf{M}}'$, the action identified by $\bar{a}^*$ will always be defined as the action that $\bar{\pi}^*$ selects, through a simple renaming. The new abstraction will satisfy the realizability assumption, since $\bar{\mathbf{M}}'$ only contains a subset of actions from $\bar{\mathbf{M}}$. Moreover, since we only removed suboptimal actions, the value of $\bar{\pi}^*$ is preserved, as well as that of its realization in $\mathbf{M}$. This shows that an action space of size one is always sufficient.
> > > > > >
> > > > > > Clearly, we do not know in advance which is the optimal option (or the optimal abstract action) to follow. This is main the reason why we allow to model more than one option behaviour at the abstract level. It will be responsibility of the abstract policy to select the optimal one. However, if some previous knowledge is available, it is always feasible to omit suboptimal option behaviours from $\bar{\mathcal{A}}$.
> > > > > >
> > > > > > Lastly, there is a second motivation for the limited number of actions. We remind that all options with the same external "behavior" can be modelled with the same action. This means that, in the example above, it suffices to have one action $\bar{a}_{\mathsf{r}}$ for collecting the reward, even though there may be multiple ways to collect rewards in the gray room. The abstraction does not need to encode where the reward is collected, and all these options can be regarded as equivalent realizations of the same "reward collection" behaviour. Similarly, it is sufficient to have one movement action $\bar{a}_3$ for reaching the third room, regardless of all the possible ways to reach it.

---

> > > > > > > ### Comment · Reviewer_UaD5 · 2024-12-03
> > > > > > >
> > > > > > > I sincerely appreciate the thoughtful and detailed explanations provided during the discussion period. However, as the discussion has progressed, I find myself leaning toward not supporting this paper.
> > > > > > >
> > > > > > > While the authors explain the motivations for using a limited number of abstract actions, they do not clearly demonstrate how to effectively reduce the size of $\bar{\mathcal{A}}$. To address this, they need to explicitly establish an inequality such as $\bar{S} \bar{A} \ll SA$, as successfully demonstrated by Wen et al., 2020. Without such evidence, the use of an HRL-based approach lacks a strong justification.
> > > > > > >
> > > > > > > Additionally, a fundamental challenge of HRL lies in learning effective options (abstract actions). Assuming the inclusion of an optimal option is a very strong assumption. The authors need to rigorously develop a method to achieve this goal without causing an exponential increase in the size of the abstract action space.
> > > > > > >
> > > > > > > As a result, I will maintain my current score.

---

### Official Review · Reviewer_gPDx · 2024-11-04

**Soundness:** 3
**Presentation:** 3
**Contribution:** 4
**Rating:** 6
**Confidence:** 2

**Summary:**

This paper proposes Realizable Abstractions as a method of relating low-level MDPs to high level abstractions, in particular ones that are suitable for hierarchical reinforcement learning.
The approach provides near-optimality guarantees for low-level options which realize higher-level abstracted behavior and are the solutions to specially constructed constrained MDPs.
A PAC algorithm is presented which is modular with respect to a PAC-Safe online learning algorithm which is used as a subroutine.

**Strengths:**

The paper addresses an important problem in hierarchical RL dealing with abstractions which relate high-level and low-level representations.

Connecting hierarchical abstractions to constrained MDPs such that options can be extracted by solving the CMDPs with off-the-shelf algorithms is interesting and, to my knowledge, novel.

The paper is well-written and the intuitive explanations for the theory are fairly easy to follow, though it is extremely notation-heavy.

**Weaknesses:**

While an algorithm is proposed (RARL), there are no empirical results to support it and validate the assumptions that are made for the guarantees in Section 4. I would like to see RARL compared with existing methods, e.g., some form of option-critic (with specified options) or deep skill chaining (for a skill discovery comparison). Additionally, as it is not clear to me how reasonable Assumptions 1-3 are, the paper would be strengthened by experiments showing how RARL is affected by violations of those assumptions.

**Questions:**

Can you provide examples of algorithms for which Assumption 1 holds? What properties need to hold for such an algorithm?

How might one construct abstractions suitable for RARL ensuring that Assumptions 2 and 3 hold (or verify that they hold for a given abstraction)? Related to the experiment described above, how do things change with as violations of the assumptions become larger? Does RARL respond gracefully?

---

> ### Author Response · Authors · 2024-11-22
>
> We thank the reviewer for his thoughtful feedback. Please, find our answer to the concerns below.
>
> We can think of the two main sections of the paper, 3 and 4, as having independent contributions: showing the theoretical properties of our realizable abstractions, and developing a hierarchical algorithm with formal correctness guarantees. Both contributions have been demonstrated theoretically. However, specifically to the second contribution (the one of Section 4), we agree with the reviewer that an experimental evaluation would help to show the practical performances of the specific algorithm we propose. What we can do in this comment, is to discuss how the algorithm concretely behaves and how strict the assumptions are.
>
> ### Assumption 1
>
> The first assumption requires that the Safe-RL algorithm we choose to apply in RARL is Probably Approximately Correct (PAC) for constrained MDPs. PAC is a common formalism for stating the correctness and efficiency of learning algorithms. In this context, the returned policy should be $\zeta$-optimal and have an expected maximum violation for all constraints of at most $\eta$, if a feasible policy exists. This assumption is required by the theoretical analysis of Theorem 8, because if any of the sub-routines we use are not correct, the whole algorithm cannot be PAC. However, this is not motivated by practical reasons: virtually all Safe-RL algorithms can be applied. Two interesting examples are CPO (Achiam 2017) and FOCOPS (Zhang 2020). They provide theoretical guarantees in the form of monotonic improvements among the trajectory of feasible policies. These are similar to the performance guarantee of TRPO. Although these are not end-to-end performance guarantees as expressed in Assumption 1, FOCOPS remains a good candidate for implementing the Safe RL algorithm, similarly to how TRPO and PPO may be applied when one requires a generic RL algorithm as a sub-routine.
>
> For more recent results about the theory of Safe RL algorithms, the reviewer may also refer to (Yang 2022).
>
> References:
> - Achiam et al. 2017. "Constrained policy optimization". ICML.
> - Yang et al. 2022. "Constrained Update Projection Approach to Safe Policy Optimization". NeurIPS.
> - Zhang et al. 2020. "First Order Constrained Optimization in Policy Space". NeurIPS.
>
>  (continues below)

---

> > ### Author Response · Authors · 2024-11-22
> >
> > ### Assumption 2
> >
> > The second assumption can be broken down in two separate requirements on the abstraction $\langle \bar{\mathbf{M}}, \phi \rangle$: admissibility and realizability. While admissibility should be satisfied strictly, realizability should not.
> >
> > - To understand what admissibility implies and how it can be satisfied, we consider two special cases. Let $\mathbf{M}$ be a goal MDP, meaning that rewards are null everywhere, exept in a terminal state $s_g$, where they equal 1. Then, in the block $\\{s\_g\\}$, we have $V^o_{\phi(s_g)}(s_g) = 1/(1-\gamma)$, while $V^o_{\bar{s}}(s) = 0$ in all other blocks. Then, an admissible abstraction for rewards is one that has $\bar\gamma = \gamma$ and a positive reward associated to the abstract goal $\phi(s_g)$ as $\bar{R}(\phi(s_g)\, \cdot) = 1$. No other constraint on the reward of the other states is required. They may assume any value from 0 to 1. Regarding transition probabilities, to satisfy $\tilde{h}\_{\bar{s} \bar{a}}(\bar{s}') \ge h^{o}\_{\bar{s}}(\bar{s}' \mid s)$ it sufficies that $\bar{T}(\bar{s}' \mid \bar{s}_p \bar{s} \bar{a})$ exceeds the discounted probability of leaving $\lfloor \bar{s} \rfloor$ via some state in $\lfloor \bar{s}' \rfloor$ with any option. One possibility could be to define go-to actions $\bar{\mathcal{A}} \coloneqq \bar{\mathcal{S}}$ and high probabilities for the success of the go-to, such as $\bar{T}(\bar{s}' \mid\bar{s} \bar{a}) = 1$ iff $\bar{s}' = \bar{a}$, 0 otherwise. These values may also be lower in specific cases. For instance, if $\mathbf{M}$ refers to the grid-world of Figure 1, and $\gamma = 0.95$, then, the probability of the transition $\bar{T}(\bar{s}_3 \mid \bar{s}_2 \bar{s}_1 \bar{a})$, with go-to action $\bar{a} = \bar{s}\_3$, may be any value in $[0.57, 1]$ because $\gamma^{11} \approx 0.57$ and any option takes at least 11 steps to complete the abstract "go-to" action. Figure 2 in appendix B of the new pdf contains a second numerical example for a 3-states MDP.
> > This answer explains how strict Assumption 2 is with respect to admissibility. As we have seen, it is relatively weak as it allows a range of probabilities and rewards. However, if this was falsified, RARL would be falsely biased to believe that some ground states have lower rewards than what they actually have, and these may not be explored at all by the algorithm. This is intentional: if the abstraction is admissible, then it can be used as an heuristic to ignore large regions of the ground MDP that have low value.
> >
> > - Realizability on rewards, on the other hand, should not be strict. The algorithm will always converge, regardless of the magnitude of the overestimation for the block values. The assumption only requires that an unknown $(\alpha, \beta$)-realizable abstraction exists over the same mapping $\phi$. The actual input of the algorithm $\langle \bar{\mathbf{M}}, \phi \rangle$ may overestimate the block values arbitrarily. In the worst case, when all rewards of $\bar{\mathbf{M}}$ are set to 1, RARL explores the blocks similarly to how an uninformed R-MAX algorithm would explore the discrete states.
> >
> > ### Assumption 3
> >
> > Assumption 3 is a technical requirement that allows us to ignore an indirect dependency in our analysis. As we will see, this is relatively mild. Suppose that at time $t$ we have just learned an option $o_1$ for a block $\lfloor \bar{s} \rfloor$. Now, consider a ground block $\lfloor\bar{s}'\rfloor$ that is reachable from $\lfloor\bar{s}\rfloor$. When learning an option $o'$ in block $\lfloor \bar{s}' \rfloor$, the trajectories will always start from the states that $o_1$ reaches in $\lfloor \bar{s}' \rfloor$. The probability distribution of these initial states is written $\nu_{t,\bar{s}\bar{s}'}$ and we say that $o'$ is a realization from $\nu_{t,\bar{s}\bar{s}'}$. Now, if we add a new option $o_2$ to those available in $\lfloor\bar{s}\rfloor$, then the entry distribution in $\lfloor \bar{s}' \rfloor$ will be a mix of the probability induced by $o_1$ and $o_2$. Without Assumption 3, it is possible to construct very specific corner cases, in which $o_1$ and $o_2$ reach different entry states in $\lfloor \bar{s}' \rfloor$, say $s_1'$ and $s_2'$. If these states are also separated within the block (we cannot reach $s_1'$ from $s_2'$ within $\lfloor \bar{s}' \rfloor$, and vice versa), then, at the time $t'$ when we learn $o_2$, the old option $o'$ may not be a realization from the new entry distributon $\nu_{t',\bar{s}\bar{s}'}$, because $s_2'$ has never been experienced before.
> >
> > In practice, this issue is never encountered if the states of each block are connected and they can be explored by the Safe-RL algorithm. The produced options will be realizations from any entry state. In addition: in the 4-rooms domain, frequently used in HRL such as (Abel 2020), Assumption 3 is satisfied; more in general, in any MDP in which separate blocks are connected by a single state, this assumption is also satisfied.

---

> > > ### Comment · Reviewer_gPDx · 2024-12-03
> > >
> > > Thank you for your responses. This gives me a better idea of how the assumptions might hold in an existing domain or be used as guidance for designing/adjusting one.
> > >
> > > It would seem that there is currently no known algorithm for which Assumption 1 would hold. I understand that assumptions made for the purpose of theoretical analysis can often be relaxed to some extent while still getting something close to the theoretical guarantees. However, without empirical evidence to demonstrate that realistic violations of the assumptions have an acceptable impact, it is hard to be convinced that RARL is practical.

---

### Official Review · Reviewer_mzzn · 2024-11-06

**Soundness:** 4
**Presentation:** 4
**Contribution:** 4
**Rating:** 8
**Confidence:** 4

**Summary:**

The goal of HRL is to replace an MDP with another simpler decision process, solve this MDP abstraction and from this solution, reconstruct a solution in the base MDP. The authors are motivated and propose a solution to what they identify as shortcommings of the currently proposed HRL schemes: first the MDP abstraction itself is often non-Markovian, and second there is rarely any theoretical guarantee on the optimality of the policy reconstructed from an optimal policy on the MDP abstraction.
The authors consider a type of MDP abstraction they call "realisable abstraction" that supposes any abstract action has a "local" realisation with similar occupancy measure and value. By compositionality, this is then shown (theorem 1) to imply a similar "global" statement in the form of a bound on the difference between the values of an abstract policy and its realisation.
The authors then present an HRL algorithm that is shown to PAC learn the optimal policy under some assumptions.

**Strengths:**

Very clear, very well written.

**Weaknesses:**

I appreciated the example illustrated in figure 1. I would suggest the authors refer to it when introducing new concepts and when discussing RARL.

**Questions:**

I would like to see how the realisability condition compares to the bisimulation relation for MDPs
https://arxiv.org/pdf/1207.4114

---

> ### Author Response · Authors · 2024-11-22
>
> We thank the reviewer for the positive comments. In the final revision, we will make sure to use Figure 1 to illustrate the main steps of the algorithm.
>
> The bisimulation relation is indeed a relevant reference for our work. However, as demonstrated by Ravindran (2004, Theorem 6 and corollary), stochastic bisimulation has exactly the same expressive power as MDP homomorphisms. Therefore, we can conclude that realizable abstractions are strictly more expressive than bisimilarity, because the same is true for MDP homomorphisms. To make these statements more precise, we have updated our paper to add two new propositions in Appendix B: Proposition 6 and 12 (these numbers refer to the updated pdf). Proposition 5 already proved that realizable abstractions are at least as expressive as MDP homomorphisms. The new Proposition 6 proves that this containment is strict, because all MDP homomorphisms are realizable abstractions but the opposite is not true. Finally, Proposition 12 proves that the same is also true for bisimilarity.
>
> Reference:
> Balaraman Ravindran. An Algebraic Approach to Abstraction in Reinforcement Learning. PhD thesis, 2004.

---

### Meta-Review · Area_Chair_qoW7 · 2024-12-21

**Metareview:**

This paper studies Realizable Abstractions in Hierarchical Reinforcement Learning, introducing a new theoretical framework and the RARL algorithm, which guarantees near-optimal policies under specific assumptions. While the theoretical contributions are intriguing, reviewers raised significant concerns about the strong assumptions required for the proposed method, lack of empirical validation, and unclear feasibility of scaling the abstract action space. During the rebuttal phase, the authors provided detailed clarifications but failed to alleviate key doubts about the practicality and novelty of the approach. Therefore, the reviewers are not convinced that the paper meets the standards for acceptance.

**Additional Comments On Reviewer Discussion:**

During the rebuttal period, reviewers raised concerns about the practicality of the strong assumptions underpinning the proposed algorithm, particularly around the admissibility and realizability of abstractions, the scalability of the abstract action space, and the lack of empirical validation. Reviewer UaD5 questioned the feasibility of managing the abstract action space without exponential growth and highlighted unclear definitions of key parameters, while Reviewer gPDx emphasized the need for empirical results to validate theoretical assumptions. Reviewer JLGr acknowledged the paper's theoretical contributions but also sought more clarity on identifying suitable abstractions and empirical support. The authors provided detailed responses, including examples and expanded explanations, and clarified procedural details and theoretical guarantees. However, key issues, such as the practicality of the assumptions and scalability concerns, remained unresolved.

---

### Decision · Program_Chairs · 2025-01-22

Reject